# Evaluation of terrestrial pan-Arctic carbon cycling using a data-assimilation system

Efrén López-Blanco[1,2], Jean-François Exbrayat[2,3], Magnus Lund[1], Torben R. Christensen[1,4], Mikkel P. Tamstorf[1], Darren Slevin[2], Gustaf Hugelius[5], Anthony A. Bloom[6], Mathew Williams[2,3]

[1] Department of Biosciences, Arctic Research Center, Aarhus University, Frederiksborgvej 399, 4000 Roskilde, Denmark
[2] School of GeoSciences, University of Edinburgh, Edinburgh, EH93FF, UK
[3] National Centre for Earth Observation, University of Edinburgh, Edinburgh, EH9 3FF, UK
[4] Department of Physical Geography and Ecosystem Science, Lund University, Sölvegatan 12, 223 62 Lund, Sweden
[5] Department of Physical Geography and Bolin Centre for Climate Research, Stockholm University, 106 91 Stockholm, Sweden
[6] Jet Propulsion Laboratory, California Institute of Technology, Pasadena, CA 91109, USA

*Correspondence to*: Efrén López-Blanco (elb@bios.au.dk) and Mathew Williams (mat.williams@ed.ac.uk)

**Keywords:** Arctic tundra, Arctic taiga, Net Ecosystem Exchange, Gross primary production, Ecosystem Respiration, carbon stocks, transit times, field observations, global vegetation models.

**Abstract.** There is a significant knowledge gap in the current state of the terrestrial carbon (C) budget. Recent studies have highlighted poor understanding particularly of C pool transit times, and whether productivity or biomass dominate these biases. The Arctic, accounting for approximately 50% of the global soil organic C stocks, has an important role in the global C cycle. Here, we use the CARDAMOM data-assimilation system to produce pan-Arctic terrestrial C cycle analyses for 2000-15. This approach avoids using traditional plant functional type or steady-state assumptions. We integrate a range of data (soil organic C, leaf area index, biomass, and climate) to determine the most likely state of the high latitude C cycle at a 1° x 1° resolution, and also to provide general guidance about the controlling biases in transit times. On average, CARDAMOM estimates regional mean rates of photosynthesis of 565 g C m$^{-2}$ yr$^{-1}$ (90% confidence interval between the 5th and 95th percentiles: 428, 741), autotrophic respiration of 270 g C m$^{-2}$ yr$^{-1}$ (182, 397) and heterotrophic respiration of 219 g C m$^{-2}$ yr$^{-1}$ (31, 1458), suggesting a pan-Arctic sink of -67 (-287, 1160) g C m$^{-2}$ yr$^{-1}$, weaker in tundra and stronger in taiga. However, our confidence intervals remain large (and so the region could be a source of C), reflecting uncertainty assigned to the regional data products. We show a clear spatial and temporal agreement between CARDAMOM analyses and different sources of assimilated and independent data at both pan-Arctic and local scales, but also identify consistent biases between CARDAMOM and validation data. The assimilation process requires clearer error quantification on LAI and biomass products to resolve these biases. Mapping of vegetation C stocks and change over time, and soil C ages linked to soil C stocks is required for better analytical constraint. Comparing CARDAMOM analyses to global vegetation models (GVM) for the same period, we conclude that transit times of vegetation C are inconsistently simulated in GVMs due to a combination of uncertainties from productivity and biomass calculations. Our findings highlight that GVMs need to focus on constraining both current vegetation C stocks and net primary production, to improve process-based understanding of C cycle dynamics in the Arctic.

**1 Introduction**

Arctic ecosystems play a significant role in the global carbon (C) cycle (Hobbie et al., 2000; McGuire et al., 2012). Slow organic matter decomposition rates due to cold and poorly drained soils in combination with cryogenic soil processes have led to an accumulation of large stocks of C stored in the soils, much of which is currently held in permafrost (Tarnocai et al., 2009). The permafrost region soil organic C (SOC) stock is more than twice the size of the atmospheric C stock; and accounts for approximately half of the global SOC stock (Hugelius et al., 2014; Jackson et al., 2017). High latitude ecosystems
are experiencing a temperature increase that is nearly twice the global average (AMAP, 2017). The expected future increase of temperature (IPCC, 2013) and precipitation (Bintanja and Andry, 2017) will likely have consequences for the Arctic net C balance. As high latitudes warm, C cycle dynamics may lead to an increase of carbon dioxide ($CO_2$) emissions through ecosystem respiration ($R_{eco}$) driven by, for example, larger heterotrophic respiration (Commane et al., 2017; Schuur et al., 2015; Zona et al., 2016), drought stress on plant productivity (Goetz et al., 2005) and episodic disturbances (Lund et al., 2017;
Mack et al., 2011). Alternatively, temperature-induced vegetation changes (Forkel et al., 2016; Graven et al., 2013; Lucht et al., 2002) may increase gross primary productivity (GPP) due to extended growing seasons (Zeng et al., 2011), $CO_2$ fertilization (Zhuang et al., 2006) and shifts in vegetation cover such as greening (Myneni et al., 1997; Zhu et al., 2016) and shrub expansion (Myers-Smith et al., 2011). Consequently, ecosystem responses may feedback on climate with unclear magnitude and sign (Anav et al., 2013; Murray-Tortarolo et al., 2013; Peñuelas et al., 2009). As a result of the significant changes that are already
affecting the structure and function of Arctic ecosystems, it is critical to understand and quantify the historical C dynamics of the terrestrial tundra and taiga, and their sensitivity to climate (McGuire et al., 2012).

      Although the land surface is estimated to offset ~30% of anthropogenic emissions of $CO_2$ (Canadell et al., 2007; Le Quéré et al., 2018), the terrestrial C cycle is currently the least constrained component of the global C budget and large uncertainties remain (Bloom et al., 2016). Despite the importance of Arctic tundra and taiga biomes in the global land C cycle,
our understanding of interactions between the allocation of C from net primary productivity (NPP), C stocks ($C_{stock}$), and transit times (TT), is deficient (Carvalhais et al., 2014; Friend et al., 2014; Hobbie et al., 2000). The TT is a concept that represents the time it takes for a particle of C to persist in a specific C stock and it is defined by the C stock and its outgoing flux, here addressed as $TT = C_{stock} / NPP$. According to a recent study by Sierra et al. (2017), TT is an important diagnostic metric of the C cycle and a concept that is independent of model internal structure and theoretical assumptions for its calculation. Terms
such as residence time (Bloom et al., 2016; Friend et al., 2014), turnover time (Carvalhais et al., 2014), and turnover rate (Thurner et al., 2016; TT = 1/turnover rate) are used in the literature to represent the concept of TT (Sierra et al., 2017). Studies have focused more on the spatial variability with climate of ecosystem productivity rather than C transit times (Friend et al., 2014; Nishina et al., 2015; Thurner et al., 2016; Thurner et al., 2017). Friend et al. (2014) detailed that transit time dominates uncertainty in terrestrial vegetation responses to future climate and atmospheric $CO_2$. They found a 30% larger variation in
modelled vegetation C change than response of NPP. Nishina et al. (2015) also suggested that long term C dynamics within ecosystems (vegetation turnover and soil decomposition) are more critical factors than photosynthetic processes (i.e. GPP or NPP). The respective contribution of bias from biomass and NPP to biases in transit times remains unquantified. Without an appropriate understanding of current state and dynamics of the C cycle, its feedbacks to climate change remains highly uncertain (Hobbie et al., 2000; Koven et al., 2015).

There are currently efforts to incorporate both in-situ and satellite-based datasets to assess C cycle retrievals and to reduce their uncertainties. At local scale, the net ecosystem exchange (NEE) of $CO_2$ between the land surface and the atmosphere is usually measured using eddy covariance (EC) techniques (Baldocchi, 2003). International efforts have led to the creation of global networks such as FLUXNET (http://fluxnet.fluxdata.org/) and ICOS (https://www.icos-ri.eu/), to harmonise data and support the reduction of uncertainties around the C cycle and its driving mechanisms. However, upscaling
field observations to estimate regional to global C budget presents important challenges due to insufficient spatial coverage of measurements and heterogeneous landscape mosaics (McGuire et al., 2012). Furthermore, harsh environmental conditions in

high latitude ecosystems and their remoteness complicates the collection of high-quality data (Lafleur et al., 2012). Given the lack of continuous, spatially distributed in situ observations of NEE in the Arctic, it remains a challenging task to calculate with certainty whether or not the Arctic is a net C sink or a net C source, and how the net C balance will evolve in the future (Fisher et al., 2014). Over the past decade, regional to global products generated from in situ networks and/or satellite observations have improved our understanding of the terrestrial C dynamics. These range from machine-learning based upscaling of FLUXNET data (Jung et al., 2017), remotely-sensed biomass products (Carvalhais et al., 2014; Thurner et al., 2014) and the creation of a global soil database (FAO/IIASA/ISRIC/ISSCAS/JRC, 2012). Due to a reliance on interpolation and upscaling with other spatial data, it is challenging to evaluate these products for inherent biases.

Global Vegetation Models (GVM) have been developed to determine global terrestrial C cycling, through representing vegetation and soil processes, including vegetation dynamics (i.e. growth, competition, and turnover) and biogeochemical (i.e. water, carbon, and nutrients cycling) responses to climate variability (Koven et al., 2011; Sitch et al., 2003; Woodward et al., 1995). The advantage of using process-based models to characterise C dynamics is that processes which drive ecosystem-atmosphere interactions can be simulated and reconstructed when data is scarce. However, C cycle modelling in GVMs typically relies on parameters retrieved from literature, prescribed plant-functional-type (PFT) and a spin-up process ensuring C stocks (biomass and SOC) reach steady state. Further, inherent differences of model structure contribute more significantly to GVM uncertainties (Exbrayat et al., 2018; Nishina et al., 2014), than do differences in climate projections (Ahlström et al., 2012). Many model inter-comparison projects have demonstrated a lack of coherence in future projections of terrestrial C cycling (Ahlström et al., 2012; Friedlingstein et al., 2014). Recent studies have used simulations from the first phase of the Inter-Sectoral Impact Model Inter-comparison Project (ISI-MIP) (Warszawski et al., 2014) to evaluate the importance of key elements regulating vegetation C dynamics, but also the estimated magnitude of their associated uncertainties (Exbrayat et al., 2018; Friend et al., 2014; Nishina et al., 2014; Nishina et al., 2015; Thurner et al., 2017). An important insight is that TTs in GVMs are a key uncertain feature of the global C cycle simulation. Further, GVMs tend not to report uncertainties in their estimates of stocks and fluxes, which weakens their analytical value.

To address these issues we integrate model and data more formally. We apply data assimilation (DA), defined as a Bayesian calibration process for a model of a dynamic system. DA, through probabilistic parameterisation, supports robust model estimates of C stocks and fluxes consistent with multiple observations and their errors (Fox et al., 2009; Luo et al., 2009; Williams et al., 2005). By following Bayesian methods, the uncertainty on observations weights the degree of data constraint, and the outcome is a set of acceptable parameterisations for a given model structure linked to likelihoods. Overall, this approach determines whether model structure, observations and forcing are (in)consistent, and thus assesses validity of model structure. By assimilating co-located climatic, ecological and biogeochemical data from remote sensing observations at a specific grid scale across landscapes and regions DA can map parameter estimation and uncertainties.

Here, we use the CARbon DAta MOdel framework (CARDAMOM) (Bloom et al., 2016; Bloom and Williams, 2015; Smallman et al., 2017) to analyse the pan-Arctic terrestrial carbon cycle at 1º resolution for the 2000-2015 period, We assimilate gridded observations of LAI, biomass and SOC stocks at these spatio-temporal scales into an intermediate complexity C model (DALEC2, which is less complex than GVMs). We compare analyses of C dynamics of Arctic tundra and taiga against (a) global products of GPP (Jung et al., 2017) and heterotrophic respiration ($R_h$) (Hashimoto et al., 2015); (b) NEE, GPP and $R_{eco}$ field observations from 8 high latitude sites included in the FLUXNET2015 dataset, and (c) 6 GVMs from the ISI-MIP2a comparison project (Akihiko et al., 2017). Our objectives are to (1) present and evaluate the analyses and uncertainties of the current state of the pan-Arctic terrestrial C cycling using a DA system, (2) quantify the degree of agreement between the CARDAMOM product with local to global scale sources of available data to assess analytical bias; and (3) use CARDAMOM as a benchmarking tool for the ISI-MIP2a models, to provide general guidance towards GVM improvements in transit time simulation. Finally, we suggest future work to be done in the context of advancing pan-Arctic C cycle modelling.

## 2 Data and methods

### 2.1 Pan-Arctic region

The spatial domain we considered in this study (Figure S1) corresponds to the extent of the Northern Circumpolar Soil Carbon Database version 2 (NCSCDv2) dataset (Hugelius et al., 2013a; Hugelius et al., 2013b), bounded by latitudes 42°N - 80°N and longitudes 180°W - 180°E, and at a spatial resolution of 1° x 1°. This area of study totals 18.0 million km$^2$ of land area. We used the GlobCover vegetation map product developed by the European Space Agency (Bontemps et al., 2011)

to separate regions dominated by non-forested (tundra) and forested (taiga) land cover types. A complete description of the classes included in each domain can be found in Figure S1 and caption. The differentiation between tundra and taiga grid cells is in agreement with the tree line delimited by Brown et al. (1997) together with the tundra domain defined from the Regional Carbon Cycle Assessment and Processes Activity reported by McGuire et al. (2012). The extensive grasslands without presence of trees in some areas such as the in South Russia, Mongolia and Kazakhstan were neglected to focus on higher

latitudes. This classification of tundra and taiga totals 8.1 and 9.9 million km$^2$ of land area, respectively.

### 2.2 The CARbon DAta MOdel framework

      Here we use the CARbon DAta MOdel framework (CARDAMOM; Bloom et al., 2016) (list of acronyms can be found in Table S1) to retrieve terrestrial C cycle dynamics, including explicit confidence intervals, in the pan-Arctic region. CARDAMOM consist of two key components: (1) an ecosystem model, the Data Assimilation Linked Ecosystem Carbon

version 2 (DALEC2) (Bloom and Williams, 2015; Williams et al., 2005), constrained by observations and (2) a data-assimilation system (Bloom et al., 2016). This framework reconciles observational datasets as part of a representation of the terrestrial C cycle in agreement with ecological theory.

### 2.2.1 DALEC2

      The DALEC2 ecosystem model simulates monthly land-atmosphere C fluxes and the evolution of six C stocks

(foliage, labile, wood, roots, soil organic matter (SOM) and surface litter) and corresponding fluxes. DALEC2 includes 17 parameters controlling the processes of plant phenology, photosynthesis (GPP), allocation of primary production to respiration and vegetation carbon stocks, plant and organic matter turnover rates, all established within specific prior ranges based on ecologically viable limits (Table S2; most priors are uniform with broad ranges). DALEC2 simulates canopy-level GPP via the Aggregated Canopy Model (ACM; Williams et al., 1997) and the most sensitive ACM parameter, related to canopy

photosynthetic efficiency, is included in the CARDAMOM calibration. DALEC allocates net primary production to the four plant stocks (foliage, labile, wood and roots) and autotrophic respiration ($R_a$) as time-invariant fractions of GPP. Plant C decays into litter and soil stocks where microbial decomposition generates heterotrophic respiration ($R_h$). Turnover of litter and soil stocks is simulated using temperature dependent first-order kinetics. For practical purposes we aggregated the different C stocks into photosynthetic ($C_{photo}$; leaf and labile), vegetation ($C_{veg}$; leaf, labile, wood and roots), soil ($C_{dom}$; litter and SOM)

and total ($C_{tot} = C_{photo} + C_{veg} + C_{dom}$) C stocks. The Net Ecosystem Exchange (NEE) is calculated as the difference between GPP and the sum of the respiration fluxes ($R_{eco} = R_a + R_h$), while Net Primary Productivity (NPP) is the difference between GPP and $R_a$. Only NEE follows the standard micrometeorological sign convection presenting the uptake of C as negative (sink), and the release of C as positive (source); both GPP and $R_{eco}$ are reported as positive fluxes. In this study, we addressed C turnover rates and decomposition processes as their inverse rates, this is the C transit time ($TT_{photo}$, $TT_{veg}$ and $TT_{dom}$),

represented as the ratio between the mean C stock and the mean C input into that stock during the simulation period.

**2.2.2 Data-assimilation system**

The intermediate complexity of the DALEC2 model compared to typical GVMs facilitates computationally intense Monte-Carlo (MC) data-assimilation to optimize the initial stock conditions and the 17 process parameters that shape C dynamics. CARDAMOM is forced with climate data from the European Centre for Medium-Range Weather Forecast Reanalysis interim (ERA-interim) dataset (Dee et al., 2011) monthly for the 2000-2015 period. A Bayesian Metropolis-Hastings Markov chain MC (MHMCMC) algorithm is used to retrieve the posterior distributions of the process parameters according to observational constraints and Ecological and Dynamic constraints (EDCs; Bloom and Williams, 2015). EDCs ensure that DALEC2 simulations of the terrestrial carbon cycle are realistic and ecologically viable and help to reduce the uncertainty in the model parameters by rejecting estimations that do not satisfy different conditions applied to C allocation and turnover rates as well as trajectories of C stocks.

Observational constraints include monthly time series of Leaf Area Index (LAI) from the MOD15A2 product (Myneni et al., 2002), estimates of vegetation biomass (Carvalhais et al., 2014) and soil organic carbon content (Hugelius et al., 2013a; Hugelius et., 2013b) (Table S3). We aggregated ~130000 1-km resolution MODIS LAI data monthly within each 1°x 1° pixel. Biomass based on remotely-sensed forest biomass (Thurner et al., 2014) and upscaled GPP (Jung et al., 2011) covering the pan-Arctic domain was aggregated to 1° resolution (Carvalhais et al., 2014). We used the NCSCD spatial explicit product (Hugelius et al., 2013a; Hugelius et al., 2013b) which was generated from 1778 soil sample locations interpolated to a 1° grid.

We apply the setup described above to 3304 1° x 1° pixels (1686 in tundra; 1618 in taiga) using a monthly time step. Each pixel is treated independently without assuming a prior land cover or plant functional type and we assume no spatial correlation between uncertainties in all pixels. In each 1° x 1° pixel, we applied the MHMCMC algorithm to determine the probability distribution of the optimal parameter set and initial conditions ($x_i$; Table S2) given observational constraints ($O_i$; LAI, SOC and biomass, Table S3) using the same Bayesian inference approach described in Bloom et al. (2016):

$$p(x_i|O_i) \propto p(x_i)\, p(O_i|x_i) \tag{1}$$

First, in the expression 1, $p\,(x_i)$ represents the prior probability distribution of each DALEC2 parameter ($x_i$) and is expressed as:

$$p(x_i) = p_{EDC}(x_i)\; e^{-0.5\left(\frac{\log(f_{auto})-\log(0.5)}{\log(1.2)}\right)^2}\; e^{-0.5\left(\frac{\log(C_{eff})-\log(17.5)}{\log(1.2)}\right)^2} \tag{2}$$

where $p_{EDC}(x_i)$ is the prior parameter probability according to the EDCs included in Table S2 and described in Bloom and Williams (2015). In addition, prior values for two parameters and their uncertainties (canopy efficiency [$C_{eff}$] and fraction of GPP respired [$f_{auto}$]) are imposed with a log-normal distribution following Bloom et al. (2016) to be consistent with the global GPP range estimated in Beer et al. (2010) and $f_{auto}$ ranges specified by DeLucia et al. (2007) respectively.

Second, $p(O|x_i)$ from expression 1 represents the likelihood of $x_i$ with respect to $O_i$, and it is calculated based on the ability of DALEC2 to reproduce (1) biomass (Carvalhais et al., 2014), (2) SOC (Hugelius et al., 2013a, Hugelius et al., 2013b), and (3) MODIS LAI (Myneni et al., 2002). The reported uncertainty on biomass data from Thurner et al. (2014) was +/- 37% at pixel scale. Because of undetermined errors related to tree cover thresholds used in the upscaling, and to reflect unknown model structural error, we slightly inflate the error estimate and use a log-transform(1.5) of ×/÷1.5 (i.e. ×/÷1.5 spans 67% of the expected error). We use the same proportional error for SOC. For MODIS LAI we inflate the proportional error further to log(2) based on well reported biases in this product for evergreen forests (De Kauwe et al., 2011) and the estimated measurement and aggregation uncertainty for boreal forest LAI of 1 m$^2$ m$^{-2}$ reported by Goulden et al. (2011). The uncertainty

assumptions in expression 3 are chosen in lack of better knowledge about the combined uncertainties arising from model representation errors and observation errors:

$$p(O_i|x_i) = e^{-0.5\left(\frac{\log(O_{biomass})-\log(M_{biomass,0})}{\log(1.5)}\right)^2} e^{-0.5\left(\frac{\log(O_{SOC})-\log(M_{SOC,0})}{\log(1.5)}\right)^2} e^{-0.5\left(\frac{\log(O_{LAI,t})-\log(M_{LAI,t})}{\log(2)}\right)^2} \qquad (3)$$

For each 1º x 1º pixel we run three MHMCMC chains with $10^7$ accepted simulations each until convergence of at least two chains. We use 500 parameter sets sampled from the second half of each chain to describe the posterior distribution of parameter sets. We produce confidence intervals of terrestrial C fluxes and stocks from the selected parameter sets. In the following we report highest confidence results (median; P50) and the uncertainty represented by the 90% confidence interval (5$^{th}$ percentile to 95$^{th}$ percentile, $\binom{P95}{P05}$). We calculate the transit time for C pools using the approach for non-steady state pools described in Bloom et al. (2016), supplementary information S3.

## 2.3 Model evaluation against independent in situ and pan-Arctic datasets

At the pan-Arctic scale, we compared CARDAMOM GPP with FLUXCOM dataset from Jung et al. (2017). We also compared our CARDAMOM $R_h$ with the global spatiotemporal distribution of soil respiration from Hashimoto et al. (2015) calculated by a climate-driven empirical model. To assess the degree of statistical agreement we calculated linear goodness-of-fit (slope, intercept, $R^2$) between CARDAMOM and the two independent datasets and determined RMSE and bias from direct comparison on model-data residuals. The mapping includes stipples representing locations where the independent datasets are within CARDAMOM's 90% confidence interval.

At a local scale, we compare CARDAMOM NEE and its partitioned components GPP and $R_{eco}$ estimates against monthly aggregated values from the FLUXNET2015 sites. We selected 8 sites (Belelli Marchesini et al., 2007; Bond-Lamberty et al., 2004; Goulden et al., 1996; Ikawa et al., 2015; Kutzbach et al., 2007; López-Blanco et al., 2017; Lund et al., 2012; Sari et al., 2017) located across sub- and high-Arctic latitudes, covering locations with different climatic conditions and dominating ecotypes (Table S4). For this evaluation, we compared the same years for both observations and CARDAMOM, and we selected data using day-time method (Lasslop et al., 2010) due to the absence of true night-time period during Arctic summers in some locations. Additionally, we selected a variable u∗ threshold to identify insufficient turbulence wind conditions from year to year similar to López-Blanco et al. (2017). In this data-model comparison we included the median (P50) ± the 90% confidence interval (percentile 5$^{th}$ to 95$^{th}$; $\binom{P95}{P05}$) including both random and u∗ filtering uncertainty following the method described in Papale et al. (2006). Some of the sites lack wintertime measurements and we filtered out data for months with less than 10% observations. We performed a point-to-grid cell comparison to assess the degree of agreement between each flux magnitude and seasonality calculating the statistics of linear fit (slope, intercept, $R^2$) per flux and site between CARDAMOM and FLUXNET2015 datasets and determined RMSE and bias from model-data residuals comparison.

## 2.4 Benchmark of Global Vegetation Models from ISI-MIP2a

We compared CARDAMOM analyses of pan-Arctic net primary production (NPP), vegetation biomass carbon stocks ($C_{veg}$) and vegetation transit times ($TT_{veg}$) against six participating GVMs in the ISI-MIP2a comparison project (Akihiko et al., 2017). In this study we have considered DLEM (Tian et al., 2015), LPJmL (Schaphoff et al., 2013; Sitch et al., 2003), LPJ-GUESS (Smith et al., 2014), ORCHIDEE (Guimberteau et al., 2018), VEGAS (Zeng et al., 2005), and VISIT (Ito and Inatomi, 2012). The specific properties and degree of complexity of each ISI-MIP2a model are summarized in Table S5. The comparisons have been performed under the same spatial resolution as the CARDAMOM spatial resolution (1° x 1°) for the 2000-2010 period. Also, the chosen GVMs from the ISI-MIP2a phase have their forcing based on ERA-Interim climate data, similar to the forcing used in CARDAMOM. We estimated the degree of agreement using the statistics of linear fit (slope,

intercept, $R^2$, RMSE, and bias) per variable and model between CARDAMOM and GVMs, but also their spatial variability including stipples where the GVM datasets are within the CARDAMOM's 90% confidence interval.

To understand the sources of errors in $TT_{veg}$ calculations, we used CARDAMOM to calculate two hypothetical $TT_{veg}$ (i.e. EXPERIMENT A $TT_{veg}$ = ISI-MIP2a $C_{veg}$ / CARDAMOM NPP and EXPERIMENT B $TT_{veg}$ = CARDAMOM $C_{veg}$ / ISI-MIP2a NPP) and then assessed the largest difference with CARDAMOM's CONTROL $TT_{veg}$. We estimated the hypothetical $TT_{veg}$ for each pixel in each model, and derived a pixel-wise measure of the contribution of biases in NPP and $C_{veg}$ to biases in $TT_{veg}$ by overlapping their distribution functions.

**3 Results**

**3.1 Pan-Arctic retrievals of C cycle**

      Overall, we found that the pan-Arctic region (Figure 1 and Table 1) acted as a small sink of C (area-weighted P50) over the 2000-2015 period with an average of -67.4 $\binom{1159.9}{-286.7}$ g C m$^{-2}$ yr$^{-1}$, P50 $\binom{P95}{P05}$, although the 90% confidence intervals remain large (and so the region could be a source of C). Tundra regions NEE was estimated at -14.9 $\binom{1116.1}{-163.4}$ g C m$^{-2}$ yr$^{-1}$, a

245 weaker sink compared to taiga regions, -110.4 $\binom{1195.8}{-387.7}$ g C m$^{-2}$ yr$^{-1}$. The photosynthetic inputs exceeded the respiratory outputs (GPP > $R_{eco}$; Table 1), although the much larger uncertainties stemming from $R_{eco}$, and more specifically from $R_h$, compared with GPP, complicate the net C sink/source estimate beyond the median's average ensembles. In the pan-Arctic region approximately half of GPP is autotrophically respired resulting in an NPP of 290.3 $\binom{410.7}{196.4}$ g C m$^{-2}$ yr$^{-1}$. Carbon use efficiency (NPP/GPP) averages $0.51\binom{0.55}{0.46}$, and marginally varied across tundra $0.51\binom{0.54}{0.46}$ and taiga $0.52\binom{0.56}{0.46}$. Despite these apparent

small variations, tundra photosynthesized and respired (respectively $327.2\binom{463.3}{236.8}$ and $310.0\binom{1536.8}{124.3}$ g C m$^{-2}$ yr$^{-1}$) approximately half as much as the Taiga region ($759.8\binom{967.9}{584.1}$ and $635.3\binom{2114.0}{285.3}$ g C m$^{-2}$ yr$^{-1}$).

      The total size of the pan-Arctic soil C stock ($C_{dom}$) averaged $24.4\binom{47.5}{10.3}$ kg C m$^{-2}$, 16-fold greater than the vegetation C stock ($C_{veg}$), $1.5\binom{5.8}{0.5}$ kg C m$^{-2}$. The soil C stock (fresh litter and SOM) is dominated by $C_{som}$, accounting for the 99%, which also dominates the total terrestrial C stock in the pan-Arctic. Among the living C stocks, 93% of the C (88% in tundra and

255 90% in taiga) is allocated to the structural stocks (wood and roots; $1.4\binom{5.6}{0.4}$ kg C m$^{-2}$) compared to 7% (12% in tundra and 10% in taiga) to the photosynthetic stock (leaves and labile; $0.1\binom{0.2}{0.1}$ kg C m$^{-2}$). On average, the total ecosystem C stock is $26.3\binom{51.0}{11.8}$ kg C m$^{-2}$ in the pan-Arctic region, with slightly lower stocks in tundra ($24.6\binom{50.6}{10.8}$ kg C m$^{-2}$) than taiga ($27.7\binom{51.2}{12.7}$ kg C m$^{-2}$). In general, the taiga region holds on average ~100 % more photosynthetic tissues, ~160 % more structural tissue and ~9 % more soil C stocks, than tundra. In other words, taiga holds ~12 % more total C than tundra. The greater living stock of C in

taiga ($2.1\binom{5.1}{0.8}$ kg C m$^{-2}$) than tundra ($0.8\binom{6.8}{0.3}$ kg C m$^{-2}$) means that the relative size of $R_a$ and $R_h$ in the two regions differs. Thus in tundra $R_a$ accounts for 51% of total ecosystem respiration, while in taiga this fraction is 57%. $R_a$ is 4% larger than $R_h$ in tundra, but 24% greater in taiga, reflecting the greater rates of C cycling in taiga. Uncertainties in estimates of soil C stock are notably higher than for living C stocks, highlighting the lack of observational and mechanistic constraint on heterotrophic respiration.

The global mean C transit time is $1.3\binom{2.1}{0.8}$ years in leaves and labile plant tissue ($TT_{photo}$), $4.5\binom{15.7}{1.7}$ years in stems and roots ($TT_{veg}$), and $120.5\binom{822.6}{9.8}$ years in litter and SOM ($TT_{dom}$). The total C transit time ($TT_{tot}$) ($133.1\binom{1013.6}{11.5}$ years) is clearly dominated by the soil C stock, highlighting the very long periods of times that C persists in Arctic soils. CARDAMOM calculated 62% longer $TT_{dom}$ in tundra compared to taiga, likely linked to lower temperatures, but uncertainties are large due to the limitations of data constraints.

## 3.2 Data assimilation and uncertainty reduction

The CARDAMOM framework generated an analysis broadly consistent with the combination of SOC, biomass and LAI in each grid cell (Figure 2), and the errors assigned to these data products. The agreement for the SOC dataset by Hugelius et al. (2013a) is a 1:1 relationship ($R^2 = 1.0$; RMSE = 0.95 kg C m$^{-2}$), reflecting a straightforward model parameterisation. The biomass product from Carvalhais et al. (2014), was well correlated ($R^2 = 0.97$; RMSE = 0.46 kg C m$^{-2}$), but CARDAMOM was consistently biased ~28% low. MODIS LAI data were also well correlated ($R^2 = 0.79$; RMSE = 0.42 kg C m$^{-2}$), but ~28% higher than CARDAMOM analyses. These biases (Figure 2) likely arise due to a low estimate in the photosynthesis model (ACM) used in CARDAMOM which propagates through the C cycle. CARDAMOM balances uncertainty in data products and the models (ACM photosynthesis model and DALEC2), to generate a weighted analysis, typical of Bayesian approaches. The CARDAMOM analysis 90% confidence interval (CI) includes the 1:1 line for biomass and LAI (Figure 2), indicating that the likelihoods on C cycle analyses include the expected value of the observations.

The degree to which posterior distributions were constrained from the prior distributions in each of the 17 model parameters and 6 initial stock sizes (Table S2) varied considerably depending on the parameters in question and their related processes (Table 2 and Figure S2). The 90% CI posterior range of foliar, wood, labile and SOM C stocks ($C_{foliar}$, $C_{wood}$, $C_{labile}$ and $C_{som}$) as well as parameters such as allocation to foliage ($f_{fol}$) and lifespan (L) were considerably reduced (>80% uncertainty reduction compared to priors) most likely controlled by the information on LAI, biomass and SOC constraints. Contrarily, parameters that have not been regulated in any way in the MHMCMC algorithm, i.e. turnover processes such as litter mineralization ($MR_{litter}$), roots turnover ($TOR_{roots}$), wood turnover ($TOR_{wood}$), decomposition rates ($D_{rate}$) and initial C stock such as litter ($C_{litter}$) were found poorly constrained (<20% uncertainty reduction). Overall, the uncertainty reduction classified by processes and ranked from most to least constrained estimated a 71% reduction for C stocks, 67% reduction for C allocation, 59% for plant phenology and 31% for C turnover related parameters. Although there are not substantial differences between tundra and taiga, $C_{roots}$ was better constrained in tundra regions (42%), while leaf onset day ($B_{day}$), leaf fall day ($F_{day}$), and leaf fall duration ($L_f$) were better constrained in taiga regions (>18% or more).

## 3.3 Independent evaluation: from global to local scale

We compared our estimates of GPP and $R_h$ with independent datasets to evaluate the model performance (Figure 3). We found GPP to be well correlated ($R^2 = 0.81$; RMSE = 0.43 kg C m$^{-2}$), but biased lower (~53%) compared to Jung et al. (2017)'s GPP estimates. There are in general very few pixels where FLUXCOM product falls within CARDAMOM's 90% confidence interval. Additionally, the $R_h$ product from Hashimoto et al. (2015) is less consistent with our estimates ($R^2 = 0.40$; RMSE = 0.09 kg C m$^{-2}$), presenting a tendency towards lower values in tundra pixels and higher values in taiga pixels. The spatial variability of $R_h$ is considerably smaller in Hashimoto et al. (2015) compared to our CARDAMOM estimates. $R_h$ falls within the 90% confidence interval of CARDAMOM in most of the pan-Arctic region due to the fact that the $R_h$ uncertainties are significant (Figure 3). This finding confirms the uncertainties previously noted in modelled respiratory processes (Table 1) where the upper P95 in $R_h$ dominated NEE's uncertainties, but also the soil C stocks and transit times.

For comparison with direct ground observations from the FLUXNET2015 dataset, we report here monthly aggregated P50 ± P05-95 estimates of NEE, GPP and $R_{eco}$ to show timing and magnitudes, but also to diagnose whether CARDAMOM is in general agreement with flux tower data. Overall, CARDAMOM performed well in simulating observed NEE ($R^2 = 0.66$; RMSE = 0.51 g C m$^{-2}$ month$^{-1}$; Bias = 0.16 g C m$^{-2}$ month$^{-1}$), GPP ($R^2 = 0.85$; RMSE = 0.89 g C m$^{-2}$ month$^{-1}$; Bias = 0.5 g C m$^{-2}$ month$^{-1}$) and $R_{eco}$ ($R^2 = 0.82$; RMSE = 0.63 g C m$^{-2}$ month$^{-1}$; Bias = 0.35 g C m$^{-2}$ month$^{-1}$) across 8 sub-Arctic and high-Arctic sites from the FLUXNET2015 dataset (Figure 4; Table S6). CARDAMOM NEE is ~25% lower than FLUXNET2015, while GPP and $R_{eco}$ are ~30% and ~10% higher, respectively. This mismatch is important in the context of the FLUXCOM GPP upscaling, 50% higher than CARDAMOM GPP. At some sites such as Hakasia, Samoylov, Poker Flat and Manitoba (NEE $R^2 = 0.73$; GPP $R^2 = 0.92$ and $R_{eco}$ $R^2 = 0.88$) CARDAMOM better matches the seasonality and the magnitude of the C

fluxes than the rest, i.e. Tiksi, Kobbefjord, Zackenberg and UCI-1998 (NEE $R^2 = 0.58$; GPP $R^2 = 0.67$ and $R_{eco}$ $R^2=0.67$). In general, CARDAMOM captured the beginning and the end of the growing season well (Figure 4), although the assimilation system has some bias due to (1) difference in timing (e.g. earlier shifts of peak of the growing season in Manitoba GPP and $R_{eco}$ and  earlier end of the growing season in Poker Flat NEE) and (2) differences in flux magnitudes (such as in Hakasia GPP and $R_{eco}$ and Kobbefjord NEE).

## 3.4 Benchmarking ISI-MIP2a models with CARDAMOM

We used our highest confidence retrievals of NPP, $C_{veg}$ and $TT_{veg}$ (i.e. retrievals including assimilated LAI, biomass and SOC) to benchmark the performance of the GVMs from the ISI-MIP2a project. In this assessment we compared not only their spatial variability across the pan-Arctic, tundra and taiga region (Figure 5), but also the degree of agreement between their mean model ensemble within the 90% confidence interval of our assimilation framework (Figure 6, Table 3). NPP estimates (RMSE = 0.1 kg C $m^{-2}$ $yr^{-1}$; $R^2= 0.44$) are in better agreement than $C_{veg}$ (RMSE = 1.8 kg C $m^{-2}$; $R^2= 0.22$) and $TT_{veg}$ (RMSE = 4.1 years; $R^2= 0.12$). The assessed GVMs estimated on average 8% lower NPP, 16% higher $C_{veg}$ and 22% longer $TT_{veg}$ than CARDAMOM across the entire pan-Arctic domain (Figure 5 and 6) on average. Thus, at regional aggregation CARDAMOM analyses agreed more closely with ISI-MIP2a models than with FLUXCOM (51% difference) and with the Carvalhais et al. (2014) biomass data (28% bias).

The poor spatial agreement regarding $TT_{veg}$ between CARDAMOM and ISI-MIP2a (Table 3) is indicative of uncertainties in the internal C dynamics of these models. For instance, the slopes in Table 3 are steep and the $R^2$ are poor – so there is a substantial disagreement in the spatial pattern, not just a large bias. For ISI-MIP2a comparison $R^2$ values ranged from 0.03-0.52 for NPP; 0.00-0.31 for $C_{veg}$; and 0.00-0.24 for $TT_{veg}$. Spatially, the stippling in Figure 6 indicates areas where the GVMs are within the 90% CI of CARDAMOM; agreement is best over the taiga domain rather than in tundra for $TT_{veg}$. The benchmark area of consistency (stippling) is more extensive for $C_{veg}$ and $TT_{veg}$ than for NPP. Thus, while there is a stronger spatial correlation for NPP between CARDAMOM and GVMs (Table 3), this is a clearer bias for NPP. Some models (LPJ-GUESS and ORCHIDEE) systematically calculate lower values in all the assessed variables while others (LPJmL and VISIT) calculate higher estimates. The models in closer agreement with CARDAMOM were DLEM (5% difference) and LPJ-GUESS (17%) while VEGAS (44%) and ORCHIDEE (56%) were the models with larger discrepancies (Table 3; Figure 5 and 6).

The attribution analysis to identify the origin of bias from ISI-MIP2a models indicated a joint split between NPP and $C_{veg}$ for $TT_{veg}$ error simulated in GVMs (Figure 7). The distribution of the differences relative to CARDAMOM revealed that the higher error (i.e. the lower overlapped area, and by extension the largest contributor to $TT_{veg}$ biases) comes from ISI-MIP2a NPP with a 69% agreement in the distribution, while $C_{veg}$ agrees 72%. In fact, the $TT_{veg}$ $R^2$ for each model (Table 3) is very close to the product of the NPP $R^2$ and $C_{veg}$ $R^2$ for that model, i.e. the uncertainty on the $TT_{veg}$ is a direct interaction of NPP and $C_{veg}$ uncertainty ($R^2$ of the correlation = 0.71). This finding supports Figure 6, which shows $TT_{veg}$ error derives equally from both NPP and $C_{veg}$.

## 4 Discussion

### 4.1. Pan-Arctic retrievals of C cycle

The CARDAMOM framework has been used to evaluate the terrestrial pan-Arctic C cycle in tundra and taiga at coarse spatio-temporal scale (at monthly and annual time steps for the 2000-2015 period and at 1° x 1° grid cells). Overall, we found that the pan-Arctic region was most likely a consistent sink of C (weaker in tundra and stronger in taiga), although the large uncertainties derived from respiratory processes (Table 1) strongly increase the 90% confidence interval uncertainty. We estimate that tundra experienced 62% longer transit times in litter and SOM C stocks than taiga ecosystems. Further, the

contribution of $R_a$ and $R_h$ to total ecosystem respiration was similar in tundra (51%, 49%) but dominated by $R_a$ in taiga (57% compared to 43% in tundra).

CARDAMOM retrievals are consistent with outcomes from relevant papers such as the (I) C flux observations and model estimates reported in McGuire et al. (2012); (II) C stocks and transit times described by Carvalhais et al. (2014), and (III) NPP, C stocks and turnover rates stated in Thurner et al. (2017):

I.  The CARDAMOM NEE estimates reported in this study for the tundra domain are inside the variability comparison of values compiled by McGuire et al. (2012) considering field observation, regional process-based models, global-process based models and inversion models. The authors reported that Arctic tundra was a sink of $CO_2$ of -150 Tg C $yr^{-1}$ (SD=45.9) across the 2000-2006 period over an area of 9.16 x $10^6$ $km^2$. Here, CARDAMOM NEE estimated -129 Tg C $yr^{-1}$ over an area of 8.1 x $10^6$ $km^2$ for the same period. This exhaustive assessment of the C balance in Arctic tundra included approximately 250 estimates using the chamber and eddy covariance method from 120 published papers (McGuire et al., 2012; Supplement 1) with an area-weighted mean of means of -202 Tg C $yr^{-1}$. The regional models, including runs from LPJ-Guess WHyMe (Wania et al., 2009a, b), Orchidee (Koven et al., 2011), TEM6 (McGuire et al., 2010), and TCF model (Kimball et al., 2009), reported a NEE of -187 Tg C $yr^{-1}$ and GPP, NPP, $R_a$ and $R_h$ of 350, 199, 151 and 182 g C $m^{-2}y^{-1}$, respectively. GVMs applications such as CLM4C (Lawrence et al., 2011), CLM4CN (Thornton et al., 2009), Hyland (Levy et al., 2004), LPJ (Sitch et al., 2003), LPJ- Guess (Smith et al., 2001), O-CN (Zaehle and Friend, 2010), SDGVM (Woodward et al., 1995), and TRIFFID (Cox, 2001) estimated a NEE of -93 Tg C $yr^{-1}$ and GPP, NPP, $R_a$ and $R_h$ of 272, 162, 83 and 144 g C $m^{-2}yr^{-1}$. For the same period, CARDAMOM has estimated 330, 167, 160 and 154 g C $m^{-2}$ $yr^{-1}$ respectively for the same gross C fluxes.

II.  Carvalhais et al. (2014) estimated a total ecosystem carbon ($C_{tot}$) of $20.5\binom{52.5}{8.0}$ kg C $m^{-2}$ for tundra and $24.8\binom{58.0}{15.2}$ kg C $m^{-2}$ for taiga, while values from CARDAMOM were $24.6\binom{50.6}{10.8}$ kg C $m^{-2}$ for tundra, and $27.7\binom{51.2}{12.7}$ kg C $m^{-2}$ in taiga (Figure 5; Table 1) for the same area.Thus, Carvalhais et al. (2014)'s $C_{tot}$ product stored 20% and 12% less carbon in tundra and taiga respectively than CARDAMOM. Overall, CARDAMOM calculated 20% and 6% longer transit times for tundra and taiga respectively, with average values of $80.8\binom{195.2}{21.8}$ years in tundra and $51.2\binom{109.3}{22.1}$ years in taiga (Table 1) compared to the $64.4\binom{259.8}{25.7}$ years in tundra and $48.2\binom{111.6}{24.9}$ years in taiga in Carvalhais et al. (2014). These numbers have been retrieved from the same biome classification and they include the 90% confidence interval of the assessed spatial variability. Also, we applied a correction factor of $TT_{gpp} = TT_{npp}*(1-fraction\ of\ GPP\ respired)$ to be comparable with Carvalhais et al. (2014) TT. Both datasets agree on the fact that high (cold) latitudes, first tundra, and second taiga have the longest transit times in the entire globe (Bloom et al., 2016; Carvalhais et al., 2014).

III.  A recent study from Thurner et al. (2017) assessed temperate and taiga-related TTs presenting a 5-year average NPP dataset applying both MODIS (Running et al., 2004; Zhao et al., 2005) and BETHY/DLR (Tum et al., 2016) products and an inovative biomass product (Thurner et al., 2014) accounting for both forest and non-forest vegetation. Our estimate of $TT_{veg}$ for the exact same period is $5.3\binom{18.2}{1.9}$ years, compared to Thurner et al. (2017)'s TT, $8.2\binom{11.5}{5.5}$ years using MODIS and $6.5\binom{8.7}{4.2}$ years using BETHY/DLR. A note of caution here, the number reported by the authors are turnover rates, which are inferred to transit times by just applying the inverse of turnover rates ($TT_{veg}$=1/turnover rates). Additionally, their NPP estimates, 0.35 and 0.45 kg C $m^{-2}$ $yr^{-1}$ from both MODIS and BETHY/DLR, is only 5% more productive as average than CARDAMOM NPP estimate, $0.4\binom{0.6}{0.3}$ kg C $m^{-2}$ $yr^{-1}$; and the biomass derived from Thurner et al. (2014), 3.0 $\pm$1.1 kg C $m^{-2}$, is ~30% lower than CARDAMOM $C_{veg}$, $2.2\binom{5.0}{1.1}$kg C $m^{-2}$, calculated for the same period and for the same taiga domain.

In general, we found a reasonable agreement between CARDAMOM and assimilated and independent data at pan-Arctic scale. CARDAMOM retrievals of assimilated data are in good agreement with the SOC (Figure 2). The simulation of $TT_{dom}$ is weakly constrained (Table 1) - our analysis adjusts TT to match mapped stocks, hence the strong match of modelled to mapped SOC. So, independent data on $TT_{dom}$ data (e.g. $^{14}C$) is required across the pan-Arctic region to provide stronger constraint on process parameters and reduce the very broad confidence intervals of CARDAMOM analyses. The low bias in mean estimates of LAI and biomass (Figure 2) likely relates to the strong prior on photosynthesis estimates from the ACM model, which lacks a temperature acclimation for high latitudes in this implementation. However, the uncertainty in the biomass and LAI analyses spans the magnitude of the bias. So, CARDAMOM generates some parameters sets that are consistent with observations. CARDAMOM produces analyses that reproduce the pattern of LAI, GPP, biomass and SOC (Figure 2 and 3) – this demonstrates that the DALEC model structure can be calibrated to simulate the links between these variables as a function of mass balance constraints, and realistic process interactions and climate sensitivities.

There are clear biases in CARDAMOM analyses compared to independent global upscaled GPP (Jung et al., 2017) and $R_h$ products (Hashimoto et al., 2015) (Figure 3). However, CARDAMOM resolves the spatial pattern in GPP effectively, while the spatial mismatch in $R_h$ estimates is marked (Figure 3), echoing the large uncertainty found in NEE (Figure 1, Table 1). One difference with Hashimoto et al. (2015)'s $R_h$ model is the lack of moisture limitation on respiration in CARDAMOM. Conversely, GPP is relatively well-constrained in space through the assimilation of LAI and a prior for productivity (Bloom et al., 2016), although an important mismatch has been found: CARDAMOM GPP is 50% lower than FLUXCOM, but 30% higher than FLUXNET2015 EC data.

The agreement between CARDAMOM analyses and EC data is high given the scale difference. A direct point-to-grid cell comparison with local observations derived from the FLUXNET2015 dataset (Figure 4, Table S6) is challenging and always difficult. CARDAMOM outputs covers 1° x 1° grid cells, whereas local eddy covariance flux measurements are in the order of 1-10 ha. Thus, for observational sites located in areas with complex terrain, such as Kobbefjord in coastal Greenland, the agreement can be expected to be low. For inland forest sites, such as Poker Flat in Alaska, there may be less differences in vegetation characteristics and local climatology between the local scale measurement footprint and the corresponding CARDAMOM grid cell. This scaling issue is likely to have a larger impact on flux magnitudes compared with seasonal dynamics. In general, CARDAMOM captured the seasonal dynamics in NEE, GPP and $R_{eco}$ well (Figure 4, Table S6), although the monthly model time-step does reduce skill in shoulder seasons. There was a consistent timing-mismatch in early season flux increase, where CARDAMOM predicts earlier growing season onset compared with observations. This is likely due to the impact of snow cover, which is not explicitly included in the CARDAMOM framework.

For a further independent evaluation of CARDAMOM outputs, we compare the tundra and boreal estimates to plot scale flux and stock information. For tundra, Street et al. (2012) calculate growing season GPP estimates of 263-380 g C m$^{-2}$ for *Empetrum nigrum* communities, and 295-386 g C m$^{-2}$ for *Betula nana* communities, which is consistent with the ranges in Figure 1 for tundra. Biomass stocks for Arctic tundra recorded in the Arctic LTER at Toolik Lake range from 105-1160 g C m$^{-2}$ (Hobbie and Kling, 2014), which are consistent with the estimates from CARDAMOM, albeit at the lower end of the model estimates. For boreal forests, Goulden et al. (2011) report annual GPP estimates across a chronosequence of stands, and thus a variation across canopy densities, which varied from 450-720 g C m$^{-2}$ yr$^{-1}$. These data are consistent with the span of GPP in CARDAMOM (Figure 1), again best matching the lower end of the model estimates. For the same study, the vegetation C stock estimates varied from 100-5000 g C m$^{-2}$, consistent with CARDAMOM, and with measurements of 10 to 40-year old boreal stands best matching the CARDAMOM median estimate of ~1500 g C m$^{-2}$. We conclude from comparisons against site data that CARDAMOM analyses are broadly consistent, with some tendency for CARDAMOM to have a high bias. This comparison is similar to the FLUXNET2015 evaluation of CARDAMOM. But it conflicts with the estimation of low bias from the comparison of CARDAMOM against FLUXCOM GPP and Carvalhais et al. (2014) biomass stock maps. It is possible

that the scale differences between site level products and landscape estimates is confusing these comparisons, but there is clearly a need to understand better these inconsistencies in C cycle estimates.

## 4.2. CARDAMOM as a model benchmarking tool

An ideal benchmarking tool for GVMs would compare model state variables and fluxes against multiple, independent, unbiased, error-characterised measurements collected repeatedly at the same temporal and spatial resolution. Of course direct measurements of key C cycle variables like these are not available. Even at FLUXNET sites GPP and $R_{eco}$ must be inferred, and NEE data often gap-filled. Satellite data can provide continuous fields, but do not directly measure ecological variables like biomass or LAI, so calibrated models are required to generate ecological products. Atmospheric conditions can introduce biases and data gaps into optical data that are poorly quantified. Upscaling of FLUXNET data requires other spatial data, e.g. MODIS LAI, which challenges the characterisation of error and generates complex hybrid products. We suggest that CARDAMOM provides some of the requirements of the ideal benchmark system – an error-characterised, complete analysis of the C cycle that is based on a range of observational products. CARDAMOM includes its own C cycle model; this has the advantage of evaluating the observational data for consistency (e.g. with mass balance), propagating error across the C cycle, and generating internal model variables such as TT. Further the model is of intermediate complexity and independent of the benchmarked models.

Using CARDAMOM as a benchmarking tool for six GVMs we found disagreements that varied among models for spatial estimates of NPP, $C_{veg}$ and $TT_{veg}$ across the Pan-Arctic (Figure 6) in comparison against CARDAMOM confidence intervals. GVM NPP estimates had a higher correlation than $TT_{veg}$ and $C_{veg}$ with CARDAMOM analyses (Table 3), but because CARDAMOM confidence intervals on NPP were relatively narrow (Figure 1) the benchmarking scores from GVM NPP were relatively poor (Figure 6). Consequently, we used CARDAMOM to calculate the relative contribution of productivity and biomass to the transit times bias by applying a simple attribution analysis (Figure 7). We concluded that the largest bias to transit times originated not by a deficient understanding of one single component, but by an equal combination of both productivity and biomass errors together. Therefore, this study partially agrees with previous studies (Friend et al., 2014; Nishina et al., 2014; Thurner et al., 2017) highlighting the deficient representation of transit times/turnover dynamics, but we further suggest that GVM and ESM modellers need to focus on the productivity and vegetation C stocks dynamics to improve inner C dynamics. A major challenge for GVMs is the spin-up problem (Exbrayat et al., 2014). GVMs need to find a way to ensure that the spin-up process produces biomass estimates consistent with the growing availability of biomass maps from earth observations. CARDAMOM solves this problem by avoiding spin-up. Its fast run time allows the biomass maps to act as a constraint on the probability distribution of model parameters. There may be opportunities to use CARDAMOM style approaches to assist the GVM community address this problem.

## 4.3 Outlook

Although CARDAMOM estimates for pan-Arctic C cycling are in moderately good agreement with observations and data constraints, we have not included important components controlling ecosystem processes that could potentially improve our understanding on C feedbacks, and with emphasis for high latitude ecosystems. For example, thaw and release of permafrost C is not represented in CARDAMOM, but the influence on vegetation dynamics, permafrost degradation and soil respiration is critical in high latitudes (Koven et al., 2015; Parazoo et al., 2018). Also, Koven et al. (2017) shown that soil thermal regimes are key to resolving the long-term vulnerability of soil C. Moreover, we have not characterized snow dynamics nor the insulating effect of snow affecting respiratory losses across wintertime periods (López-Blanco et al., 2018). Further, methane emissions, another important contributor to total C budget (Mastepanov et al., 2008; Zona et al., 2016), was neglected from this modelling exercise since it is not easy to model due to its three complex transport mechanisms (Walter et al., 2001).

However, our approach to use an intermediate complexity model has the strong advantage of allowing very large ($10^7$) model ensembles per pixel, and thus a thorough exploration of model-parameter interactions, that is not feasible with typical GVMs. Other viable options include using emulators (Fer et al., 2018) and particle filters (Arulampalam et al., 2002), but MCMC methods provide the most detailed description of error distributions. There remains a strong argument to utilize intermediate complexity models like DALEC2 to evaluate the minimum level of detail required to represent ecosystem processes consistent with local observations, and to allow testing of alternate model structures. And, assimilating further data products, for instance patterns in soil hydrology and snow states across the pan-Arctic from earth observation, could provide useful information on spatio-temporal controls on soil activity and microbial metabolism to constrain below ground processes. This information would need to be tied to process level information on SOM turnover generated from experimental studies, and included in updated versions of DALEC. Thus, more field observations are crucial across the pan-Arctic, specifically on decomposition and TT of SOC (He et al., 2016) and respiratory processes such as partitioning of $R_{eco}$ into $R_a$ and $R_h$ (Hobbie et al., 2000; McGuire et al., 2000), across the growing season and also during wintertime (Commane et al., 2017; Zona et al., 2016).

Our approach has used estimated observation error, and inflated this to include unknown errors associated with model process representation. We currently lack any better knowledge of the combined uncertainties arising from model representation errors and observation errors. We acknowledge that all models are an imperfect representation of C dynamics, which generates irreconcilable model-data errors due to the inherent assumptions in model structure. Future analyses should investigate model structural error, using for example error-explicit Bayesian approaches (Xu et al., 2017), or comparing the likelihoods of alternate model structures, of varying complexity. Using multiple sources of data, we have highlighted systematic errors in the model at landscape scale (Figure 2 and 3) for LAI, GPP and biomass. However, these biases are not consistent for site-scale evaluations. Thus, a next step would be to include explicitly both random and systematic process errors for C fluxes in the data assimilation. These errors could be determined from field scale evaluation of model process representation (Table 2) using e.g. FLUXNET2015 data. We also need to understand better the error associated with landscape heterogeneity of C stocks and fluxes, to upscale from flux tower observations, or direct measurements of LAI, to landscape pixel. This could be achieved by constructing robust observation error models (Dietze, 2017) from field to pixel scale, for e.g. GPP, LAI and foliar N. Evaluation of the sensitivity of C cycling DA analyses to observation error has shown relatively low sensitivity to data gaps and random error on net ecosystem flux data (Hill et al., 2012), but further analyses of error sensitivity are required for multiple streams of stock data.

**5 Conclusions**

The Arctic is experiencing rapid environmental changes, which will influence the global C cycle. Using a data-assimilation framework we have evaluated the current state of key C flux, stocks and transit times for the pan-Arctic region, 2000-15. We found that the pan-Arctic was a likely sink of C, weaker in tundra and stronger in taiga, but uncertainties around the respiration losses are still large, and so the region could be a source of C. Comparisons with global and local scale datasets demonstrate the capabilities of CARDAMOM for analysing the C cycle in the Arctic domain. CARDAMOM is a data-constrained and data-integrated analysis, evaluated for internal consistency, and is therefore a good candidate to benchmark performance of global vegetation/ecosystem models. We conclude that a GVM bias found in transit time of vegetation C is the result of a joint combination of uncertainties from productivity processes and biomass in GVMs, and thus these are a major component of error in their forecasts. While spatial patterns in GVM predictions of NPP are reasonable, particularly in taiga, they have significant biases against the CARDAMOM benchmark. Improved mapping of vegetation and soil C stocks and change over time is required for better analytical constraint. Moreover, future work is required on assimilating data on soil hydrology, permafrost and snow dynamics to improve accuracy and decrease uncertainties on belowground processes. This

work establishes the baseline for further process-based ecological analyses using the CARDAMOM data-assimilation system as a technique to constrain the pan-Arctic C cycle.

**Data and software availability**

CARDAMOM output used in this study is available from Exbrayat and Williams (2018) from the University of Edinburgh's DataShare service at https://doi.org/10.7488/ds/2334. The DALEC2 code is also available on Edinburgh
DataShare at https://doi.org/10.7488/ds/2504. Contact MW for access to the CARDAMOM software.

**Acknowledgements**

This work was supported in part by a scholarship from the Aarhus-Edinburgh Excellence in European Doctoral Education Project and by the eSTICC (eScience tools for investigating Climate Change in Northern High Latitudes) project, part of the Nordic Center of Excellence. This work was also supported by the Natural Environment Research Council (NERC)
through the National Center for Earth Observation. Data-assimilation procedures were performed using the Edinburgh Compute and Data Facility resources. This work used eddy covariance data acquired and shared by the FLUXNET community, including these networks: AmeriFlux, AfriFlux, AsiaFlux, CarboAfrica, CarboEuropeIP, CarboItaly, CarboMont, ChinaFlux, Fluxnet-Canada, GreenGrass, ICOS, KoFlux, LBA, NECC, OzFlux-TERN, TCOS-Siberia, and USCCC. The ERA-Interim reanalysis data are provided by ECMWF and processed by LSCE. The FLUXNET eddy covariance data processing and
harmonization was carried out by the European Fluxes Database Cluster, AmeriFlux Management Project, and Fluxdata project of FLUXNET, with the support of CDIAC and ICOS Ecosystem Thematic Center, and the OzFlux, ChinaFlux and AsiaFlux offices. We thank Nuno Carvalhais for discussion that helped to focus our ideas. For their roles in producing, and making available the ISI-MIP model output, we acknowledge the modelling groups and the ISI-MIP coordination team.

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

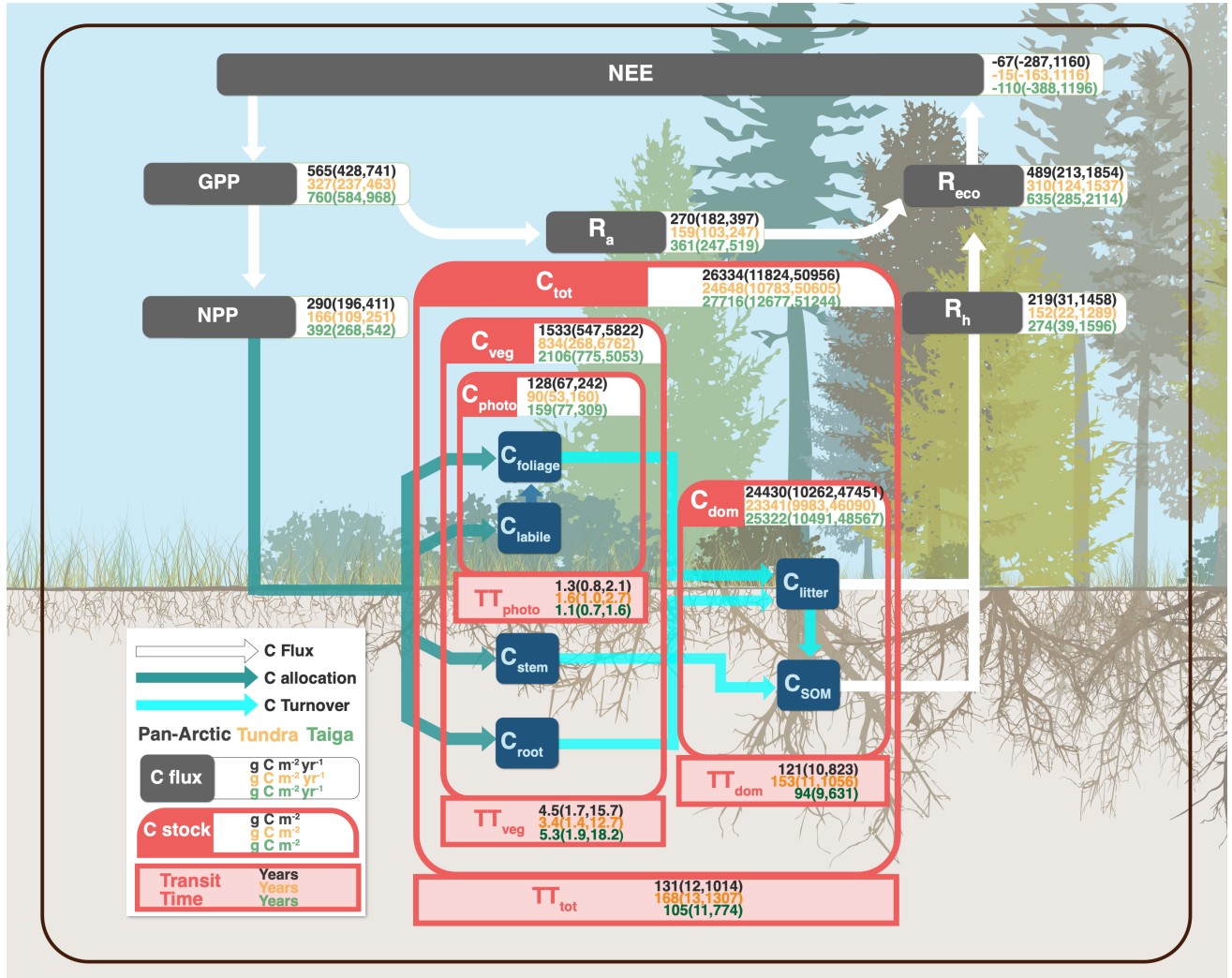

Figure 1. Schematic diagram of the terrestrial C processes modelled in CARDAMOM for the pan-Arctic (black values), tundra (yellow values) and taiga (green values) domains. The values characterize the median for the 2000-2015 period and the parentheses delimit the 90% confidence interval. C processes represented include flows for C fluxes in white [NEE, Net Ecosystem Exchange; GPP, Gross Primary Production; NPP, Net Primary Production; $R_{eco}$, ecosystem Respiration; $R_a$, autotrophic Respiration; $R_h$, heterotrophic Respiration], C allocation in blue [to labile, leaf, stem and root], and C turnover in cyan [from leaf, wood, roots and litter]. C stocks are represented in dark blue boxes [labile, leaf, stem, root, litter and SOM, Soil Organic Matter] and aggregated into photosynthetic ($C_{photo}$ = leaf + labile), vegetation ($C_{veg}$ = leaf + labile + wood + roots), soil ($C_{dom}$ = litter + SOM) and total ($C_{tot}$ = $C_{photo}$ + $C_{veg}$ + $C_{dom}$) C stocks in red boxes. Analogy, transit times (TT) are also aggregated into photosynthetic ($TT_{photo}$ = leaf + labile), vegetation ($TT_{veg}$ = leaf + labile + wood + roots), soil ($TT_{dom}$ = litter + SOM) and total ($TT_{tot}$ = $TT_{photo}$ + $TT_{veg}$ + $TT_{dom}$) C transit times.

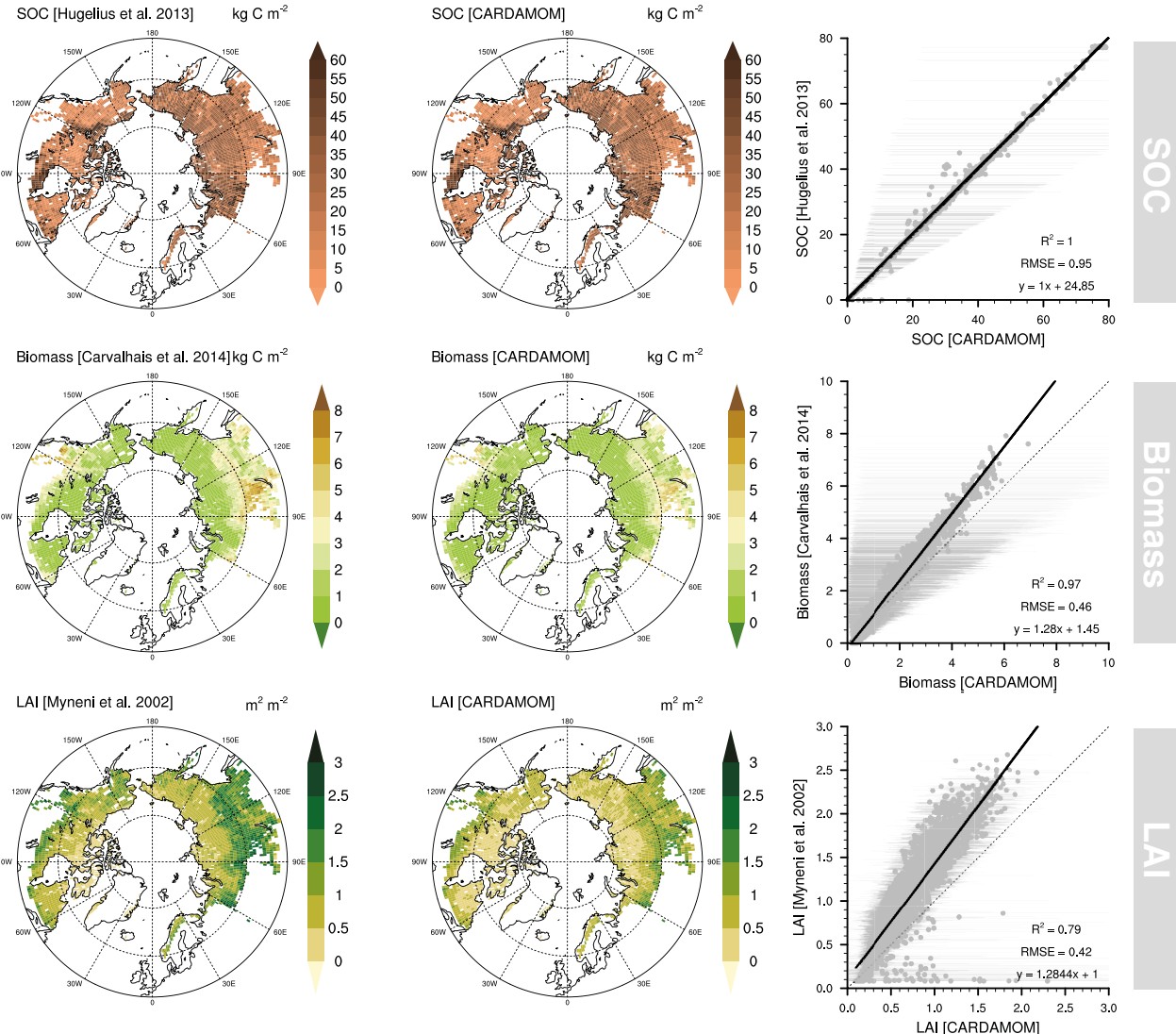

**Figure 2. Original soil organic carbon [SOC; Hugelius et al. 2013], biomass [Carvalhais et al. 2014] and leaf area index [LAI; Myneni**

**et al. 2002] datasets used in the data assimilation process within the CARDAMOM framework (left hand side), assimilated SOC,**

**biomas and LAI integrated in CARDAMOM (center), and their respective goodness-of-fit statistics between original and assimilated**

**datasets (right hand side). The error bars represent the 90% confidence interval of the assimilated variable in CARDAMOM.**

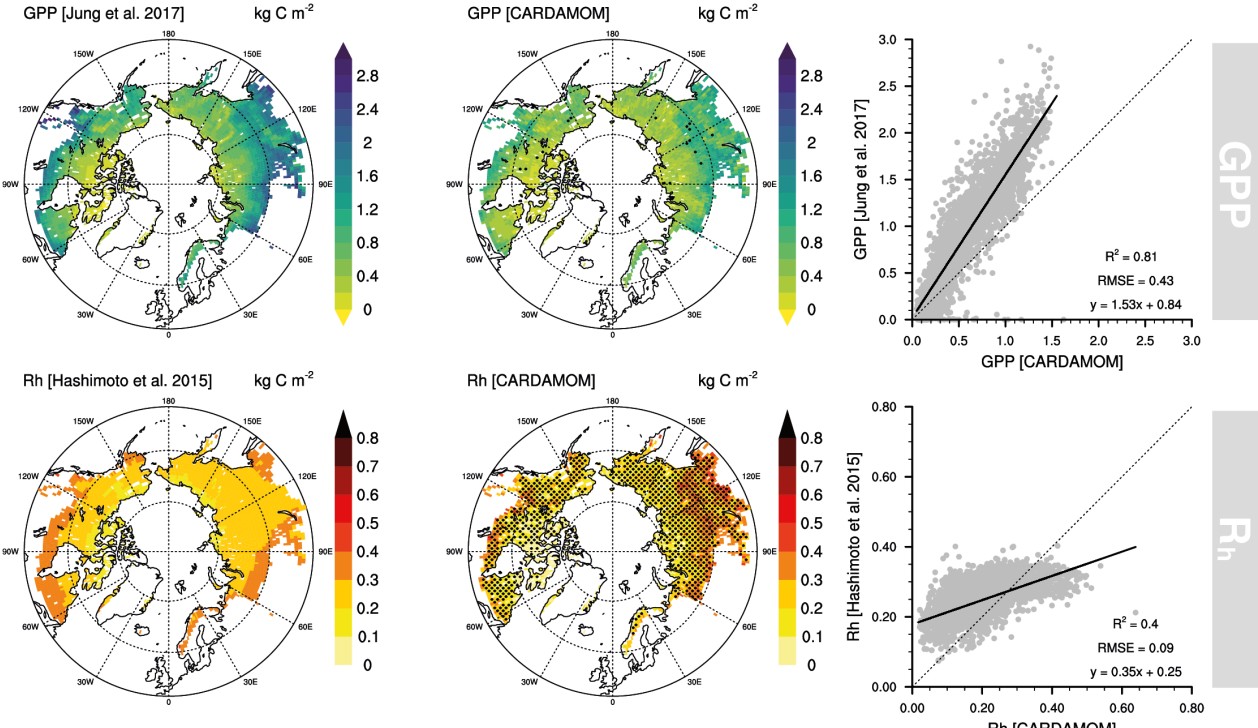

**Figure 3. Original gross primary productitvity [GPP; Jung et al., 2017] and heterotropic respiration [R_h; Hashimoto et al., 2015]**

**datasets used in the data validation process (left hand side), estimated GPP and R_h by CARDAMOM (center), and their respective**

**goodness-of-fit statistics between original and assimilated datasets (right hand side). Stippling indicates locations where the**

**independent datasets are within the CARDAMOM's 5th and 95th percentiles.**

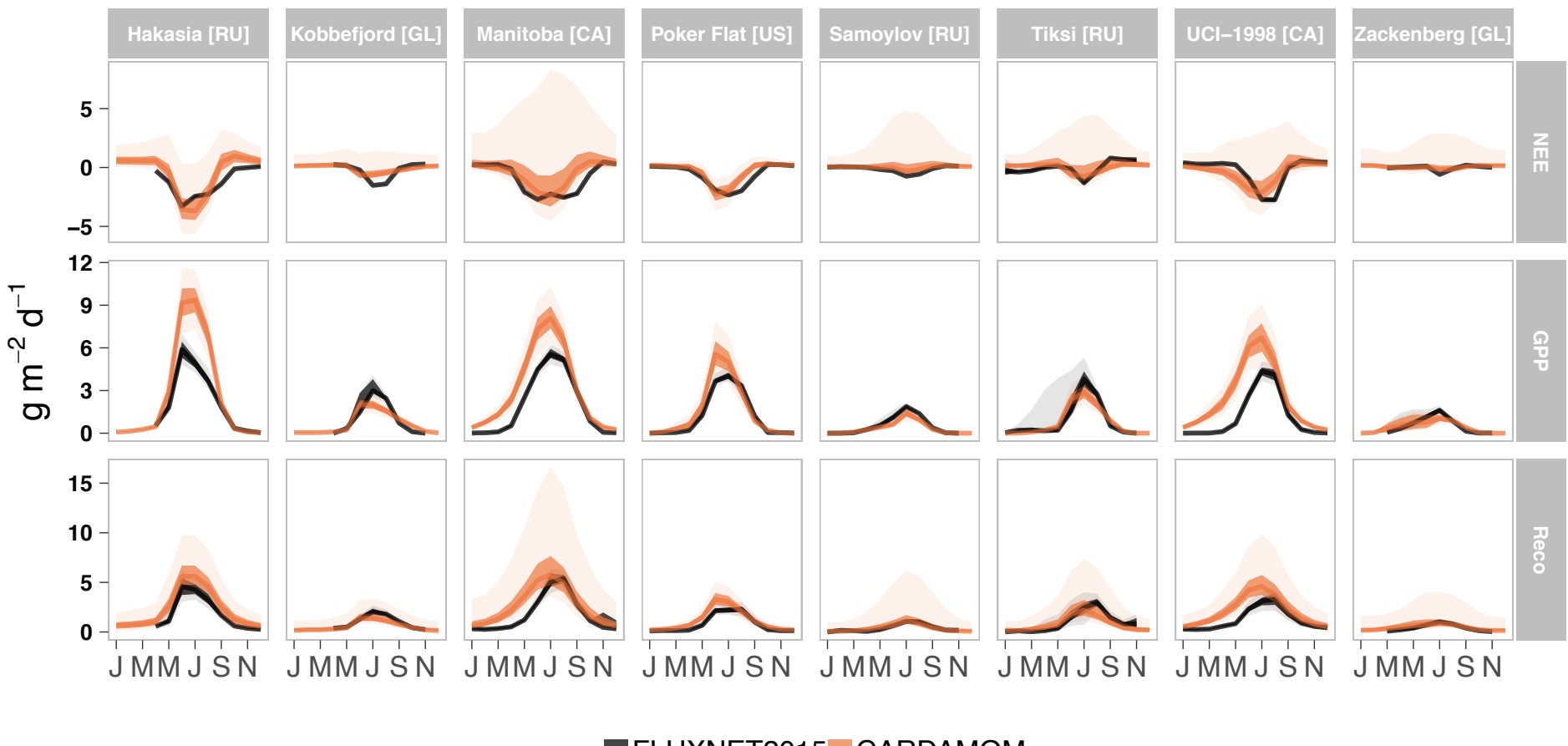

3  **Figure 4. Monthly-aggregated seasonal variability of observed [FLUXNET2015] and modelled [CARDAMOM] C fluxes [NEE, Net Ecosystem Exchange; GPP, Gross Primary Production; R$_{eco}$,**

4  **ecosystem Respiration] across eight low- and high-Arctic sites [Hakasia, Kobefjord, Manitoba, Poker Flat, Samoylov, Tiksi, UCI-1998 and Zackenberg]. Each of these sites, located in different**

5  **countries [RU-Russia, GL-Greenland, CA-Canada, US-Unite States,] feature different meteorological conditions and vegetation types (Table S4). Uncertainties represent the 25th and 50th**

6  **percentiles (darker shade) and the 5th and 95th percentiles (lighter shade) of both field observations and the CARDAMOM framework.**

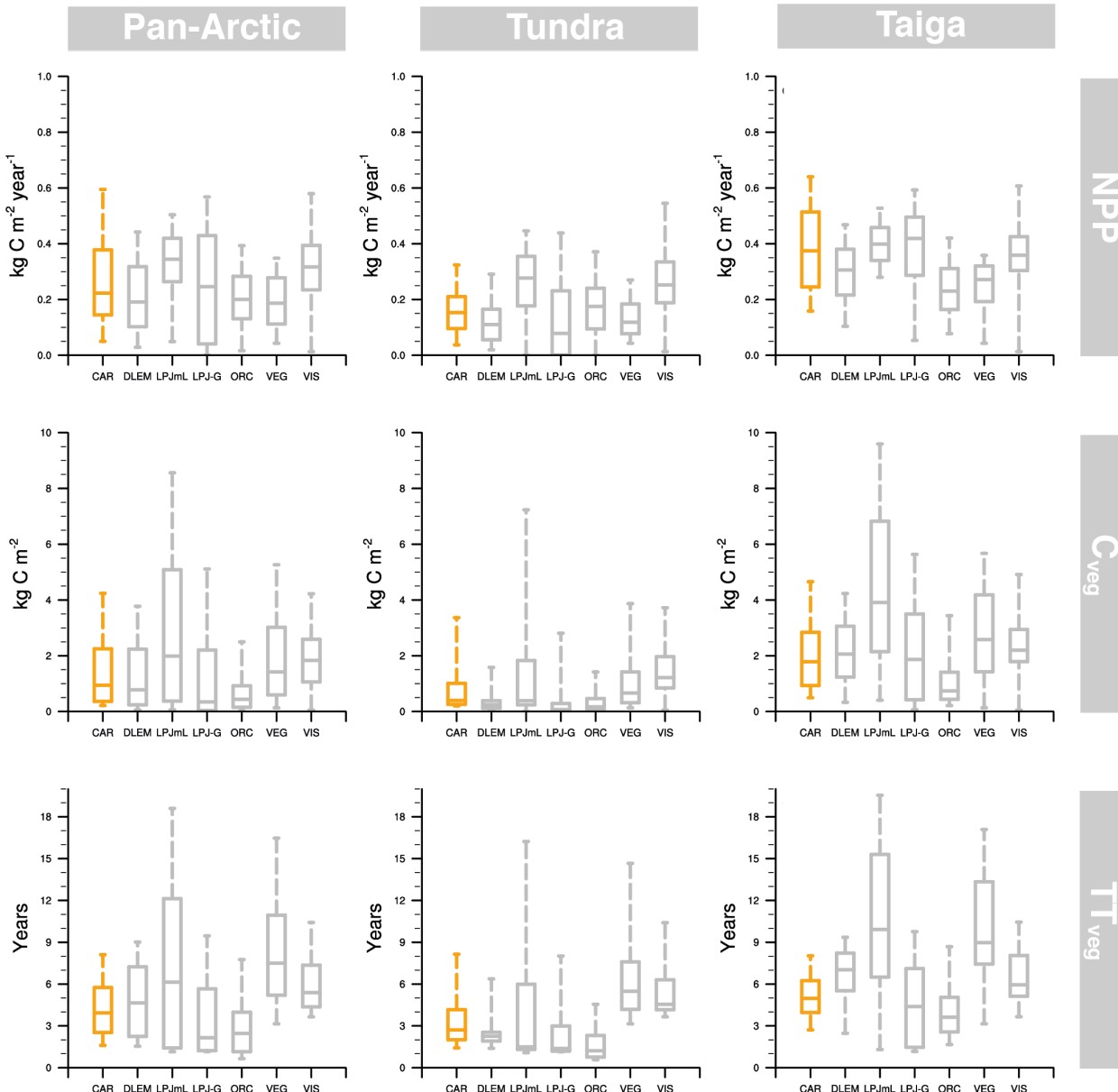

**Figure 5. Central tendency and variability of NPP [Net Primary Production], C_{veg} [Vegetation C stock], TT_{veg} [Vegetation transit time] estimated by CARDAMOM (orange) and ISI-MIP2a models (grey) in the Pan-Arctic, tundra and taiga regions. The box whisker plots comprise the estimations between the 5th and 95th percentiles, and the box encompasses the 25^{th} to 75^{th} percentiles. The line in each box mark the median of studied variables in each region.**

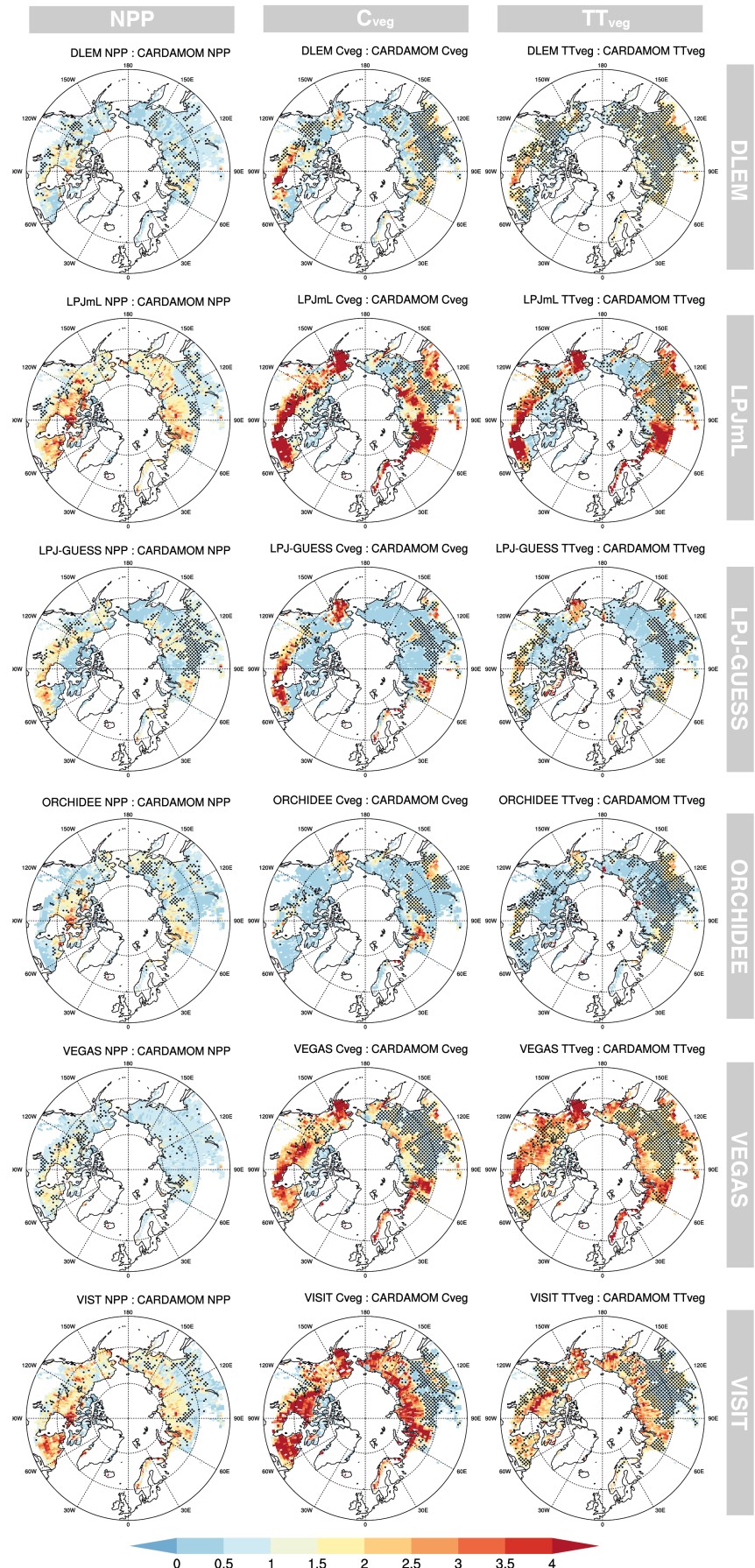

33

34

35 **Figure 6. NPP [Net Primary Production], $C_{veg}$ [Vegetation C stock] and $TT_{veg}$ [Vegetation transit time] ratios between ISI-MIP2a**

36 **model ensembles [DLEM, LPJmL, LPJ-GUESS, ORCHIDEE, VEGAS and VISIT] and CARDAMOM. Stippling indicates locations**

37 **where the ISI-MIP2a model mean is within the CARDAMOM's 5th and 95th percentiles.**

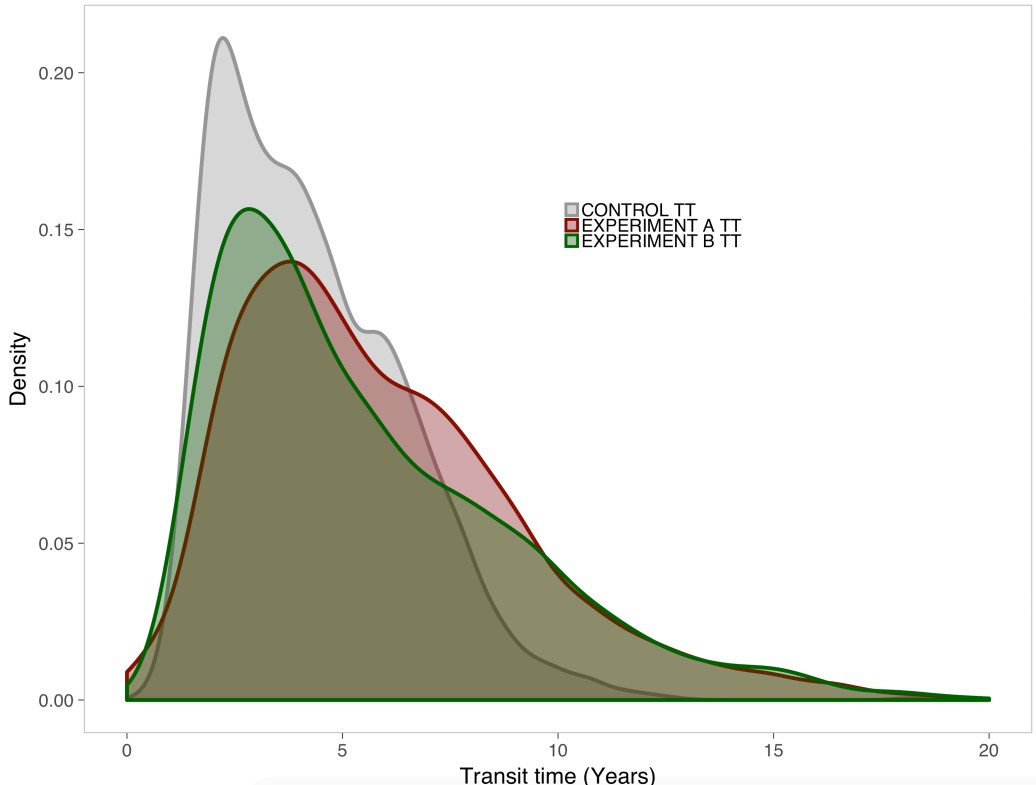

38

**Figure 7. Distribution functions derived from the attribution analysis used to estimate the origin of vegetation transit time (TT$_{veg}$)**

**bias from ISI-MIP2a models. The CONTROL TT (grey) includes both biomass (C$_{veg}$) and net primary production (NPP) estimated**

**by CARDAMOM. EXPERIMENT A TT (dark red) incorporates C$_{veg}$ from ISI-MIP2a and NPP from CARDAMOM while**

**EXPERIMENT B TT (dark green) includes NPP from ISI-MIP2a and C$_{veg}$ from CARDAMOM. The lower the overlapped area is**

**between control and experimental TT, the larger the contribution for TT biases is. For readability purposes, the scale in X-axis is**

**delimited to 20 years.**

45

**Table 1.** Multi-year (2000-2015) annual average of main ecosystem C fluxes [NEE, GPP, NPP, $R_{eco}$, $R_a$, $R_h$; g C m$^{-2}$ yr$^{-1}$], C stocks [$C_{photo}$, $C_{veg}$, $C_{dom}$, $C_{tot}$; kg C m$^{-2}$] and transit times [$TT_{photo}$, $TT_{veg}$, $TT_{dom}$, $TT_{tot}$; years] for the pan-Arctic, tundra (non-forested) and taiga (forested), region. The averages contain the the median in bold (50$^{th}$ percentile) and the uncertainty estimations across the 90% confidence range between the 5th and 95th percentiles assuming no spatial correlation between uncertainties in all pixels. We assume spatial correlation between pixels: P50, P05 and P95 represents the area-weighted aggregate of all pixels' media, P05 and P95.

| | | Pan-Arctic | | | | | Tundra | | | | | Taiga | | | | |
|---|---|---|---|---|---|---|---|---|---|---|---|---|---|---|---|---|
| | | P05 | P25 | P50 | P75 | P95 | P05 | P25 | P50 | P75 | P95 | P05 | P25 | P50 | P75 | P95 |
| **C fluxes** | **NEE** | -286.7 | -170.5 | **-67.4** | 149.9 | 1159.9 | -163.4 | -84.5 | **-14.9** | 176.2 | 1116.1 | -387.7 | -241.0 | **-110.4** | 128.2 | 1195.8 |
| | **GPP** | 427.8 | 504.1 | **565.0** | 633.5 | 740.7 | 236.8 | 285.0 | **327.2** | 379.4 | 463.3 | 584.1 | 683.5 | **759.8** | 841.6 | 967.9 |
| | **NPP** | 196.4 | 248.3 | **290.3** | 337.1 | 410.7 | 109.1 | 139.3 | **165.9** | 198.3 | 250.7 | 268.0 | 337.6 | **392.1** | 450.8 | 541.8 |
| | **$R_{eco}$** | 212.8 | 345.8 | **488.8** | 764.0 | 1854.1 | 124.3 | 211.9 | **310.0** | 540.3 | 1536.8 | 285.3 | 455.5 | **635.3** | 947.3 | 2114.0 |
| | **$R_a$** | 181.8 | 229.2 | **269.8** | 317.4 | 396.5 | 102.8 | 132.0 | **158.5** | 191.5 | 247.4 | 246.6 | 308.8 | **361.0** | 420.6 | 518.6 |
| | **$R_h$** | 31.0 | 116.6 | **219.0** | 446.6 | 1457.6 | 21.6 | 79.9 | **151.5** | 348.8 | 1289.4 | 38.7 | 146.7 | **274.3** | 526.7 | 1595.4 |
| **C pools** | **$C_{photo}$** | 0.1 | 0.1 | **0.1** | 0.2 | 0.2 | 0.1 | 0.1 | **0.1** | 0.1 | 0.2 | 0.1 | 0.1 | **0.2** | 0.2 | 0.3 |
| | **$C_{veg}$** | 0.5 | 1.0 | **1.5** | 2.6 | 5.8 | 0.3 | 0.5 | **0.8** | 2.0 | 6.8 | 0.8 | 1.4 | **2.1** | 3.1 | 5.1 |
| | **$C_{dom}$** | 10.3 | 18.3 | **24.4** | 32.2 | 47.5 | 10.0 | 17.4 | **23.3** | 31.0 | 46.1 | 10.5 | 19.0 | **25.3** | 33.2 | 48.6 |
| | **$C_{tot}$** | 11.8 | 20.0 | **26.3** | 34.5 | 51.0 | 10.8 | 18.4 | **24.6** | 33.0 | 50.6 | 12.7 | 21.3 | **27.7** | 35.7 | 51.2 |
| **Transit times** | **$TT_{photo}$** | 0.8 | 1.0 | **1.3** | 1.6 | 2.1 | 1.0 | 1.2 | **1.6** | 2.0 | 2.7 | 0.7 | 0.9 | **1.1** | 1.2 | 1.6 |
| | **$TT_{veg}$** | 1.7 | 2.8 | **4.5** | 7.5 | 15.7 | 1.4 | 2.2 | **3.4** | 5.9 | 12.7 | 1.9 | 3.2 | **5.3** | 8.8 | 18.2 |
| | **$TT_{dom}$** | 9.8 | 51.5 | **120.5** | 245.9 | 822.6 | 11.0 | 61.6 | **152.8** | 318.7 | 1055.9 | 8.7 | 43.3 | **94.1** | 186.3 | 631.4 |
| | **$TT_{tot}$** | 11.5 | 56.9 | **133.1** | 276.0 | 1013.6 | 12.5 | 67.0 | **167.7** | 357.6 | 1306.8 | 10.7 | 48.5 | **104.7** | 209.4 | 774.3 |

**Table 2. Parameter uncertainty reduction in percentage ranked from least (red) to most (blue) constrained in the pan-Arctic, tundra and taiga domains. The reduction percentage is calculated based on the difference between the 90% CI prior range and the 90% CI posterior range.**

| Parameter | Name | Process | Pan-Arctic | Tundra | Taiga |
|---|---|---|---|---|---|
| $MR_{litter}$ | Litter mineralization | Turnover | 3.3 | 3.6 | 2.9 |
| $TOR_{roots}$ | Root turnover | Turnover | 4.8 | 7.2 | 2.2 |
| $TOR_{wood}$ | Wood turnover | Turnover | 9.0 | 8.5 | 9.7 |
| $C_{litter}$ | Litter C stock | Stocks | 13.9 | 13.7 | 14.1 |
| $D_{rate}$ | Decomposition rate | Turnover | 18.2 | 18.6 | 17.8 |
| $f_{rau}$ | Fraction of GPP respired (Autotropic respiration) | Allocation | 30.9 | 31.7 | 30.2 |
| $L_f$ | Leaf fall duration | Phenology | 37.3 | 25.0 | 51.1 |
| LMA | Leaf mass per area | Phenology | 42.8 | 46.3 | 38.9 |
| $C_{roots}$ | Fine root C stock | Stocks | 52.4 | 72.1 | 30.3 |
| $R_l$ | Labile C release duration | Phenology | 53.1 | 52.0 | 54.4 |
| $f_{wood}$ | Fraction of NPP to wood C pool | Allocation | 65.8 | 68.1 | 63.3 |
| $F_{day}$ | Leaf fall day | Phenology | 67.0 | 51.1 | 84.8 |
| $MR_{som}$ | Soil organic matter mineralization | Turnover | 69.1 | 69.6 | 68.6 |
| $f_{labile}$ | Fraction of NPP to labile C pool | Allocation | 74.2 | 75.5 | 72.8 |
| $C_{eff}$ | Canopy efficency | Phenology | 74.7 | 75.5 | 73.7 |
| $f_{roots}$ | Fraction of NPP to roots C pool | Allocation | 75.7 | 74.7 | 76.8 |
| $B_{day}$ | Leaf onset day | Phenology | 76.2 | 67.4 | 86.1 |
| $C_{SOM}$ | Soil organic matter C stock | Stocks | 80.7 | 81.4 | 80.0 |
| L | Lifespan | Turnover | 83.4 | 76.4 | 91.4 |
| $f_{foliar}$ | Fraction of NPP to foliage C pool | Allocation | 88.0 | 88.6 | 87.4 |
| $C_{labile}$ | Labile C stock | Stocks | 92.2 | 95.3 | 88.8 |
| $C_{wood}$ | Woody C stock | Stocks | 92.6 | 90.1 | 95.5 |
| $C_{foliar}$ | Foliar C stock | Stocks | 95.2 | 96.0 | 94.3 |

**Table 3. Statistics of linear fit between the CARDAMOM framework (independent) and the ISI-MIP2a models (dependent) per individual model and per NPP [Net Primary Production; kg C m$^{-2}$ yr$^{-1}$], C$_{veg}$ [Vegetation C stock; kg C m$^{-2}$] and TT$_{veg}$ [Vegetation transit time; years]. The units for RMSE and bias are kg C m$^{-2}$ yr$^{-1}$ in NPP, kg C m$^{-2}$ yr$^{-1}$ in C$_{veg}$ and years in TT$_{veg}$.**

| | | Panarctic | | | | | Tundra | | | | | Taiga | | | | |
|---|---|---|---|---|---|---|---|---|---|---|---|---|---|---|---|---|
| | | Intercept | Slope | R$^2$ | RMSE | Bias | Intercept | Slope | R$^2$ | RMSE | Bias | Intercept | Slope | R$^2$ | RMSE | Bias |
| NPP (kg C m$^{-2}$ y$^{-1}$) | DLEM | 0.04 | 0.61 | 0.58 | 0.09 | -0.07 | 0.04 | 0.48 | 0.23 | 0.08 | -0.05 | 0.12 | 0.47 | 0.44 | 0.08 | -0.09 |
| | LPJmL | 0.19 | 0.51 | 0.43 | 0.10 | 0.06 | 0.12 | 0.88 | 0.38 | 0.10 | 0.10 | 0.31 | 0.23 | 0.21 | 0.07 | 0.02 |
| | LPJ-GUESS | 0.01 | 0.93 | 0.61 | 0.12 | -0.01 | -0.03 | 1.00 | 0.38 | 0.12 | -0.03 | 0.13 | 0.67 | 0.45 | 0.12 | 0.00 |
| | ORCHIDEE | 0.14 | 0.27 | 0.17 | 0.10 | -0.06 | 0.07 | 0.64 | 0.31 | 0.09 | 0.01 | 0.20 | 0.12 | 0.03 | 0.10 | -0.14 |
| | VEGAS | 0.07 | 0.46 | 0.60 | 0.06 | -0.07 | 0.05 | 0.55 | 0.36 | 0.07 | -0.02 | 0.12 | 0.36 | 0.52 | 0.05 | -0.13 |
| | VISIT | 0.18 | 0.47 | 0.26 | 0.13 | 0.04 | 0.10 | 0.95 | 0.30 | 0.13 | 0.09 | 0.30 | 0.18 | 0.06 | 0.12 | -0.01 |
| C$_{veg}$ (kg C m$^{-2}$) | DLEM | 0.44 | 0.61 | 0.40 | 1.00 | -0.13 | 0.38 | 0.10 | 0.03 | 0.65 | -0.37 | 0.92 | 0.61 | 0.43 | 0.91 | 0.11 |
| | LPJmL | 1.70 | 0.88 | 0.15 | 2.80 | 1.48 | 1.40 | 0.16 | 0.01 | 2.30 | 0.65 | 2.80 | 0.79 | 0.12 | 2.80 | 2.42 |
| | LPJ-GUESS | 0.30 | 0.69 | 0.30 | 1.40 | -0.15 | 0.37 | 0.13 | 0.02 | 0.95 | -0.41 | 0.51 | 0.81 | 0.33 | 1.50 | 0.13 |
| | ORCHIDEE | 0.40 | 0.23 | 0.12 | 0.82 | -0.71 | 0.33 | 0.04 | 0.01 | 0.46 | -0.50 | 0.71 | 0.20 | 0.06 | 1.00 | -0.94 |
| | VEGAS | 1.10 | 0.64 | 0.27 | 1.40 | 0.58 | 1.20 | 0.10 | 0.01 | 1.30 | 0.37 | 1.30 | 0.76 | 0.38 | 1.30 | 0.80 |
| | VISIT | 1.60 | 0.23 | 0.06 | 1.30 | 0.49 | 1.40 | 0.03 | 0.00 | 1.10 | 0.53 | 2.30 | 0.11 | 0.01 | 1.30 | 0.44 |
| TT$_{veg}$ (yr) | DLEM | 1.90 | 0.69 | 0.29 | 2.30 | 0.56 | 2.30 | 0.18 | 0.05 | 1.80 | -0.42 | 3.40 | 0.63 | 0.29 | 1.70 | 1.56 |
| | LPJmL | 4.00 | 0.75 | 0.07 | 6.10 | 2.91 | 4.10 | 0.08 | 0.00 | 5.10 | 0.82 | 7.30 | 0.60 | 0.03 | 5.80 | 5.27 |
| | LPJ-GUESS | 1.30 | 0.54 | 0.14 | 2.90 | -0.68 | 1.70 | 0.28 | 0.04 | 2.90 | -0.81 | 0.95 | 0.71 | 0.16 | 2.80 | -0.53 |
| | ORCHIDEE | 1.40 | 0.34 | 0.10 | 2.20 | -1.42 | 1.60 | 0.04 | 0.00 | 1.70 | -1.78 | 2.30 | 0.35 | 0.07 | 2.10 | -1.03 |
| | VEGAS | 5.90 | 0.62 | 0.11 | 3.90 | 4.23 | 6.60 | 0.10 | 0.00 | 3.80 | 3.42 | 5.50 | 0.93 | 0.17 | 3.40 | 5.12 |
| | VISIT | 5.40 | 0.12 | 0.01 | 2.30 | 1.65 | 5.20 | 0.06 | 0.00 | 2.30 | 1.92 | 6.70 | -0.04 | 0.00 | 2.10 | 1.36 |