# Peer review of "Evaluation of terrestrial pan-Arctic carbon cycling using a data-assimilation system"

_Earth System Dynamics, 2018_

## Referee Comment (RC1) · M. Forkel (Referee) · 20 Jun 2018

Review of "Evaluation of terrestrial pan-Arctic carbon cycling using a dataassimilation system"

Matthias Forkel, Wien, 2018-06-20

1 Summary

López-Blanco et al. apply a land carbon data assimilation system to assess carbon fluxes, stocks, and turnover times in arctic and boreal regions. Within the CARDAMON system, parameters of the DALEC2 model are optimized per 1° grid cell against observational datasets of LAI, biomass, and soil organic carbon. From the optimized model, carbon stocks, fluxes and turnover times are computed and then compared against results from global vegetation models (GVMs). The approach is very valuable because the carbon turnover in land ecosystem is a main uncertain feature of the global carbon cycle. I really appreciate this work; however, the paper needs substantial revisions before I can recommend publication in ESD (see major comments).

Also the structure of many chapters needs to be revised because information are either repeated often at several places or is not given at the appropriate places (see specific comments).

2 Major comments

2.1 Tundra-taiga transition and Mongolian grasslands

The grassland region in Mongolia is rather a "steppe" than a tundra (lines 110-112, Fig. S1). Please separate steppe and tundra by either using a latitude threshold, temperature conditions, or a biome map.

2.2 Computation of transit times

Based on our theoretical assumptions on carbon turnover times (Carvalhais et al., 2014) [supplement], your computations of transit times are partly wrong. The turnover (or transit) time is defined by the C stock of a carbon pool and its outgoing flux. For example the transit time of vegetation is TT_vegetation = biomass / T whereby T includes all processes that remove C from vegetation (litter fall, disturbance, mortality, etc.). Under the steady state assumption (i.e. T = NPP), the transit time of the entire vegetation can be defined as TT_veg = biomass / NPP. Accordingly for the entire ecosystem, the transit time can be defined under the steady state assumption as TT_eco = (biomass + SOC) / Reco = (biomass + SOC) / GPP.

In your calculations, all transit time are computed based on NPP However, only a fraction of NPP goes into the different C pools which is in DELC well defined based on

the allocation parameters. Hence the correct computation of a transit time for a certain carbon pool should be based on a fraction of NPP:

NPP_photo = a_foliage * NPP + a_labile * NPP [I assume that C_photo contains the foliar and labile C pools of DALEC2 but this is not described in the paper.]

TT_photo = C_photo / NPP_photo

TT_veg = C_veg / (NPP – NPP_photo) [should this rather be named TT_wood?]

TT_soil = C_soil / litterfall [?] = C_soil / Rh [Why you name this TT_dom?]

2.3 GVMs with GCM climate forcing

What is the reason for using GVM results that are based on climate forcing from GCMs (lines 176-182)? ISIMIP provides also historical forcing that is based on observed climate data and at least LPJmL provides also model output based on historical data (ISIMIP2A). I assume that the historical climate data better represents climate conditions than the (even though bias-corrected) GCM outputs. Differences in climate forcing can have huge impacts in GVMs. Hence, the comparison between CARDAMON (forced with reanalysis data) and GVM outputs is per se unfair and not comparable.

I request that the comparison between CARDAMON and GVMs should be made comparable by either taking GVM outputs from the historical forcing with ERA-Interim data or by running the optimized CARDAMON with the same GCM forcing.

2.4 Biases with biomass and GPP – wrong use of data and parameter uncertainties?

CARDAMON underestimates the biomass and FLUXCOM GPP. The overestimation of FLUXNET GPP is contradictory but the source of the mismatch is almost impossible to assess given the scale mismatch between FLUXNET sites and 1° grid cells. However, if we would assume that both biomass and FLUXCOM GPP are consistent; this could tell us that CARDAMON only needs a higher GPP to gain higher biomass. I'm wondering if it was actually possible to constrain both biomass and GPP within the assimilation

framework. Were there any prior parameters used that constrain GPP? As far as I can understand the setup of the approach, there were no data and no parameters included that would constrain GPP (apart from LAI that, however, likely only constrains the seasonality of GPP). To better understand the assimilation results, it is necessary to show maps of reduction of uncertainty of each DALEC2 model parameter. Which parameters were mostly reduced (phenology, allocation, C pools, turnover rates)?

Please also note that the biomass map by Thurner et al. (2014) is largely in agreement with in situ observations of forest carbon density in Russia and slightly underestimates in the USA. If CARDAMON underestimates the biomass, this implies that it would even stronger underestimate the in situ observations than the biomass map. From the results, I get the feeling that the assimilation is over-confident in the SOC data and degrades the performance with the biomass map. Hence the key question is how data uncertainties were used as weights in the assimilation? The uncertainties in SOC are much larger than in biomass (Carvalhais et al., 2014); so I expect that CARDAMON should rather fit the biomass map than the SOC map if these data uncertainties were correctly used.

In summary, please report:

1. Which data uncertainties were used and how they were included in the assimilation;

2. How parameter prior uncertainties were included in the assimilation;

3. How the cost function was designed and the different datasets weighted;

4. What are the changes between prior and posterior parameter uncertainties.

2.5 Benchmarking ISIMIP with CARDAMON

At this point, I will not further comment on chapter 3.3. given the inconsistencies in climate forcing between CARDAMON and GVMs and given the fact that it is not clear how data uncertainties were treated in the assimilation and hence affect the CARDAMON results.

3 Specific comments

1 Introduction: I suggest to slightly restructure the Introduction to make things a bit more clear. For example, several topics are mentioned twice: "transit times" (around lines 48 and 76), the available data (lines 46-59 and 83-86), and the specific features of the arctic carbon cycle (1. Paragraph, lines 55-59). In addition , the meaning of "transit times" is never explained. I suggest to :

- keep the first paragraph as it is,

- to rewrite the second paragraph: define "transit" and/or "turnover" and /or "residence" time and why it is important,

- to write in the third paragraph about the available in situ and satellite-based data to assess "turnover" times and the associated uncertainties,

- to write in the fourth paragraph about the inabilities and uncertainties of GVMs with respect to turnover times,

- and finally to present model-data integration and CARDAMON as the potential "solution" in the last paragraph including the definition of your objectives.

Line 37: Use either "warming" or "temperature increase" but not "warming increase" because this would be an acceleration in temperature increase.

Line 40-41: In addition to Lucht et al. and Myneni et al., you could also cite more recent related publications (Forkel et al., 2016; Graven et al., 2013; Zhu et al., 2016) [I don't request to include my paper!]

Line 48 and lines 75-82: "transit times" – Carvalhais et al. use "turnover" time, Friend et al. "residence" time, and Thurner et al. (2016) "turnover rate". Is there a reason why you use "transit time" and why you are not using one of the other terms? Please provide a short definition of these terms or the term that you are using and how they differ.

[Figure]

Line 61: "PFT or spin-up": The "or" should be replaced by "and".

Line 74: Relevant is also the work by Thurner et al. (2016)

Lines 84-85: Please provide references.

Line 87: A reference to a general overview paper on model-data integration might be useful.

Line 95-96: It is not clear to me how your analysis will provide further insight into GVMs that goes beyond the work of Friend et al. (2014), Carvalhais et al. (2014), and Thurner et al. (2017). Please make clear what kind of additional knowledge you are expecting from your analysis on the problems of GVMs.

Lines 106-107: Please define which classes you used to separate forest and non-forest.

Section 2.2: The description of CARDAMON refers mostly to previous work. However, to understand _this_ paper, I suggest to provide some more details or equations with respect to the following questions:

- LAI, biomass, and SOC are used as data sets in a cost function for parameter estimation and not as forcing data. Is this correct?

- What is the cost function? How are the differences in the number of data points weighted (LAI is a time series, SOC and biomass only single values per grid cell)?

- Why is the MHMCMC algorithm used three times? Does it not explore the full parameter space if it is applied only once? Or are there difference in initial values?

- Can you make a conceptual figure that shows which data sets go into the assimilation and which are only used as independent evaluation data?

Line 147: What is the difference between "photosynthetic" and "vegetation" C stocks? Is photosynthesis not vegetation?
Line 159: Did you directly compare the 1° grid cell with the FLUXNET site? If yes, how are the FLUXNET sites representative for the 1° grid cell? If no, did you run CARDAMON with the site meteorological data? Add: Only from the discussion (lines 318-238), I now learn that you did a grid cell to point comparison. This should be already mentioned in the methods and be recalled at the appropriate place in the results section.

Lines 166-168: This is a repetition from lines 123-125. Please merge the two sentences.

Line 171: TT_veg was already mentioned at line 148. I suggest to remove both occurrence of TT_veg and to already define TT_veg in the new second paragraph of the introduction.

Line 171: At the end CARDAMON is also just only a GVM but with grid cell-specific parameters. I don't see how CARDAMON then serve as a benchmark for the other GVMs. Would it be not enough to directly benchmark the GVMs against the reference data? You should try to better motivate already in the Introduction why you can use CARDAMON as a benchmark for GVMs.

Line 173: LPJmL (capital L). Please indicate which version of LPJmL was used. Is it the most recent version (LPJmL4) (Schaphoff et al., 2018a)? LPJmL4 includes also a new permafrost module (Schaphoff et al., 2013) and a data-constrained phenology module (Forkel et al., 2014) and hence better reproduces boreal and arctic carbon stocks and carbon cycling than the previous versions (Forkel et al., 2016; Schaphoff et al., 2018b).

Lines 184-194: I got really confused by this paragraph because initially I got the impression that you "jump" across all results without explanation. Please make clear that this paragraph is a summary of all results by either using a heading or a suitable topic sentence.

Line 202: Which of the used data sets constrain the separation between GPP and NPP?

Line 216: Do you find spatial pattern in TT_photo that would resample the distribution of evergreen and deciduous trees?

Line 219: "Interestingly" – Please tell me why this is "interestingly".

Line 257: "as noted " – Please check.

Lines 330-331: This sentence should be merged with the numbers given at lines 349-352.

Line 341: Are you sure to use the right reference for LPJ-GUESS-WhyMe?

References

Carvalhais, N., Forkel, M., Khomik, M., Bellarby, J., Jung, M., Migliavacca, M., ÎlJu, M., Saatchi, S., Santoro, M., Thurner, M., Weber, U., Ahrens, B., Beer, C., Cescatti, A., Randerson, J. T. and Reichstein, M.: Global covariation of carbon turnover times with climate in terrestrial ecosystems, Nature, 514(7521), 213–217, doi:10.1038/nature13731, 2014.

Forkel, M., Carvalhais, N., Schaphoff, S., v. Bloh, W., Migliavacca, M., Thurner, M. and Thonicke, K.: Identifying environmental controls on vegetation greenness phenology through model–data integration, Biogeosciences, 11(23), 7025–7050, doi:10.5194/bg-11-7025-2014, 2014.

Forkel, M., Carvalhais, N., Rödenbeck, C., Keeling, R., Heimann, M., Thonicke, K., Zaehle, S. and Reichstein, M.: Enhanced seasonal CO2 exchange caused by amplified plant productivity in northern ecosystems, Science, aac4971, doi:10.1126/science.aac4971, 2016.

Graven, H. D., Keeling, R. F., Piper, S. C., Patra, P. K., Stephens, B. B., Wofsy, S. C., Welp, L. R., Sweeney, C., Tans, P. P., Kelley, J. J., Daube, B. C., Kort, E. A., Santoni,

G. W. and Bent, J. D.: Enhanced Seasonal Exchange of $CO_2$ by Northern Ecosystems Since 1960, Science, 341(6150), 1085–1089, doi:10.1126/science.1239207, 2013.

Schaphoff, S., Heyder, U., Ostberg, S., Gerten, D., Heinke, J. and Lucht, W.: Contribution of permafrost soils to the global carbon budget, Environ. Res. Lett., 8(1), 014026, doi:10.1088/1748-9326/8/1/014026, 2013.

Schaphoff, S., von Bloh, W., Rammig, A., Thonicke, K., Biemans, H., Forkel, M., Gerten, D., Heinke, J., Jägermeyr, J., Knauer, J., Langerwisch, F., Lucht, W., Müller, C., Rolinski, S. and Waha, K.: LPJmL4 – a dynamic global vegetation model with managed land – Part 1: Model description, Geosci Model Dev, 11(4), 1343–1375, doi:10.5194/gmd-11-1343-2018, 2018a.

Schaphoff, S., Forkel, M., Müller, C., Knauer, J., von Bloh, W., Gerten, D., Jägermeyr, J., Lucht, W., Rammig, A., Thonicke, K. and Waha, K.: LPJmL4 – a dynamic global vegetation model with managed land – Part 2: Model evaluation, Geosci Model Dev, 11(4), 1377–1403, doi:10.5194/gmd-11-1377-2018, 2018b.

Thurner, M., Beer, C., Santoro, M., Carvalhais, N., Wutzler, T., Schepaschenko, D., Shvidenko, A., Kompter, E., Ahrens, B., Levick, S. R. and Schmullius, C.: Carbon stock and density of northern boreal and temperate forests, Glob. Ecol. Biogeogr., 23(3), 297–310, doi:10.1111/geb.12125, 2014.

Thurner, M., Beer, C., Carvalhais, N., Forkel, M., Santoro, M., Tum, M. and Schmullius, C.: Large‐scale variation in boreal and temperate forest carbon turnover rate is related to climate, Geophys. Res. Lett., doi:10.1002/2016GL068794, 2016.

Zhu, Z., Piao, S., Myneni, R. B., Huang, M., Zeng, Z., Canadell, J. G., Ciais, P., Sitch, S., Friedlingstein, P., Arneth, A., Cao, C., Cheng, L., Kato, E., Koven, C., Li, Y., Lian, X., Liu, Y., Liu, R., Mao, J., Pan, Y., Peng, S., Peñuelas, J., Poulter, B., Pugh, T. A. M., Stocker, B. D., Viovy, N., Wang, X., Wang, Y., Xiao, Z., Yang, H., Zaehle, S. and Zeng, N.: Greening of the Earth and its drivers, Nat. Clim. Change, 6(8), 791–795,

doi:10.1038/nclimate3004, 2016.

---

## Referee Comment (RC2) · Anonymous Referee #2 · 7 Aug 2018

In this paper the authors take the CARDOMOM + DALEC Bayesian calibration system an apply it specifically to the arctic using a number of regional-scale data products. Once the model is fit to data, it is then used to assess carbon pools and benchmark global vegetation models. The scale and scope of the analysis is quite impressive – building up their system to this point was clearly a lot of work and the attempt to synthesize multiple data constraints at a regional scale is really important, especially for a highly influential and understudied region like the arctic.

That said, I do have a few high level concerns about what the authors have done. The easiest of these to address is that the details of what was actually done was insufficient and teasing out important high-level facets of CARDOMOM are left to the reader tracking down earlier papers. Particularly important is to clarify whether DALEC is calibrated independently for every pixel, in some sort of spatially correlated manner, or with a single parameterization for the two PFTs across the whole region. My recollection from earlier papers made me think the first (independent fits), but in reading the results it is hard to distinguish parameter uncertainty from parameter spatial heterogeneity. The authors need to be more explicit about this. Likewise, the authors need to be more clear about whether this is really a data assimilation system, or if it's just a calibration system. This matters because in DA (e.g. EnKF) the analysis provides a formal synthesis of observations and process understanding, but in a calibration system your estimated states are ultimately just a forward model run. To me, it feels like the authors are treating a forward model run as if it were a reanalysis product. If this is true what the authors did is still valuable but they should be more open about this and the limitations of this approach.

Second, in light of the earlier point about reanalysis vs forward simulation, I am really uncomfortable about the author's use of their model as benchmark for other models. This is particularly true given the non-trivial biases in some of the verification (biomass) and validation (GPP, Rh) analyses and the lack of independent validation of a number of the other processes in the model (e.g. turnover). I think this manuscript could stand alone without the GVM component.

Third, I'm really concerned about how the authors assimilate these derived data products. There's not really any discussion of how the observation errors in the data and process error in the model are treated. There's not any discussion of how the authors handled the non-independence of spatial pixels in these data products. Indeed the authors seem to treat data products as if they are truly data, which likely results in an overestimation of the true information content in the data. For example, if I have 10 observations I can Krige a map that has 10k grid cells, but my true sample size remains 10 not 10k and any data assimilation system needs to reflect that.

Detailed Comments:

L126: 1) Is calibration really data assimilation? 2) inclusion of process error?

L135: What is the actual underlying sample size? Derived data products can massively conflate the actual information content. Errors in these data products are hugely autocorrelated and that observation uncertainty is not captured correctly in these products. Also, many of these constraints are not data (GPP, LAI, biomass) but just different models.

L144: 500 samples per chain? That's way too small. Also, what's the effective sample size after accounting for autocorrelation? I'd recommend the authors shoot for an effective sample size around ~5000 total, which likely will require a much larger total number of samples given their reliance on Metropolis-Hastings. Not stated explicitly whether this is one global parameter set or one per grid cell? My memory from Bloom et al 2016 is the latter.

L146: A 90% CI is typical. Reason for not 95% norm?

L154: This isn't independent of the calibration product

L164: should really include the 95% CI in addition to the interquartile

L169: You can't compare a complex model against a (mis)calibrated simple model and call it a benchmark. Especially true if you're looking at the marginal distributions of indirectly inferred latent variables.

L177: If looking at the historical period, why weren't models run under reanalysis meteorology rather than GCMs?

L184: Drop this whole paragraph – it's a bit confusing to give a summary of the results before presenting the results without making it clear that this is a summary of highlights. Right now it just feel like you're going though the results really quickly without much explanation.

L187: A 28% bias against the data that the model was calibrated to seems like a pretty big problem.

L190: "This mismatch is important in the context of FLUXCOM, as noted" what do you mean "as noted" you never noted anything

L203: "and marginally varied across tundra" I don't understand what you mean here

L209: Distinguish tundra and taiga. These numbers don't seem plausible for tundra

L211: That the tundra numbers are so close to the taiga numbers doesn't seem correct. How well do these numbers validate against direct field data (not derived data products)?

L216: A transit time of 4.3 years in the woody tissues of a spruce tree seems really fast give their lifespan. How does this compare to field data (e.g. isotopes)

L217: The CI on the SOM is really large (essentially 10-1000 years). Is this just the prior?

L228: This results needs additional explanation with regards to what this test statistic applies to. You calibrated a mechanistic model via MCMC, this isn't a t-test. What specifically changed that much?

L234: What do you mean priors, isn't this the data?

L257: This is almost the exact same sentence as L190

L295: Is this statement that CARDOMOM is more sensitive biomass than soil C really fair? In one case you're comparing whether a data constraint is included at all, while in the other your comparing different derived data products, which are likely relying on similar underlying raw data. I think for this to be fair you would want to include a version where you don't have any soil C constraint.

L308: There's a 28% bias in biomass, how is that "good agreement". The Discussion

seems to be missing the critical point that if a model is faced with multiple constraints and can't reconcile them then there's either inconsistencies in the data, structural errors in the model, or both. And why is there no comparison to LAI and GPP constraints? Also, there seems to be no discussion of how observations error in the data are derived/treated and how you're handling the process error in the model (is this a fit parameter or just ignored).

L312: I haven't looked into the details of the Jung 2011 product vs the Jung 2017 product, but I'm skeptical that these are independent. Would be good to state more explicitly what each product is upscaling to generate GPP (FLUXNET? SIF?). If they're both FLUXNET-based then they're not independent if they're just applying different algorithms to upscale the same underlying data.

L314: "One difference between these two models is..." What two models?

L317: I'd recommend making this sentence the start of the next paragraph

L319: How do you know that the issue is only one of scale difference, and not some other error in the model or DA system? What could you do to confirm this (e.g. run with local drivers)?

L326: This error in timing is an example of why it might be better to run a system that performs both state and parameter data assimilation, rather than just parameters.

L328: It's a bit surprising that you're running a model in the arctic that doesn't include snow or permafrost. I see that this point is in the Discussion, but it seems really important to be more upfront about this earlier in the paper, as it's a pretty limiting assumption and should lead to greater caution in how confidently you interpret the results. It also begs the question as to why you didn't couple CARDOMOM to a more sophisticated land model for this analysis.

L365: But is there any direct field constraint (e.g. isotope data)

---

## Author Comment (AC1)

Final author response to **"Evaluation of terrestrial pan-Arctic carbon cycling using a data-assimilation system"**

Efrén López-Blanco, Jean-François Exbrayat, Magnus Lund, Torben R. Christensen, Mikkel P. Tamstorf, Darren Slevin, Gustaf Hugelius, Anthony A. Bloom, Mathew Williams

October 24th, 2018

We thank the two reviewers for their ideas and suggestions to improve this paper. We have carefully considered them all and changed our manuscript accordingly. In the following we include a point-by-point response to the reviews, and attached a marked-up manuscript version showing the differences to the initially submitted version. Please note that the line numbers point to the non-marked manuscript.

In general lines we cautiously considered each of the **REF#1** and **REF#2** comments, paying special attention to the tundra-taiga domain split, biases with biomass and GPP, the uncertainty reduction of the model parameters, and the GVM benchmarking exercise. We have included detailed answers to the questions raised, and have performed additional implementations, tests and changes to provide more support on these items. We paid special attention to 6 issues:

- We revised the paper to specifically improve the introduction to facilitate a better understanding of the research context. We highlighted the explanation of transit time, its importance and the difference with other similar terminology used in literature.
- The structure of some sections has been modified to improve comprehension and readability. Now it reads as:

[Figure]

- We improved the method section with supporting equations and supplementary tables to better describe the experimental design and the data-assimilation framework.
- We implemented a new conceptual diagram in Figure 1 summarizing the C flux, stocks and transit times numbers described in section 3.1. We also implemented a new Table 2 to illustrate the uncertainty reduction (priors vs posteriors) from all parameters in the model. Similarly, Figure 2, 3, 4, 5, 6, and 7, as well as Table 1, 3 were all implemented as result of the specific requirements from the Referees.
- We substituted ISI-MIP models (forced with GCMs data) for ISI-MIP2a models (forced with ERA-Interim data) in the benchmarking analysis. Apart from the fact that these new models are run with similar climatic forcing, the ISI-MIP2a models also cover a larger temporal period (2000-2010) compared to the old 2000-2004 period.
- We carefully edit the paper for clarity and communication (wrong wording and repetitions). Also, we incorporated a list of acronyms in the supplementary material to facilitate the large number of variable names, parameters and models.
* * *
**Review of "Evaluation of terrestrial pan-Arctic carbon cycling using a data- assimilation system"**

**Matthias Forkel, Wien, 2018-06-20**
* * *
**1 Summary**

**López-Blanco et al. apply a land carbon data assimilation system to assess carbon fluxes, stocks, and turnover times in arctic and boreal regions. Within the CARDAMON system, parameters of the DALEC2 model are optimized per 1◦ grid cell against observational datasets of LAI, biomass, and soil organic carbon. From the optimized model, carbon stocks, fluxes and turnover times are computed and then compared against results from global vegetation models (GVMs). The approach is very valuable because the carbon turnover in land ecosystem is a main uncertain feature of the global carbon cycle. I really appreciate this work; however, the paper needs substantial revisions before I can recommend publication in ESD (see major comments). Also the structure of many chapters needs to be revised because information are either repeated often at several places or is not given at the appropriate places (see specific comments).**

We thank Matthias Forkel for taking the time to assess our manuscript. We believe the comments have substantially improved the manuscript. We thoroughly considered each of the comments, paying special attention to the structure of the paper, to the tundra-taiga domain split, biases with biomass and GPP, the uncertainty reduction of the model parameters, and the GVM benchmarking exercise with the required ISI-MIP2a comparison

**2 Major comments**

**2.1 Tundra-taiga transition and Mongolian grasslands**

**The grassland region in Mongolia is rather a "steppe" than a tundra (lines 110-112, Fig. S1). Please separate steppe and tundra by either using a latitude threshold, temperature conditions, or a biome map.**

The reviewer is correct that the steppe should not be considered as tundra in lines 132-134, Fig. S1. We decided to remove the grasslands/steppes, and focus on higher latitudes and Arctic ecosystems consistent with the focus of the paper. We set the southern boundary of the taiga as our limit - that cuts out the Mongolian grasslands and looks more natural (see Figure S1).

**2.2 Computation of transit times**

**Based on our theoretical assumptions on carbon turnover times (Carvalhais et al., 2014) [supplement], your computations of transit times are partly wrong. The turnover (or transit) time is defined by the C stock of a carbon pool and its outgoing flux. For example the transit time of vegetation is TT_vegetation = biomass / T whereby T includes all processes that remove C from vegetation (litter fall, disturbance, mortality, etc.). Under the steady state assumption (i.e. T = NPP), the transit time of the entire vegetation can be defined as TT_veg = biomass / NPP. Accordingly for the entire ecosystem, the transit time can be defined under the steady state assumption as TT_eco = (biomass + SOC) / Reco = (biomass + SOC) / GPP.**
**In your calculations, all transit time are computed based on NPP However, only a fraction of NPP goes into the different C pools which is in DELC well defined based on the allocation parameters. Hence the correct computation of a transit time for a certain carbon pool should be based on a fraction of NPP:**
**NPP_photo = a_foliage * NPP + a_labile * NPP [I assume that C_photo contains the foliar and labile C pools of DALEC2 but this is not described in the paper.]**
**TT_photo = C_photo / NPP_photo**
**TT_veg = C_veg / (NPP – NPP_photo) [should this rather be named TT_wood?] TT_soil = C_soil / litterfall [?] = C_soil / Rh [Why you name this TT_dom?]**

First, we need to clarify the terminology used here: $C_{photo}$ indeed corresponds to the sum of the foliar and labile pools, $C_{veg}$ refers to the sum of all vegetation pools (foliar, labile, wood and roots) while $C_{dom}$ is the sum of the litter and SOM pools.
We were not clear here - we have used C input into each pool to calculate each pool's transit time (as Matthias suggests). The text in the original paper section 2.2 was wrong in saying:

> "In this study, we addressed C turnover rates and decomposition processes as their inverse rates, this is the C transit time ($TT_{photo}$, $TT_{veg}$ and $TT_{dom}$), represented as the ratio between each C stock and NPP."

The last bit should be (S2.2.1, L155-158):

> "In this study, we addressed C turnover rates and decomposition processes as their inverse rates, this is the C transit time ($TT_{photo}$, $TT_{veg}$ and $TT_{dom}$), represented as the ratio between the mean C stock and the mean C input into that stock during the simulation period.".

To clarify, the transit time for each C pool at grid cell is derived following Bloom et al. (2016) procedure (explained in detail in their Supplementary material, equation 8) as:

$$TT_{pool} = \frac{C_{pool}}{F_{in} - \Delta C_{pool}}$$

Where $C_{pool}$ is the mean pool size, $F_{in}$ is the mean annual C pool input, and $\Delta C_{pool}$ is the mean annual change in pool size through 2000-2015 used to correct the calculation for any changes in mean stocks over the study period.

**2.3 GVMs with GCM climate forcing**

**What is the reason for using GVM results that are based on climate forcing from GCMs (lines 176-182)? ISIMIP provides also historical forcing that is based on observed climate data and at least LPJmL provides also model output based on historical data (ISIMIP2A). I assume that the historical climate data better represents climate conditions than the (even though bias-corrected) GCM outputs. Differences in climate forcing can have huge impacts in GVMs. Hence, the comparison between CARDAMON (forced with reanalysis data) and GVM outputs is per se unfair and not comparable.**
**I request that the comparison between CARDAMON and GVMs should be made comparable by either taking GVM outputs from the historical forcing with ERA-Interim data or by running the optimized CARDAMON with the same GCM forcing.**

We understand Matthias' point about consistency between climate drivers. Therefore, we downloaded and used the ISIMIP2a simulations instead of the original ISIMIP1. This new implementation presents many advantages like a longer overlap of simulations (2001-2010) and more similar climate drivers based on the ERA-Interim reanalysis. Overall ISI-MIP2a models are closer related to CARDAMOM than ISI-MIP. ISI-MIP2a models estimated lower NPP, $C_{veg}$ and $TT_{veg}$ than ISI-MIP, and the uncertainties were also lower. In general, we found higher $R^2$ and lower RMSEs in NPP than $C_{veg}$ and $TT_{veg}$. Moreover, by using ISI-MIP2a models we remove odd comparisons such as with HYBRID, which may have led to a bias in the attribution analysis we earlier preformed.

We therefore changed the method's part related to ISI-MIP2a (S2.4, L223-232):

[revised manuscript text omitted]

**2.4 Biases with biomass and GPP – wrong use of data and parameter uncertainties?**
**CARDAMON underestimates the biomass and FLUXCOM GPP. The overestimation of FLUXNET GPP is contradictory but the source of the mismatch is almost impossible to assess given the scale mismatch between FLUXNET sites and 1◦ grid cells. However, if we would assume that both biomass and FLUXCOM GPP are consistent; this could tell us that CARDAMON only needs a higher GPP to gain higher biomass. I'm wondering if it was actually possible to constrain both biomass and GPP within the assimilation framework. Were there any prior parameters used that constrain GPP? As far as I can understand the setup of the approach, there were no data and no parameters included that would constrain GPP (apart from LAI that, however, likely only constrains the seasonality of GPP). To better understand the assimilation results, it is necessary to show maps of reduction of uncertainty of each DALEC2 model parameter. Which parameters were mostly reduced (phenology, allocation, C pools, turnover rates)?**
**Please also note that the biomass map by Thurner et al. (2014) is largely in agreement with in situ observations of forest carbon density in Russia and slightly underestimates in the USA. If CARDAMON underestimates the biomass, this implies that it would even stronger underestimate the in situ observations than the biomass map. From the results, I get the feeling that the assimilation is over-confident in the SOC data and degrades the performance with the biomass map. Hence the key question is how data uncertainties were used as weights in the assimilation? The uncertainties in SOC are much larger than in biomass (Carvalhais et al., 2014); so I expect that CARDAMON should rather fit the biomass map than the SOC map if these data uncertainties were correctly used.**

Matthias asks why the biomass relationship is not 1:1, i.e. why does CARDAMOM not force biomass or LAI to match the assimilated data from Carvalhais et al. (2014) and Myneni et al. (2002), when we do effectively get SOM to match the assimilated data.

It is correct that we do not have a direct constraint on the magnitude of GPP, except for the prior we provide (Table S1). The magnitude of GPP is not so well constrained in CARDAMOM as is its spatial and temporal variability by LAI. We rely on the Aggregated Canopy Model (ACM; Williams et al. 1997) global calibration which is based on SPA runs with some fixed leaf N content. The comparison here suggests that the ACM prior is biased compared to the Arctic, and hence the mismatch. We could assimilate the FLUXCOM GPP data, but here wanted to use FLUXCOM as an independent check. By not assimilating Jung et al., 2017 we have a clearer idea of model reliability and the calculation of transit times. We can conclude that the mismatch is due to using different models; (Jung et al. (2017) uses a range of machine-learning techniques to upscale flux data, we use a process-based ecosystem model DALEC2).

The likely answer to why CARDAMON underestimates FLUXCOM GPP perhaps is that our GPP estimate is biased low to Jung et al. (due to the calibration of ACM); this leads to less NPP which results in a low bias on LAI and biomass. The new implementation in Figure 2 shows that our error includes the 1:1 line, so we are not far out. However, we note that the CARDAMOM regional mean GPP estimate of 314 gC $m^{-2}$ $yr^{-1}$ in tundra is intermediate between the regional models' (350) and global models' (272) reported in McGuire et al. (2012) estimates, as noted in the Discussion).

We add the following text to the discussion (S4.1, L386-400):

> "In general, we found a reasonable agreement between CARDAMOM and assimilated and independent data at pan-Arctic scale. CARDAMOM retrievals of assimilated data are in good agreement with the SOC (Figure 2). The simulation of $TT_{dom}$ is weakly constrained (Table 1) - our analysis adjusts TT to match mapped stocks, hence the strong match of modelled to mapped SOC. So, independent data on $TT_{dom}$ data (e.g. $^{14}C$) is required across the pan-Arctic region to provide stronger constraint on process parameters and reduce the very broad confidence intervals of CARDAMOM analyses. The low bias in mean estimates of LAI and biomass (Figure 2) likely relates to the strong prior on photosynthesis estimates from the ACM model, which lacks a temperature acclimation for high latitudes in this implementation. However, the uncertainty in the biomass and LAI analyses spans the magnitude of the bias. So, CARDAMOM generates some parameters sets that are consistent with observations. CARDAMOM produces analyses that reproduce the pattern of LAI, GPP, biomass and SOC (Figure 2 and 3) – this demonstrates that the DALEC model structure can be calibrated to simulate the links between these variables as a function of mass balance constraints, and realistic process interactions and climate sensitivities. Biases could be reduced by assimilation of data with better resolved errors. Greater confidence in LAI and biomass data would increase the weight on their assimilation, and result in analyses closer to these data, overriding model priors by adjusting photosynthesis upwards. Further experiments can evaluate this possibility. Certainly the need for robust characterisation of error for data products is of critical importance for improved analyses"

On top of that, the mapping of uncertainty reduction was a very good idea, which we have implemented (Table 2 and Figure S2):

Table 2. Parameter uncertainty reduction in percentage ranked from least (red) to most (blue) constrained in the pan-Arctic, tundra and taiga domains. The reduction percentage is calculated based on the difference between the 90% CI prior range and the 90% CI posterior range.

| Parameter | Name | Process | Pan-Arctic | Tundra | Taiga |
|---|---|---|---|---|---|
| $MR_{litter}$ | Litter mineralization | Turnover | 3.3 | 3.6 | 2.9 |
| $TOR_{roots}$ | Root turnover | Turnover | 4.8 | 7.2 | 2.2 |
| $TOR_{wood}$ | Wood turnover | Turnover | 9.0 | 8.5 | 9.7 |
| $C_{litter}$ | Litter C stock | Stocks | 13.9 | 13.7 | 14.1 |
| $D_{rate}$ | Decomposition rate | Turnover | 18.2 | 18.6 | 17.8 |
| $f_{rau}$ | Fraction of GPP respired (Autotropic respiration) | Allocation | 30.9 | 31.7 | 30.2 |
| $L_f$ | Leaf fall duration | Phenology | 37.3 | 25.0 | 51.1 |
| LMA | Leaf mass per area | Phenology | 42.8 | 46.3 | 38.9 |
| $C_{roots}$ | Fine root C stock | Stocks | 52.4 | 72.1 | 30.3 |
| $R_l$ | Labile C release duration | Phenology | 53.1 | 52.0 | 54.4 |
| $f_{wood}$ | Fraction of NPP to wood C pool | Allocation | 65.8 | 68.1 | 63.3 |
| $F_{day}$ | Leaf fall day | Phenology | 67.0 | 51.1 | 84.8 |
| $MR_{som}$ | Soil organic matter mineralization | Turnover | 69.1 | 69.6 | 68.6 |
| $f_{labile}$ | Fraction of NPP to labile C pool | Allocation | 74.2 | 75.5 | 72.8 |
| $C_{eff}$ | Canopy efficency | Phenology | 74.7 | 75.5 | 73.7 |
| $f_{roots}$ | Fraction of NPP to roots C pool | Allocation | 75.7 | 74.7 | 76.8 |
| $B_{day}$ | Leaf onset day | Phenology | 76.2 | 67.4 | 86.1 |
| $C_{SOM}$ | Soil organic matter C stock | Stocks | 80.7 | 81.4 | 80.0 |
| L | Lifespan | Turnover | 83.4 | 76.4 | 91.4 |
| $f_{foliar}$ | Fraction of NPP to foliage C pool | Allocation | 88.0 | 88.6 | 87.4 |
| $C_{labile}$ | Labile C stock | Stocks | 92.2 | 95.3 | 88.8 |
| $C_{wood}$ | Woody C stock | Stocks | 92.6 | 90.1 | 95.5 |
| $C_{foliar}$ | Foliar C stock | Stocks | 95.2 | 96.0 | 94.3 |

[Figure]

Figure S2. Posterior distributions of parameters estimated for the CARDAMOM assimilation framework in the Pan-Arctic pan-Arctic, tundra and taiga. All y-axes have been scaled to indicate prior ranges (Table S1). Whiskers indicate 90% confidence interval (P05-P95), box indicates interquartile range (P25-P75) and center line represents the median (P50). The whisker plots represents the average (spatial variability in pan-Arctic, tundra and taiga domains) for each percentile (P05, P25, P50, P75, P95).

A new paragraph in section the implemented section 3.2 rank the uncertainty reduction (prior range vs posterior range) per parameter (S3.2, L274-285):

> "The degree to which posterior distributions were constrained from the prior distributions in each of the 17 model parameters and 6 initial stock sizes (Table S2) varied considerably depending on the parameters in question and their related processes (Table 2 and Figure S2). The 90% CI posterior range of foliar, wood, labile and SOM C stocks ($C_{foliar}$, $C_{wood}$, $C_{labile}$ and $C_{som}$) as well as parameters such as allocation to foliage ($f_{fol}$) and lifespan (L) were considerably reduced (>80% uncertainty reduction compared to priors) most likely controlled by the information on LAI, biomass and SOC constraints. Contrarily, parameters that have not been not regulated in any way in the MHMCMC algorithm, i.e. turnover processes such as litter mineralization ($MR_{litter}$), roots turnover ($TOR_{roots}$), wood turnover ($TOR_{wood}$), decomposition rates ($D_{rate}$) and initial C stock such as litter ($C_{litter}$) were found poorly constrained (<20% uncertainty reduction). Overall, the uncertainty reduction classified by processes and ranked from most to least constrained estimated a 71% reduction for C stocks, 67% reduction for C allocation, 59% for plant phenology and 31% for C turnover related parameters. Although there are not substantial differences between tundra and taiga, $C_{roots}$ was better constrained in tundra regions (42%), while leaf onset day ($B_{day}$), leaf fall day ($F_{day}$), leaf fall duration ($L_f$) were better constrained in taiga regions (>18% or more)."

**In summary, please report:**

**1. Which data uncertainties were used and how they were included in the assimilation;**

We defined uncertainties used in the likelihood function based on Bloom et al. (2016): log (1.5) for SOC and biomass, both assumed to be representative of initial conditions, and log (2) for LAI. Please see at point 3 the cost function used and how data uncertainty were included in the assimilation.

**2. How parameter prior uncertainties were included in the assimilation;**

In each 1° x 1° pixels, we applied the MHMCMC algorithm to determine the probability distribution of the optimal parameter set and initial conditions ($x_i$; Table S2) given observational constraints ($O_i$; LAI, SOC and biomass, Table S3) using the same Bayesian inference approach described in Bloom et al. (2016):

$$p(x_i|O_i) \propto p(x_i)\, p(O_i|x_i) \tag{1}$$

First, in the expression 1, $p(x_i)$ represents the prior probability distribution of each DALEC2 parameter ($x_i$) and is expressed as:

$$p(x_i) = p_{EDC}(x_i)\; e^{-0.5\left(\frac{\log(f_{auto})-\log(0.5)}{\log(1.2)}\right)^2}\; e^{-0.5\left(\frac{\log(C_{eff})-\log(17.5)}{\log(1.2)}\right)^2} \tag{2}$$

where $p_{EDC}(x_i)$ is the prior parameter probability according to the EDCs included in Table S2 and described in Bloom and Williams (2015). In addition, prior values for two parameters and their uncertainties (canopy efficiency[$C_{eff}$] and fraction of GPP respired [$f_{auto}$]) are imposed with a log-normal distribution following Bloom et al. (2016) to be consistent with the global GPP range estimated in Beer et al. (2010) and $f_{auto}$ ranges specified by DeLucia et al. (2007) respectively.

**3. How the cost function was designed and the different datasets weighted;**

$p(O|x_i)$ from expression 1 represents the likelihood of $x_i$ with respect to $O_i$, and it is calculated based on the ability of DALEC2 to reproduce (1) biomass (Carvalhais et al., 2014), (2) SOC (Hugelius et al., 2013a, Hugelius et al., 2013b), and (3) MODIS LAI (Myneni et al., 2002). Because MODIS LAI, SOC and biomass data lack specific uncertainty estimates, we used the same broad uncertainty factors as per Bloom et al. 2016: log-transformed (1.5) for SOC and biomass (i.e. ×/÷ 1.5 spans 67% of the expected error), both assumed to be representative of initial conditions, and log(2) for LAI:

$$p(O_i|x_i) = e^{-0.5\left(\frac{\log(O_{biomass})-\log(M_{biomass,0})}{\log(1.5)}\right)^2} e^{-0.5\left(\frac{\log(O_{SOC})-\log(M_{SOC,0})}{\log(1.5)}\right)^2} e^{-0.5\left(\frac{\log(O_{LAI,t})-\log(M_{LAI,t})}{\log(2)}\right)^2} \quad (3)$$

**4. What are the changes between prior and posterior parameter uncertainties.**

This question has been addressed earlier on in point 2.4, referring to new Table 2, Figure S2 and Text in S3.2, L274-285.

In order to improve a solid piece all together with the above set of requests, and because the initial text in the method part referred mostly to previous work and lacked a more specific description of CARDAMON, we decided to redraft this new section (S2.2.2, L160-202) to provide some more details and equations:

[revised manuscript text omitted]

the uncertainty represented by the 90% confidence interval (5[th] percentile to 95[th] percentile, $\left(\begin{smallmatrix} P95 \\ P05 \end{smallmatrix}\right)$).”

**2.5 Benchmarking ISIMIP with CARDAMON**
**At this point, I will not further comment on chapter 3.3. given the inconsistencies in climate forcing between CARDAMON and GVMs and given the fact that it is not clear how data uncertainties were treated in the assimilation and hence affect the CARDAMON results.**

We both addressed new analysis with ISI-MIP2a datasets (responses to point 2.3) and the new information provided based on data uncertainties (point 2.4B).

**3 Specific comments**

**1 Introduction: I suggest to slightly restructure the Introduction to make things a bit more clear. For example, several topics are mentioned twice: "transit times" (around lines 48 and 76), the available data (lines 46-59 and 83-86), and the specific features of the arctic carbon cycle (1. Paragraph, lines 55-59). In addition, the meaning of "transit times" is never explained. I suggest to:**
**- keep the first paragraph as it is,**

We slightly implemented it with few more items (S1, L35-53):

"Arctic ecosystems play a significant role in the global carbon (C) cycle (Hobbie et al., 2000; McGuire et al., 2009; McGuire et al., 2012). Slow organic matter decomposition rates due to cold and poorly drained soils in combination with cryogenic soil processes have led to an accumulation of large stocks of C stored in the soils, much of which is currently held in permafrost (Tarnocai et al., 2009). The permafrost region soil organic C (SOC) stock is more than twice the size of the atmospheric C stock; and accounts for approximately half of the global SOC stock (Hugelius et al., 2014; Jackson et al., 2017). High latitude ecosystems are experiencing a temperature increase that is nearly twice the global average (AMAP, 2017). The expected future increase of temperature (IPCC, 2013), precipitation (Bintanja and Andry, 2017), and growing season length (Aurela et al., 2004; Groendahl et al., 2007) will likely have consequences the the Arctic net C balance. As high latitudes warm, C cycle dynamics may lead to an increase of carbon dioxide ($CO_2$) emissions through ecosystem respiration ($R_{eco}$) driven by for example larger heterotrophic respiration (Commane et al., 2017; Schuur et al., 2015; Zona et al., 2016), drought stress on plant productivity (Goetz et al., 2005) and episodic disturbances (Lund et al., 2017; Mack et al., 2011). However, temperature-induced vegetation changes may counterbalance those effects by photosynthetic enhancement (Forkel et al., 2016; Graven et al., 2013; Lucht et al., 2002; Zhou et al., 2001; Zhu et al., 2016). Two examples are the increase of gross primary productivity (GPP) due to extended growing seasons, nutrient availability and $CO_2$ fertilization (Abbott et al., 2016; Myers-Smith et al., 2015; Myneni et al., 1997) and the shifts in vegetation dynamics such as shrub expansion (Myers-Smith et al., 2011). Consequently, phenology shifts may feedback on climate with unclear magnitude and sign (Anav et al., 2013; Murray-Tortarolo et al., 2013; Peñuelas et al., 2009). As a result of the significant changes that are already affecting the structure and function of Arctic ecosystems, it is critical to understand and quantify the C dynamics of the terrestrial tundra and taiga and their responses to climate change (McGuire et al., 2012)."

**- to rewrite the second paragraph: define "transit" and/or "turnover" and /or "residence" time and why it is important,**

In S1, L54-71:

"Although the land surface is estimated to offset 30% of anthropogenic emissions of $CO_2$ (Canadell et al., 2007; Le Quéré et al., 2018), the terrestrial C cycle is currently the least constrained component of the global C budget and large uncertainties remain (Bloom et al., 2016). Despite the importance of Arctic tundra and taiga biomes in the global land C cycle, our understanding of interactions between the allocation of C from net primary productivity (NPP), C stocks ($C_{stock}$), and transit times (TT), is deficient (Carvalhais et al., 2014; Friend et al., 2014; Hobbie et al., 2000). The TT is a concept that represents the time it takes for a particle of C to persist in a specific C stock and it is defined by the C stock and its outgoing flux, here addressed as $TT = C_{stock} / NPP$ at steady state. According to a recent study by Sierra et al. (2017), TT is an important diagnostic metric of the C cycle and a concept that is independent of model internal structure and theoretical assumptions for its calculation. Terms such as residence time (Bloom et al., 2016; Friend et al., 2014), turnover time (Carvalhais et al., 2016), and turnover rate (Thurner et al., 2016; TT = 1/turnover rate) are used in the literature to represent the concept of TT (Sierra et al. 2017). Studies have focused more on the spatial variability with climate of ecosystem productivity rather than for C transit time dynamics (Friend et al., 2014; Nishina et al., 2015; Thurner et al., 2016; Thurner et al., 2017). Friend et al. (2014) detailed that transit time dominates uncertainty in terrestrial vegetation responses to future climate and atmospheric $CO_2$. They found a 30% larger variation in modelled vegetation C change than response of NPP. Nishina et al. (2015) also suggested that long term C dynamics within ecosystems (vegetation turnover and soil decomposition) are more critical factors than photosynthetic processes (i.e. GPP or NPP). The respective contribution of bias from biomass and NPP to biases in transit times remains unquantified. Without an appropriate understanding of current state and dynamics of the C cycle, its feedbacks to climate change remains highly uncertain (Hobbie et al., 2000; Koven et al., 2015b)."

**- to write in the third paragraph about the available in situ and satellite-based data to assess "turnover" times and the associated uncertainties,**

In S1, L72-87:

"There are currently efforts to incorporate both in-situ and satellite-based datasets to assess C cycle retrievals and to reduce uncertainties. At local scale, the net ecosystem exchange (NEE) of $CO_2$ between the land surface and the atmosphere is usually measured using eddy covariance EC techniques (Baldocchi, 2003). International efforts have led to the creation of global networks such as FLUXNET (http://fluxnet.fluxdata.org/) and ICOS (https://www.icos-ri.eu/), to harmonise data and support the reduction of uncertainties around the C cycle and its driving mechanisms. However, upscaling field observations to estimate regional to global C budget presents important challenges due to insufficient spatial coverage of measurements and heterogeneous landscape mosaics (McGuire et al., 2012). Furthermore, harsh environmental conditions in high latitude ecosystems and their remoteness complicates

the collection of high-quality data (Grøndahl et al., 2008; Lafleur et al., 2012). Given the lack of continuous, spatially distributed in situ observations of NEE in the Arctic, it remains a challenging task to calculate with certainty whether or not the Arctic is a net C sink or a net C source, and how the net C balance will evolve in the future (Fisher et al., 2014). Over the past decade, regional to global products generated from in situ networks and/or satellite observations have improved our understanding of the terrestrial C dynamics. These range from machine-learning based upscaling of FLUXNET data (Jung et al., 2017), remotely-sensed biomass products (Carvalhais et al., 2014; Thurner et al., 2014) and the creation of a global soil database (FAO/IIASA/ISRIC/ISSCAS/JRC, 2012). However, these products tend to lack clear error estimates. Due to a reliance on interpolation and upscaling with other spatial data, it is challenging to evaluate these products for inherent biases. "

**- to write in the fourth paragraph about the inabilities and uncertainties of GVMs with respect to turnover times,**

In S1, L88-103:

"Global Vegetation Models (GVM) have been developed to determine global terrestrial C cycles and represent vegetation ecosystem processes including the structural (i.e. growth, competition, and turnover) and biogeochemical (i.e. water, carbon, and nutrients cycling) responses to climate variability (Clark et al., 2011; Fisher et al., 2014; Friend and White, 2000; Ito and Inatomi, 2012; Pavlick et al., 2013; Sitch et al., 2003; Smith et al., 2001; Woodward et al., 1995). The advantage of using process-based models to characterise C dynamics is that processes which drive ecosystem-atmosphere interactions can be simulated and reconstructed when data is scarce. However, C cycle modelling in GVMs typically relies on pre-arranged parameters retrieved from literature, prescribed plant-functional-type (PFT) and spin-up processes until the C stocks (biomass and SOC) reach their steady state. Further, inherent differences of model structure contribute more significantly to GVM uncertainties (Exbrayat et al., 2018; Nishina et al., 2014), than from differences in climate projections (Ahlström et al., 2012). Many model inter-comparison projects have demonstrated a lack of coherence in future projections of terrestrial C cycling (Ahlström et al., 2012; Friedlingstein et al., 2014). Recent studies have used simulations from the first phase of the Inter-Sectoral Impact Model Inter-comparison Project (ISI-MIP) (Warszawski et al., 2014) to evaluate the importance of key elements regulating vegetation C dynamics, but also the estimated magnitude of their associated uncertainties (Exbrayat et al., 2018; Friend et al., 2014; Nishina et al., 2014; Nishina et al., 2015; Thurner et al., 2017). An important insight is that TTs in GVMs are a key uncertain feature of the global C cycle simulation. Further, GVMs tend not to report uncertainties in their estimates of stocks and fluxes, which weakens their analytical value."

**- and finally to present model-data integration and CARDAMON as the potential "solution" in the last paragraph including the definition of your objectives.**

In S1, L104-122:

"An approach to address these issues is to integrate models and data more formally. Data assimilation quantifies how model parameters can be adjusted to estimate C stocks and fluxes consistent with multiple observations (Fox et al., 2009; Luo et al., 2009;

Williams et al., 2005). By following Bayesian methods, the uncertainty on observations weights the degree of data constraint, and the outcome is a set of acceptable parameterisations linked to likelihoods. Overall, this approach determines whether model structure, observations and forcing are (in)consistent, and thus assesses validity of model structure. By assimilating co-located climatic, ecological and biogeochemical data from remote sensing observations at a specific grid scale across landscapes and regions we can map parameter estimation and uncertainties.

Here, we use the CARbon DAta MOdel framework (CARDAMOM) (Bloom and Williams, 2015; Bloom et al., 2016; Smallman et al., 2017) to retrieve the pan-Arctic terrestrial carbon cycle at 1° resolution for the 2000-2015 period in agreement with gridded observations of LAI, biomass and SOC stocks. We compare analyses of C dynamics of Arctic tundra and taiga against (a) global products of GPP (Jung et al., 2017) and heterotrophic respiration ($R_h$) (Hashimoto et al., 2015); (b) NEE, GPP and $R_{eco}$ field observations from 8 sub- and high- Arctic sites included in the FLUXNET2015 dataset, and (c) 6 extensively used GVMs from the ISI-MIP2a comparison project (Warszawski et al., 2014). Our objectives are to (1) present and evaluate the analyses and uncertainties of the current state of the pan-Arctic terrestrial C cycling using a model-data fusion system, (2) quantify the degree of agreement between the CARDAMOM product with local to global scale sources of available data, and (3) use CARDAMOM as a benchmarking tool for the ISI-MIP2a models to provide general guidance towards GVM improvements in transit time simulations, taking the advantage that this assimilation system produces error estimates, and is constrained by observations. Finally, we suggest future work to be done in the context of advancing pan-Arctic C cycling modelling."

**Line 37: Use either "warming" or "temperature increase" but not "warming increase" because this would be an acceleration in temperature increase.**

This has been changed to temperature increase (S1, L40).

**Line 40-41: In addition to Lucht et al. and Myneni et al., you could also cite more recent related publications (Forkel et al., 2016; Graven et al., 2013; Zhu et al., 2016) [I don't request to include my paper!]**

Thanks for the recommendation, all the suggested publications have been added (S1, L46-47).

**Line 48 and lines 75-82: "transit times" – Carvalhais et al. use "turnover" time, Friend et al. "residence" time, and Thurner et al. (2016) "turnover rate". Is there a reason why you use "transit time" and why you are not using one of the other terms? Please provide a short definition of these terms or the term that you are using and how they differ.**

The "transit time" terminology instead of others arose from a paper by Sierra et al. (2017): https://onlinelibrary.wiley.com/doi/abs/10.1111/gcb.13556

The new paragraph two of the introduction (S1, L54-71) has been restructured based on your previous comment and we clarified why we used transit time and not the other terms. We also provided a short definition and how they differ.

**Line 61: "PFT or spin-up": The "or" should be replaced by "and".**

Changed accordingly (S1, L94).

**Line 74: Relevant is also the work by Thurner et al. (2016)**

Reference included now (S1, L65).

**Lines 84-85: Please provide references.**

References have been provided accordingly (S1, L82-85):

> "Over the past decade, regional to global products generated from in situ networks and/or satellite observations have improved our understanding of the terrestrial C dynamics. These range from machine-learning based upscaling of FLUXNET data (Jung et al., 2017), remotely-sensed biomass products (Carvalhais et al., 2014; Thurner et al., 2014) and the creation of a global soil database (FAO/IIASA/ISRIC/ISSCAS/JRC, 2012)."

**Line 87: A reference to a general overview paper on model-data integration might be useful.**

An updated text, including extra information and general references regarding model-data integration, have been implemented (S1, L104-110):

> "An approach to address these issues is to integrate models and data more formally. Data assimilation quantifies how model parameters can be adjusted to estimate C stocks and fluxes consistent with multiple observations (Fox et al., 2009; Luo et al., 2009; Williams et al., 2005). By following Bayesian methods, the uncertainty on observations weights the degree of data constraint, and the outcome is a set of acceptable parameterisations linked to likelihoods. Overall, this approach determines whether model structure, observations and forcing are (in)consistent, and thus assesses validity of model structure. By assimilating co-located climatic, ecological and biogeochemical data from remote sensing observations at a specific grid scale across landscapes and regions we can map parameter estimation and uncertainties."

**Line 95-96: It is not clear to me how your analysis will provide further insight into GVMs that goes beyond the work of Friend et al. (2014), Carvalhais et al. (2014), and Thurner et al. (2017). Please make clear what kind of additional knowledge you are expecting from your analysis on the problems of GVMs.**

Our system is constrained by both model structure and varied observations, using the EDCs to make sure that processes are realistic and ecologically viable. The previous studies have largely focused either on model analyses (Friend et al) or on combining data products (Carvalhais, Thurner et al) to generate TT. Here we use CARDAMOM to combine the information contained in model structure with independent observational data to produce a consistent, robust analysis. Our approach avoids using PFTs and steady state assumptions (typical in the GVMs). By including a mass balance constraint on the C cycle we evaluate consistency among different data sets (e.g. SOM, biomass, LAI, climate) using our model structure to generate TT. Thus the novelty of this study is a data-constrained descriptions of C cycling for numerous live

and dead pools, and their transit times, with errors at pixel scale. We use the complete assessments to assess better GVMs, to identify how to produce more constrained forecasts of this sensitive region.

We refined the aim (3) in the following (S1, L116-121):

> "Our objectives are to (1) present and evaluate the analyses and uncertainties of the current state of the pan-Arctic terrestrial C cycling using a model-data fusion system, (2) quantify the degree of agreement between the CARDAMOM product with local to global scale sources of available data, and (3) use CARDAMOM as a benchmarking tool for the ISI-MIP2a models to provide general guidance towards GVM improvements in transit time simulation, taking the advantage that this assimilation system produces error estimates, and is constrained by observations."

**Lines 106-107: Please define which classes you used to separate forest and non- forest.**

This new line has been included in the text to better refer to the different classes used to separate tundra and taiga domains (S2.1, L129-130):

> "A complete description of the classes included in each domain can be found in Figure S1 and caption."

Figure S1 caption states now:

> "Figure S1. Spatial domain defined by the Northern Circumpolar Soil Carbon Database version 2 (NCSCDv2) region. The tundra- taiga regions were separated based on the presence-absence of forested areas using the GlobCover map (http://due.esrin.esa.int/page_globcover.php). Forested areas (taiga) included: closed to open broadleaved evergreen or semi-deciduous forest (>5m), closed (>40%) broadleaved deciduous forest (>5m), open (15-40%) broadleaved deciduous forest/woodland (>5m), closed (>40%) needleleaved evergreen forest (>5m), open (15-40%) needleleaved deciduous or evergreen forest (>5m) and closed to open (>15%) mixed broadleaved and needleleaved forest (>5m). Non-forested areas (tundra) included the rest of classes: mosaic forest or shrubland (50-70%) / grassland (20-50%),mosaic grassland (50-70%) / forest or shrubland (20-50%), closed to open (>15%) (broadleaved or needleleaved, evergreen or deciduous) shrubland, closed to open (>15%) herbaceous vegetation (grassland, savannas or lichens/mosses), sparse (<15%) vegetation, closed to open (>15%) broadleaved forest regularly flooded (semi-permanently or temporary), closed (>40%) broadleaved forest or shrubland permanently flooded, and closed to open (>15%) grassland or woody vegetation on regularly flooded or waterlogged, post-flooding or irrigated croplands (or aquatic), rainfed croplands, mosaic cropland (50-70%) / vegetation (grassland/shrubland/forest) (20-50%), mosaic vegetation (grassland/shrubland/forest) (50-70%) / cropland (20-50%), bare areas and permanent snow and ice. On top of that, latitudes lower than 52°N within the tundra domain were neglected to focus on higher latitudes."

**Section 2.2: The description of CARDAMON refers mostly to previous work. However, to understand _this_ paper, I suggest to provide some more details or equations with respect to the following questions:**

**- LAI, biomass, and SOC are used as data sets in a cost function for parameter estimation and not as forcing data. Is this correct?**

This is correct, we now better describe this in the methods (see our latter comment in point 2.4 (major comments) including the new section 2.2.2 from the main text). On top of this, we included a new Table S3 describing which datasets were used as forcing, data constraints and independent validation.

**- What is the cost function? How are the differences in the number of data points weighted (LAI is a time series, SOC and biomass only single values per grid cell)?**

The cost function was already described before in point 2.4 (major comments) including the new section 2.2.2 from the main text.
Regarding the second question, basically Equations 2 and 3 answer this: we did not weight for differences in the number of data points.

**- Why is the MHMCMC algorithm used three times? Does it not explore the full parameter space if it is applied only once? Or are there difference in initial values?**

The process is repeated to make sure independent chain converge to the same posterior distribution. This is a standard procedure in MCMC analyses, see for example Vrugt et al. (2003) and Kuczera and Parent (1998). In CARDAMOM we only keep parameters from the chains which have converged.

Vrugt, J. A., H. V. Gupta, W. Bouten, and S. Sorooshian (2003), A Shuffled Complex Evolution Metropolis algorithm for optimization and uncertainty assessment of hydrologic model parameters, Water Resour. Res., 39, 1201, doi: 10.1029/2002WR001642, 8.

Kuczera, G., and Parent, E.: Monte Carlo assessment of parameter uncertainty in conceptual catchment models: the Metropolis algorithm, Journal of Hydrology, 211, 69-85, https://doi.org/10.1016/S0022-1694(98)00198-X, 1998.

**- Can you make a conceptual figure that shows which data sets go into the assimilation and which are only used as independent evaluation data?**

We implemented a new Table S3 (S2.2.2, L171) instead, since we also implemented a conceptual Figure 1 (S3.1, L235) representing key C fluxes, stocks and transit times which is already complex enough:

Table S3. Forcing dataset, observational constraints and independent validation datasets used in this study's experimental design.

| Dataset | Source | Forcing | Constraint | Validation |
|---|---|---|---|---|
| **ERA-Interim** | Dee et al. (2011) | X | | |
| **MODIS - LAI** | Myneni et al. (2002) | | X | |
| **NCSCD - SOC** | Hugelius et al. (2003) | | X | |
| **Biomass** | Carvalhais et al. (2014) | | X | |
| **NEE, GPP, $R_{eco}$** | FLUXNET2015 | | | X |
| **GPP** | Jung et al. (2017) | | | X |
| **$R_h$** | Hashimoto et al. (2015) | | | X |

**Line 147: What is the difference between "photosynthetic" and "vegetation" C stocks? Is photosynthesis not vegetation?**

Text now has been updated to explicitly define "photosynthetic" and "vegetation" C stocks (S2.2.1, L150-152):

> "For practical purposes we aggregated the different C stocks into photosynthetic ($C_{photo}$; leaf and labile), vegetation ($C_{veg}$; leaf, labile, wood and roots), soil ($C_{dom}$; litter and SOM) and total ($C_{tot} = C_{photo} + C_{veg} + C_{dom}$) C stocks."

**Line 159: Did you directly compare the 1◦ grid cell with the FLUXNET site? If yes, how are the FLUXNET sites representative for the 1◦ grid cell? If no, did you run CARDAMON with the site meteorological data? Add: Only from the discussion (lines 318-238), I now learn that you did a grid cell to point comparison. This should be already mentioned in the methods and be recalled at the appropriate place in the results section.**

We included a whole paragraph discussion on how FLUXNET2015 sites are representative for the 1◦ grid cell in the S4.1. However, the referee is right that it was only mentioned in the discussion part, so we implemented the following text in S2.3, L219-221:

> "We performed a point-to-grid cell comparison to assess the degree of agreement between each flux magnitude and seasonality calculating the statistics of linear fit (slope, intercept, $R^2$, RMSE, and bias) per flux and site between CARDAMOM and FLUXNET2015 datasets."

**Lines 166-168: This is a repetition from lines 123-125. Please merge the two sentences.**

We have merged the two sentences as requested (S2.2.1, L152-155):

> "The Net Ecosystem Exchange (NEE) is calculated as the difference between GPP and the sum of the respiration fluxes ($R_{eco} = R_a + R_h$), while Net Primary Productivity (NPP) is the difference between GPP and $R_a$. Only NEE follows the standard micrometeorological sign convection presenting the uptake of C as negative (sink), and the release of C as positive (source); both GPP and $R_{eco}$ are reported as positive fluxes."

**Line 171: TT_veg was already mentioned at line 148. I suggest to remove both occurrence of TT_veg and to already define TT_veg in the new second paragraph of the introduction.**

We followed this request, and we restructured the introduction to define transit times (TT) in the second paragraph as you suggested above.

**Line 171: At the end CARDAMON is also just only a GVM but with grid cell-specific parameters. I don't see how CARDAMON then serve as a benchmark for the other GVMs. Would it be not enough to directly benchmark the GVMs against the reference data? You should try to better motivate already in the Introduction why you can use CARDAMON as a benchmark for GVMs.**

We implemented already aim (3) in the introduction, see above. We use CARDAMOM because it provides estimates of TT with errors that are consistent with theories of C cycling and with multiple observational data. CARDAMOM avoids the significant problem of spin-up associated with GVMs.

Moreover, we implemented the following text in the discussion (S4.2, L420-431):

> "An ideal benchmarking tool for GVMs would compare model state variables and fluxes against multiple, independent, unbiased, error-characterised measurements collected repeatedly at the same temporal and spatial resolution. Of course direct measurements of key C cycle variables like these are not available. Even at FLUXNET sites GPP and $R_{eco}$ must be inferred, and NEE data often gap-filled. Satellite data can provide continuous fields, but do not directly measure ecological variables like biomass or LAI, so calibrated models are required to generate ecological products. Atmospheric conditions can introduce biases and data gaps into optical data that are poorly quantified. Upscaling of FLUXNET data requires other spatial data, e.g. MODIS LAI, which challenges the characterisation of error and generates complex hybrid products. We suggest that CARDAMOM provides some of the requirements of the ideal benchmark system – an error-characterised, complete analysis of the C cycle that is based on a range of observational products. CARDAMOM includes its own C cycle model; this has the advantage of evaluating the observational data for consistency (e.g. with mass balance), propagating error across the C cycle, and generating internal model variables such as TT. Further the model is of low complexity and independent of the benchmarked models.
> "

**Line 173: LPJmL (capital L). Please indicate which version of LPJmL was used. Is it the most recent version (LPJmL4) (Schaphoff et al., 2018a)? LPJmL4 includes also a new permafrost module (Schaphoff et al., 2013) and a data-constrained phenology module (Forkel et al., 2014) and hence better reproduces boreal and arctic carbon stocks and carbon cycling than the previous versions (Forkel et al., 2016; Schaphoff et al., 2018b).**

Thanks for the correction, we now refer to the model as "LPJmL". LPJmL4 seems a very good candidate to be compared against CARDAMOM. However, the LPJmL version (version name is missing) available in the ISI-MIP2a database only includes the permafrost module (Schaphoff et al., 2013) you mentioned with runs dated from 2016: https://www.isimip.org/impactmodels/details/81/#tab_isimip2a

**Lines 184-194: I got really confused by this paragraph because initially I got the impression that you "jump" across all results without explanation. Please make clear that this paragraph is a summary of all results by either using a heading or a suitable topic sentence.**

Following Matthias comment, together with the one from **REF#2**, we decided to drop the summary paragraph in the results section. We believe this also contributes to a lighter text.

**Line 202: Which of the used data sets constrain the separation between GPP and NPP?**

We only have a prior of Ra:GPP = 0.5

**Line 216: Do you find spatial pattern in TT_photo that would resample the distribution of evergreen and deciduous trees?**

In this initial study using CARDAMOM over the pan-Arctic region we decided to partition results into tundra (grass/shrub dominated) and taiga (forest dominated) areas. Given the fact that the text is already quite dense, we believe further analysis such as the one you proposed (partition evergreen and deciduous trees) would add an extra layer of complexity to this manuscript. However, we find this suggestion quite relevant and attractive, and so it would be wise to have it addressed in coming papers.

**Line 219: "Interestingly" – Please tell me why this is "interestingly".**

We removed the wording "Interestingly", plus we rephrased the sentence to (S3.1, LX-X):

> "CARDAMOM calculated 62% longer $TT_{dom}$ in tundra compared to taiga, likely linked to lower temperatures, but uncertainties are large due to the limitations of data constraints."

**Line 257: "as noted " – Please check.**

We rephrased the sentence to (S3.3, LX-X):

> "This mismatch is important in the context of the FLUXCOM GPP upscaling, 50% higher than CARDAMOM."

**Lines 330-331: This sentence should be merged with the numbers given at lines 349- 352.**

We restructured the section 4.1. to improve readability and clarity (L347-349):

> "CARDAMOM retrievals are consistent with outcomes from relevant papers such as the (I) C flux observations and model estimates reported in McGuire et al. (2012); (II) C stocks and transit times described by Carvalhais et al. (2014), and (III) NPP, C stocks and turnover rates stated in Thurner et al. (2017):"

This paragraph is now followed by a list of 3 numbers (I, II, III) focusing on each of the previous relevant papers addressed above.

[revised manuscript text omitted]

Wania, R., Ross, I., and Prentice, I. C.: Integrating peatlands and permafrost into a dynamic global vegetation model: 2. Evaluation and sensitivity of vegetation and carbon cycle processes, Global Biogeochemical Cycles, 23, doi:10.1029/2008GB003413, 2009b.
* * *
**Interactive comment on Earth Syst. Dynam. Discuss., https://doi.org/10.5194/esd-2018-19, 2018.**
* * *
* * *
**In this paper the authors take the CARDOMOM + DALEC Bayesian calibration system an apply it specifically to the arctic using a number of regional-scale data products. Once the model is fit to data, it is then used to assess carbon pools and benchmark global vegetation models. The scale and scope of the analysis is quite impressive – building up their system to this point was clearly a lot of work and the attempt to synthesize multiple data constraints at a regional scale is really important, especially for a highly influential and understudied region like the arctic.**

We are thankful for the reviewer's insightful and thorough comments. We believe this review has substantially improved the manuscript, spotting incomplete areas and highlighting convincing areas that perhaps we could further emphasize. We have carefully considered the reviewer's remarks and clarified our manuscript accordingly.

**That said, I do have a few high level concerns about what the authors have done. The easiest of these to address is that the details of what was actually done was insufficient and teasing out important high-level facets of CARDOMOM are left to the reader tracking down earlier papers. Particularly important is to clarify whether DALEC is calibrated independently for every pixel, in some sort of spatially correlated manner, or with a single parameterization for the two PFTs across the whole region. My recollection from earlier papers made me think the first (independent fits), but in reading the results it is hard to distinguish parameter uncertainty from parameter spatial heterogeneity. The authors need to be more explicit about this. Likewise, the authors need to be more clear about whether this is really a data assimilation system, or if it's just a calibration system. This matters because in DA (e.g. EnKF) the analysis provides a formal synthesis of observations and process understanding, but in a calibration system your estimated states are ultimately just a forward model run. To me, it feels like the authors are treating a forward model run as if it were a reanalysis product. If this is true what the authors did is still valuable but they should be more open about this and the limitations of this approach.**

We fully understand this point, agreeing with the **REF#1** (one of the major critic comments). DALEC2 is independently calibrated in each pixel. We tried to do a better job in showing the within pixel parameter error. In order to do so, we added to the main text, figures, tables and SI:

1) An implemented section 2.2 (The CARbon Data Model framework) in methods (L136-141)

> "Here we use the CARbon DAta MOdel framework (CARDAMOM; Bloom et al., 2016) (list of acronyms can be found in Table S1) to retrieve terrestrial C cycle dynamics, including explicit confidence intervals, in the pan-Arctic region.

CARDAMOM consist of two key components: (1) an ecosystem model, the Data Assimilation Linked Ecosystem Carbon version 2 (DALEC2) (Bloom and Williams, 2015; Williams et al., 2005), constrained by observations and (2) a data-assimilation system (Bloom et al., 2016). This framework reconciles observational datasets as part of a representation of the terrestrial C cycle in agreement with ecological theory.

2) An implemented section 2.2.1 (DALEC2) in methods (L143-158)

"DALEC2 ecosystem model simulates land-atmosphere C fluxes and the evolution of six C stocks (foliage, labile, wood, roots, soil organic matter (SOM) and surface litter) and corresponding fluxes. DALEC2 includes 17 parameters controlling the processes of plant phenology, photosynthesis, allocation of primary production to respiration and vegetation carbon stocks, plant and organic matter turnover rates, all established within specific prior ranges based on ecologically viable limits (Table S2). DALEC2 simulates canopy-level GPP via the Aggregated Canopy Model (ACM; Williams et al., 1997) and its allocation to the four plant stocks (foliage, labile, wood and roots) and autotrophic respiration ($R_a$) as time-invariant fraction of GPP. Plant C decays into litter and soil stocks where microbial decomposition generates heterotrophic respiration ($R_h$). Turnover of litter and soil stocks is simulated using temperature dependent first-order kinetics. For practical purposes we aggregated the different C stocks into photosynthetic ($C_{photo}$; leaf and labile), vegetation ($C_{veg}$; leaf, labile, wood and roots), soil ($C_{dom}$; litter and SOM) and total ($C_{tot} = C_{photo} + C_{veg} + C_{dom}$) C stocks. The Net Ecosystem Exchange (NEE) is calculated as the difference between GPP and the sum of the respiration fluxes ($R_{eco} = R_a + R_h$), while Net Primary Productivity (NPP) is the difference between GPP and $R_a$. Only NEE follows the standard micrometeorological sign convection presenting the uptake of C as negative (sink), and the release of C as positive (source); both GPP and $R_{eco}$ are reported as positive fluxes. In this study, we addressed C turnover rates and decomposition processes as their inverse rates, this is the C transit time ($TT_{photo}$, $TT_{veg}$ and $TT_{dom}$), represented as the ratio between each C stock and the NPP allocated into that stock."

3) An implemented section 2.2.2 (Data-assimilation) in methods (L160-202) to be more explicit about the experimental design including equations as noted above in responding to the **REF#1**,

4) A new section in the results section regarding Data assimilation and uncertainty reduction (S3.2, L274-285) together with a new Table 2 and Figure S2 as noted above in responding to the **REF#1**.

**Second, in light of the earlier point about reanalysis vs forward simulation, I am really uncomfortable about the author's use of their model as benchmark for other models. This is particularly true given the non-trivial biases in some of the verification (biomass) and validation (GPP, Rh) analyses and the lack of independent validation of a number of the other processes in the model (e.g. turnover). I think this manuscript could stand alone without the GVM component.**

In this manuscript we want to recognise that the uncertainties are large – uncertainties are rarely if ever calculated and presented, and this means analyses have been overconfident. We agree that the full descriptions of C cycling with errors are novel. We aim to go further to use the

complete assessments of C cycling to assess better GVM, to allow more constrained forecasts for this region.

As noted above in responding to the **REF#1**, CARDAMOM reconciles observational datasets as part of a representation of the terrestrial C cycle in agreement with ecological theory – CARDAMOM provides observationally-constrained estimates of C dynamics. A key reason to use CARDAMOM as a benchmark for GVMs is that CARDAMOM produces parameter likelihoods for each pixel based on data, and it does not assume PFTs or steady states, hence it is much more strongly data constrained that a GVM. CARDAMOM takes data and the model to produce parameter maps, whereas GVMS take PFT maps of parameters to produce flux/stock outputs. For this reason, we used our data assimilation system as a benchmarking tool for six GVMs.

Our analyses of the C cycle are independent of the GVMs. As the referee notes there are biases in the verification and validation. However, because our approach takes into account the error in assimilated data, we produce uncertainties on our C cycle estimates. Our analytical uncertainties on e.g. biomass include the data within their confidence intervals, so the potential for this bias is explicit within our outputs, and propagated into e.g. TT estimates. Independent estimates of TT do not exist – one purpose of this study is to provide robust estimates of TT from CARDAMOM to compare with GVMs. The novelty here is that we can locate which GVMs and for which regions the TT estimates of models are outside CARDAMOM confidence intervals (stippling in Figure 6).

**Third, I'm really concerned about how the authors assimilate these derived data products. There's not really any discussion of how the observation errors in the data and process error in the model are treated. There's not any discussion of how the authors handled the non-independence of spatial pixels in these data products. Indeed the authors seem to treat data products as if they are truly data, which likely results in an overestimation of the true information content in the data. For example, if I have 10 observations I can Krige a map that has 10k grid cells, but my true sample size remains 10 not 10k and any data assimilation system needs to reflect that.**

As noted above in responding to the **REF#1**, errors from the observational products are not available from the data providers but we still defined them in the likelihood function based on Bloom et al. (2016). Therefore, there were errors attached to each observation, e.g. MODIS LAI, biomass for each pixel. We assumed independent data in each grid cell for LAI and biomass deriving from satellite products. We recognise that the algorithms used to produce e.g. LAI may include spatial assumptions that generate correlated biases. We acknowledge that the soil C data are interpolated using machine learning approaches. Our response is to include a large error on the data for each pixel in the absence of a detailed, spatially defined error.

As we pointed earlier (**REF#1 response**), we included a full new description about how these data were assimilated and which uncertainties were considered in each dataset (S2.2.2, L178-202). More specifically in this new section we mention that "Each pixel is treated independently without assuming a prior land cover type and we assume no spatial correlation between uncertainties in all pixels."

**Detailed Comments:**

**L126: 1) Is calibration really data assimilation? 2) inclusion of process error?**

The correct term for CARDAMOM is data assimilation or model-data fusion. Again, new sections 2.2.1 (DALEC2) and 2.2.2 (Data-assimilation) were implemented in the methods section to better address parameter uncertainty reduction. Data assimilation explicitly propagates data uncertainty into model calibration.

**L135: What is the actual underlying sample size? Derived data products can massively conflate the actual information content. Errors in these data products are hugely autocorrelated and that observation uncertainty is not captured correctly in these products. Also, many of these constraints are not data (GPP, LAI, biomass) but just different models.**

We have updated section 2.2.2 with the following text (L169-177):

> "Observational constraints include monthly time series of Leaf Area Index (LAI) from the MOD15A2 product (Myneni et al., 2002), estimates of vegetation biomass and soil organic carbon content (Table S3). We aggregated ~130,000 1-km resolution MODIS LAI data monthly within each 1x1 degree pixel.  We aggregated biomass data at 0.5° resolution from Carvalhais et al. (2014) to 1° resolution. These are based on remotely-sensed forest biomass and upscaled GPP based on data driven estimates (Jung et al., 2011) covering the pan-Arctic domain. We used the NCSCD spatial explicit product (Hugelius et al., 2013a; Hugelius et al., 2013b)  which was generated from 1778 soil sample locations interpolated to a 1° grid. There is significant uncertainty for these data, due to the models involved in generating LAI and biomass, and the interpolation process for soils. Hence we apply broad confidence intervals commensurate with this uncertainty (Equation 3)."

**L144: 500 samples per chain? That's way too small. Also, what's the effective sample size after accounting for autocorrelation? I'd recommend the authors shoot for an effective sample size around ~5000 total, which likely will require a much larger total number of samples given their reliance on Metropolis-Hastings. Not stated explicitly whether this is one global parameter set or one per grid cell? My memory from Bloom et al 2016 is the latter.**

We apologise that we did not explain our process more clearly; we ran three chains accumulating $10^7$ accepted parameters We ensured convergence in at least 2 chains and gathered 500 sampled parameter sets from each chain. This sampling was used to generate a distribution of model analyses that we then analysed and plotted, see section 2.2.2.

**L146: A 90% CI is typical. Reason for not 95% norm?**

A 90% CI is a widely used in the literature so we decided to assess this specific uncertainty range in our analysis.

**L154: This isn't independent of the calibration product**

Indeed, this is independent of the calibration product. We do not assimilate GPP, but LAI. The new text in Section 2.2.2 hopefully clarifies this, together with the new Table S3.

**L164: should really include the 95% CI in addition to the interquartile**

We changed from 50% CI to 90% CI. Figure 4 has been updated accordingly, and uncertainties represent the 25th and 50th percentiles (darker shade) and the 5th and 95th percentiles (lighter shade) of both field observations and the CARDAMOM framework.

**L169: You can't compare a complex model against a (mis)calibrated simple model and call it a benchmark. Especially true if you're looking at the marginal distributions of indirectly inferred latent variables.**

This point is arguable – what do we mean by a benchmark? As the referee has acknowledged, all global/gridded data products involve some degree of modelling (e.g. FLUXCOM, MODIS LAI), and hence are not direct measurements. Thus, a comparison against a data product is open to this same criticism. In response we make the following addition to the text in the discussion (S4.2, L420-431):

> "An ideal benchmarking tool for GVMs would compare model state variables and fluxes against multiple, independent, unbiased, error-characterised measurements collected repeatedly at the same temporal and spatial resolution. Of course direct measurements of key C cycle variables like these are not available. Even at FLUXNET sites GPP and $R_{eco}$ must be inferred, and NEE data often gap-filled. Satellite data can provide continuous fields, but do not directly measure ecological variables like biomass or LAI, so calibrated models are required to generate ecological products. Atmospheric conditions can introduce biases and data gaps into optical data that are poorly quantified. Upscaling of FLUXNET data requires other spatial data, e.g. MODIS LAI, which challenges the characterisation of error and generates complex hybrid products. We suggest that CARDAMOM provides some of the requirements of the ideal benchmark system – an error-characterised, complete analysis of the C cycle that is based on a range of observational products. CARDAMOM includes its own C cycle model; this has the advantage of evaluating the observational data for consistency (e.g. with mass balance), propagating error across the C cycle, and generating internal model variables such as TT. Further the model is of low complexity and independent of the benchmarked models"

**L177: If looking at the historical period, why weren't models run under reanalysis meteorology rather than GCMs?**

We have adjusted our analysis in response to this comment. A complete new GVM exercise was performed since both **REF#1** and **REF#2** raised the same criticism. As we pointed earlier (**REF#1 response**), we used the ISI-MIP2a simulations instead of the original ISI-MIP1. This new implementation presents many advantages like the full overlap of the studied period (2001-2010) and more similar climate drivers. Earlier we also specified that we changed the method's part related to ISI-MIP2a (S2.4, L223-232), the results section (S3.4, L311-338) and discussion (S4.2, L420-446).

**L184: Drop this whole paragraph – it's a bit confusing to give a summary of the results before presenting the results without making it clear that this is a summary of highlights.**

**Right now it just feel like you're going though the results really quickly without much explanation.**
This summarizing has been deleted accordingly since **REF#1** suggested that too.

**L187: A 28% bias against the data that the model was calibrated to seems like a pretty big problem.**

As discussed for **REF#1**, this bias likely arises from a lower predicted rate of photosynthesis in Arctic systems in the ACM photosynthesis model. The newly implemented Figure 2 shows that the 90% CI includes the 1:1 line. Thus, our analysis is not inconsistent with the data. We should note that the data may also be biased. Further, at regional scale our GPP estimates lie within the ranges of GVMs and regional models.

**L190: "This mismatch is important in the context of FLUXCOM, as noted" what do you mean "as noted" you never noted anything**

We rephrased the sentence to (S3.3, L302-303):

"This mismatch is important in the context of the FLUXCOM GPP upscaling, 50% higher than CARDAMOM GPP."

**L203: "and marginally varied across tundra" I don't understand what you mean here**

Wrong wording, we meant (S3.1, L242):

"and marginally varied between tundra $0.50\binom{0.54}{0.46}$ and taiga $0.52\binom{0.56}{0.46}$."

**L209: Distinguish tundra and taiga. These numbers don't seem plausible for tundra**

As requested, we implemented to (S3.1, L247-249):

"Among the living C stocks, 93% of the C (88% in tundra and 90% in taiga) is allocated to the structural stocks (wood and roots; $1.4\binom{5.6}{0.4}$ kg C m$^{-2}$) compared to 7% (12% in tundra and 10% in taiga) to the photosynthetic stock (leaves and labile; $0.1\binom{0.2}{0.1}$ kg C m$^{-2}$)."

**L211: That the tundra numbers are so close to the taiga numbers doesn't seem correct. How well do these numbers validate against direct field data (not derived data products)?**

The relatively small differences between tundra and taiga are a result of the coarse scale of the data and the relatively low values of biomass in the assimilated product (Figure 2). It is not appropriate to evaluate biomass products against field data due to the scale mismatch. As the other referee has noted:
"**…the biomass map by Thurner et al. (2014) is largely in agreement with in situ observations of forest carbon density in Russia and slightly underestimates in the USA.**"

**L216: A transit time of 4.3 years in the woody tissues of a spruce tree seems really fast give their lifespan. How does this compare to field data (e.g. isotopes)**

The 4.3 years does not differentiate between branch size of woody components. This value represents the median's of all woody material in the entire pan-Arctic region, including twigs, branches, stems and coarse roots.

**L217: The CI on the SOM is really large (essentially 10-1000 years). Is this just the prior?**

Yes, due to the lack of data on SOM turnover, the TT for SOM remains poorly constrained.

**L228: This results needs additional explanation with regards to what this test statistic applies to. You calibrated a mechanistic model via MCMC, this isn't a t-test. What specifically changed that much?**

This entire paragraph has been dropped. Please check answer to **REF#2** in L295 (three points later).
This change arose mainly due to the fact that the presented sensitivity analysis did not included a version without a soil C constraint. The fair analogy should be done by removing soil C as a constraint, or by using other biomass product as constraint. Otherwise, the information content in both data-streams cannot be comparable by this exercise that eliminates one data stream and compares two of them.

**L234: What do you mean priors, isn't this the data?**

Confusing wording here, we implemented the text to (S3.2, LX-X):

> "The CARDAMOM framework generated an analysis broadly consistent with the combination of SOC, biomass and LAI in each grid cell (Figure 2), and the errors assigned to these data products (Figure 2)."

**L257: This is almost the exact same sentence as L190**

Since the initial summarizing paragraph in the results section has been dropped following **REF#1** and **REF#2** suggestions, this sentence now can be kept, but rephrased to (S3.3, L303-304):

> "This mismatch is important in the context of the FLUXCOM GPP upscaling, 50% higher than CARDAMOM GPP."

**L295: Is this statement that CARDOMOM is more sensitive biomass than soil C really fair? In one case you're comparing whether a data constraint is included at all, while in the other your comparing different derived data products, which are likely relying on similar underlying raw data. I think for this to be fair you would want to include a version where you don't have any soil C constraint.**

The referee is completely right, the presented sensitivity analysis did not include a version without a soil C constraint. The fair analogy should be done by removing soil C as a constraint, or by using other biomass product as constraint. Otherwise, the information content in both data-streams cannot be comparable by this exercise that eliminates one data stream and compares two of them. Therefore, and because the manuscript is complex enough, this entire paragraph has been dropped.

**L308: There's a 28% bias in biomass, how is that "good agreement". The Discussion seems to be missing the critical point that if a model is faced with multiple constraints and can't reconcile them then there's either inconsistencies in the data, structural errors in the model, or both. And why is there no comparison to LAI and GPP constraints? Also, there seems to be no discussion of how observations error in the data are derived/treated and how you're handling the process error in the model (is this a fit parameter or just ignored).**

As we pointed earlier (**REF#1 response**), the magnitude of biomass is not so well constrained in CARDAMOM (although the 90% CI include the 1:1 line, see new Figure 2) as is its spatial and temporal variability (well constrained by LAI). We do not have a direct constraint on the magnitude of GPP, except for the prior we provide (Table S1). We rely on the Aggregated Canopy Model (ACM; Williams et al. 1997) calibration which is based on SPA runs with some fixed leaf N content. Perhaps ACM is biased compared to the Arctic, lacking a temperature acclimation, and hence the mismatch. We now include reference to this issue in the text, and include comparison to GPP explicitly earlier in the results and discussion. We note the importance of data error in the resulting biases in both calibration and validation, and the need for more robust error characterisation of data products.

We note that the Calvalhais et al. (2014) biomass data relies on various assumptions in areas of low tree cover, covering much of the high latitudes. In these areas GPP data from Jung et al. are used for calculating herbaceous biomass. Thus, there is a dependence between these data products. We need fully independent, error characterised data to make the next steps forward. Hence the broad errors set on data products in this analysis.

A new comparison set of assimilated LAI has been implemented in Figure 2 as was requested by **REF#2**.

We have adjusted the text in 3.2 to read (L264-273):

> "The CARDAMOM framework generated an analysis broadly consistent with the combination of SOC, biomass and LAI in each grid cell (Figure 2), and the errors assigned to these data products. The agreement for the SOC dataset by Hugelius et al. (2013a) is a 1:1 relationship ($R^2 = 1.0$; RMSE = 0.97 kg C m$^{-2}$), reflecting a straightforward model parameterisation. The biomass product from Carvalhais et al. (2014), was well correlated ($R^2 = 0.97$; RMSE = 0.46 kg C m$^{-2}$), but CARDAMOM was consistently biased ~28% low. MODIS LAI data were also well correlated, but ~28% higher than CARDAMOM analyses. These biases likely arise due to a low estimate in the photosynthesis model (ACM) used in CARDAMOM (Figure 3) which propagates through the C cycle. CARDAMOM balances uncertainty in data products and the models (ACM photosynthesis model and DALEC2), to generate a weighted analysis, typical of Bayesian approaches. The CARDAMOM analysis 90% confidence interval (CI) includes the 1:1 line for biomass and LAI (Figure 2), indicating that the likelihoods on C cycle analyses include the expected value of the observations."

The capacity of DALEC2 to reproduce the patterns in LAI, SOM, biomass and GPP is a strong indicator that the model structure is valid. What we notice is bias on the estimation of some of these parameters across the region. We relate this bias to a strong model prior on photosynthesis and large uncertainty on the LAI and biomass data. An increase in data confidence would resolve this problem.

In the discussion 4.1 (L386-400) we now state:

> "In general, we found a reasonable agreement between CARDAMOM and assimilated and independent data at pan-Arctic scale. CARDAMOM retrievals of assimilated data are in good agreement with the SOC (Figure 2). The simulation of $TT_{dom}$ is weakly constrained (Table 1) - our analysis adjusts TT to match mapped stocks, hence the strong match of modelled to mapped SOC. So, independent data on $TT_{dom}$ data (e.g. $^{14}C$) is required across the pan-Arctic region to provide stronger constraint on process parameters and reduce the very broad confidence intervals of CARDAMOM analyses. The low bias in mean estimates of LAI and biomass (Figure 2) relates to the strong prior on photosynthesis estimation from the ACM model, which lacks a temperature acclimation for high latitudes in this implementation. However, the uncertainty in the biomass and LAI analyses spans the magnitude of the bias. So, CARDAMOM generates some parameters sets that are consistent with observations. CARDAMOM produces analyses that reproduce the pattern of LAI, GPP, biomass and SOC (Figure 2 and 3) – this demonstrates that the DALEC2 model structure can be calibrated to simulate the links between these variables as a function of mass balance constraints, and realistic process interactions and climate sensitivities. Biases could be reduced by assimilation of data with smaller errors. Greater confidence in LAI and biomass data would increase the weight on their assimilation, and result in analyses closer to these data, overriding model priors by adjusting photosynthesis upwards. Further experiments can evaluate this sensitivity. Certainly, the need for robust characterisation of error for data products is of critical importance for improved analyses."

**L312: I haven't looked into the details of the Jung 2011 product vs the Jung 2017 product, but I'm skeptical that these are independent. Would be good to state more explicitly what each product is upscaling to generate GPP (FLUXNET? SIF?). If they're both FLUXNET-based then they're not independent if they're just applying different algorithms to upscale the same underlying data.**

We only used Jung et al 2017's GPP product as independent validation, thus we do not see why we should state more explicitly what each product is upscaling to generate GPP. However, it seems that there is a misunderstanding - **REF#2** believes that GPP has been assimilated in CARDAMOM. We hope now that the new section 2.2.2 in the method section had clarified this point.

**L314: "One difference between these two models is. . ." What two models?**

This sentence has been changed accordingly (4.1, L403-404):

> "One difference with Hashimoto et al. (2015)'s $R_h$ model is the lack of moisture limitation on respiration in CARDAMOM."

**L317: I'd recommend making this sentence the start of the next paragraph**

This sentence has been also changed accordingly to properly conclude the paragraph (4.1, L404-407):

"Conversely, GPP is relatively well-constrained in space through the assimilation of LAI and a prior for productivity (Bloom et al., 2016), although an important mismatch has been found: CARDAMOM GPP is 50% lower than FLUXCOM, but 30% higher than FLUXNET2015 EC data."

**L319: How do you know that the issue is only one of scale difference, and not some other error in the model or DA system? What could you do to confirm this (e.g. run with local drivers)?**

We cannot be certain here, but mismatch on LAI due to local variability seems the most likely cause of mismatch. We could confirm this by running using LAI data directly determined from the study site at an appropriate scale.

**L326: This error in timing is an example of why it might be better to run a system that performs both state and parameter data assimilation, rather than just parameters.**

We believe parameter estimation should be acceptable – given that the parameters we incorporate for calibration include those that drive phenology, and thus state changes.

**L328: It's a bit surprising that you're running a model in the arctic that doesn't include snow or permafrost. I see that this point is in the Discussion, but it seems really important to be more upfront about this earlier in the paper, as it's a pretty limiting assumption and should lead to greater caution in how confidently you interpret the results. It also begs the question as to why you didn't couple CARDOMOM to a more sophisticated land model for this analysis.**

Our goal is to use a C model as simple as possible, with strong data constraint. DALEC has direct state variable linkage to LAI, biomass, SOM. Key processes are climate-sensitive (GPP, $R_h$). For forecasts of the future C cycle we agree that simulating the changes to permafrost and hydrology would be vital, but for analyses of the current C cycle and its internal dynamics, snow and permafrost are secondary factors.

**L365: But is there any direct field constraint (e.g. isotope data)**

We recognise the value of using independent data on turnover rates (e.g. from isotopes) to evaluate analytical estimates. Such comparison was beyond the scope of this paper, but is the target of current and future research.
* * *
**Interactive comment on Earth Syst. Dynam. Discuss., https://doi.org/10.5194/esd-2018-19, 2018.**

---

## Author Response (AR3)

Report #1 on "Evaluation of terrestrial pan-Arctic carbon cycling using a dataassimilation system" by Efrén López-Blanco et al.

M. Forkel (Referee) matthias.forkel@geo.tuwien.ac.at Submitted on 19 Nov 2018

The authors substantially revised the manuscript and addressed my comments appropriately.

I only disagree how uncertainties where used during the data assimilation. Specifically, the authors state at several places (lines 86, 184-85) that the (biomass) dataset lacks uncertainty or error estimates and hence they used a global uncertainty factor of 1.5 in the cost function.

It is clearly a wrong statement that the biomass maps by Carvalhais et al. (2014) miss uncertainty estimates.

In this dataset, uncertainty was provided based on an ensemble of biomass estimates. This biomass map is also based on the map of forest biomass by Thurner et al. (2014) which also includes a detailed estimate of uncertainties for various vegetation carbon pools.

Please remove the wrong statements about missing uncertainty estimates for the biomass datasets and describe why you did not use these uncertainty estimates or how a potential use could affect your results. With these changes, I'm happy to accept the manuscript for publication.

We apologise for the lack of clarity about uncertainty derivation for the analysis. Here we have adjusted the text on the Introduction section (S1) to remove the sentence

"However, these products tend to lack clear error estimates."

On S2.2.2, L188-195 we have adjusted the text to:

"The reported uncertainty on biomass data from Thurner et al. (2014) was +/- 37% at pixel scale. Because of undetermined errors related to tree cover thresholds used in the upscaling, and to reflect unknown model structural error, we slightly inflate the error estimate and use a log-transform(1.5) of ×/ $\div$ 1.5 (i.e. ×/ $\div$ 1.5 spans 67% of the expected error). We use the same proportional error for SOC. For MODIS LAI we inflate the proportional error further to log(2) based on well reported biases in this product for evergreen forests (De Kauwe et al. 2011) and the estimated measurement and aggregation uncertainty for boreal forest LAI of 1 m2 m-2 reported by Goulden et al. (2011). The uncertainty assumptions in expression 3 are chosen in lack of better knowledge about the combined uncertainties arising from model representation errors and observation errors:"

In the Discussion we also now review the challenges associated with generating observation and model errors (see response to reviewer 2).

Report #2 on "Evaluation of terrestrial pan-Arctic carbon cycling using a dataassimilation system" by Efrén López-Blanco et al.

Anonymous Referee #2

Received and published: 06 Feb 2019

I'm going to be upfront that I'm very torn about what to recommend with respect to this paper. On the one hand, I acknowledge the incredible amount of work that went into this project and believe that there is important and interesting science coming out of this project. On the other hand, based on the responses to questions raised, it is now clear there are definitely things here that I don't think were done correctly. What complicates this is that many of the things done wrong (especially with respect to model process error) were also done wrong in previous papers on the Bayesian calibration of terrestrial carbon models (both by this team and others). This helps explain such mistakes, but it doesn't justify them, and I worry that continuing to allow papers to make the same mistakes just perpetuates the situation. The crux of the issue is really in how the authors are treating the error term in their likelihood. First, they are ascribing 100% of the error as coming from the observations, and not acknowledging (statistically) that their model is imperfect (though their own Results and Discussion clearly demonstrate that the model is far from perfect). By incorrectly ascribing 100% of the error to observations, and none to process error (model misspecification, stochastic events, unaccounted for heterogeneity), the authors are also missing that (unlike observation error) process error propagates forward into model predictions. This means that modeled fluxes and pools are going to be consistently overconfident by an unknown (but potentially nontrivial) amount. Second, not only do the author ascribe all the error to observations, but they treat that observation error as a known parameter, despite acknowledging that the data products used don't have error estimates. This is a significant departure from standard statistical modeling, where the variance is an unknown fit parameter. For example, when you fit a linear regression the model has three unknown parameters (slope, intercept, sigma) and sigma is virtually never treated as an a prior known quantity. While treating sigma as a known shouldn't have large effects on the mean values of the model parameters (though this is far from guaranteed when dealing with nonlinear models; Jensen's Inequality), more important is that it can have a real effect on the uncertainties about the model parameters. By subjectively choosing the observation error, one is also subjectively choosing the confidence intervals on the parameters. And since in CARDAMOM the only uncertainties that are included in predictions are parameter uncertainties, this also means you are subjectively choosing the uncertainty in the predictive confidence intervals. Ideally, these models should be refit including an unknown, fit model process error, and then that process error should be propagated into predictions/hindcasts. This process error ideally should also be in addition to, not instead of, an observation error (which may not be a known, but may have an informative prior on it)

We recognise the reviewer's concerns about using the correct process for error characterisation in analyses such as that we present here. We agree that our model is not perfect and that identification of process error is critical. We also regret that we did not provide the necessary information on how data uncertainties were derived. We do appreciate the reviewer's concern about effective error characterisation, and have adjusted the text to reflect this, and to make recommendations about how to address this better.

We did specifically focus on identification of model process error by comparison with independent data (GPP, Rh). Thus, we identified biases in our estimates of LAI, GPP and biomass at landscape scale, and suggest that these likely reflect systematic bias in our photosynthesis model. A next step is to analyse the representation of photosynthesis process error and include this in further analyses. On the other hand, we note that independent evaluation of fluxes at site scale (FLUXNET2015) does not match the GPP bias at landscape pixel scale (FLUXCOM). New site level comparisons (see below) also suggest CARDAMOM produces reasonable or slightly high biased results. We conclude that further investigations into heterogeneity error are required, linked to process error calculation on products such as FLUXCOM as well as our GPP model.

We have adjusted the text (S2.2.2, L188-195) to clearly state that error in the biomass product is reported, and have explained why we have inflated this error in our analysis. We also note that MODIS LAI products have large reported biases, and local observations have important errors, which justifies the larger error we assigned to these data. Our point here is to report an honest overview of uncertainty assumptions used in CARDAMOM:

"The reported uncertainty on biomass data from Thurner et al. (2014) was +/- 37% at pixel scale. Because of undetermined errors related to tree cover thresholds used in the upscaling, and to reflect unknown model structural error, we slightly inflate the error estimate and use a log-transform(1.5) of ×/ $\div$ 1.5 (i.e. ×/ $\div$ 1.5 spans 67% of the expected error). We use the same proportional error for SOC. For MODIS LAI we inflate the proportional error further to log(2) based on well reported biases in this product for evergreen forests (De Kauwe et al. 2011) and the estimated measurement and aggregation uncertainty for boreal forest LAI of 1 m2 m-2 reported by Goulden et al. (2011). The uncertainty assumptions in expression 3 are chosen in lack of better knowledge about the combined uncertainties arising from model representation errors and observation errors:"

We note the reviewer's concerns about making forecasts without properly accounting for model process error. This paper involves an analysis of historical fluxes constrained by contemporary forcing and data. We do not make forecasts or hindcasts, so this criticism is not relevant for this paper.

We have adjusted the text in the discussion (S4.3; L487-501) to reflect the lack of robust knowledge on the interactions between random and systematic biases in the observations, model representation errors and errors in the model drivers:

"Our approach has used estimated observation error, and inflated this to include unknown errors associated with model process representation. We currently lack any better knowledge of the combined uncertainties arising from model representation errors and observation errors. We acknowledge that all models are an imperfect representation of C dynamics, which generates irreconcilable model-data errors due to the inherent assumptions in model structure. Future analyses should investigate model structural error, using for example error-explicit Bayesian approaches (Xu et al., 2017), or comparing the likelihoods of alternate model structures, of varying complexity. Using multiple sources of data, we have highlighted systematic errors in the model at landscape scale (Figure 2 and 3) for LAI, GPP and biomass. However, these biases are not consistent for site-scale evaluations. Thus, a next step would be to include explicitly both random and systematic process errors for C fluxes in the data assimilation. These errors could be determined from field scale evaluation of model process representation (Table 2) using e.g. FLUXNET2015 data. We also need to understand better the error associated with landscape heterogeneity of C stocks and fluxes, to upscale from flux tower observations, or direct measurements of LAI, to landscape pixel. This could be achieved by constructing robust observation error models (Dietze, 2017) from field to pixel scale, for e.g. GPP, LAI and foliar N. Evaluation of the sensitivity of C cycling DA analyses to observation error has shown relatively low sensitivity to data gaps and random error on net ecosystem flux data (Hill et al., 2012), but further analyses of error sensitivity are required for multiple streams of stock data."

**Additional points of concern:**

1) Neither the DALEC2 model nor the CARDAMOM system appear to be publically archived. This means this work can't be reproduced or expanded upon by others. I don't know if such lack of openness is within the letter of the law of this journal, but it's definitely a deviation from the current norms of the community.

We agree that openness is critical to scientific advances. We have submitted the code for DALEC2 on Edinburgh DataShare. We are working to release a community version of CARDAMOM. At present we invite researchers to contact us to gain access to the code.

We have adjusted the text (L517-520):

**"Data and software availability**

CARDAMOM output used in this study is available from Exbrayat and Williams (2018) from the University of Edinburgh's DataShare service at https://doi.org/10.7488/ds/2334. The DALEC2 code is also available on Edinburgh DataShare at https://doi.org/10.7488/ds/2504. Contact MW for access to the CARDAMOM software."

2) As noted in my original review, I'm not comfortable with this system being called data assimilation, at least not with some additional qualifier being added (e.g. "parameter data assimilation") to make it clear that the outputs are deterministic model forward simulations not a reanalysis. To me, calling this data assimilation is like calling linear regression "machine learning." Sure people do it, but it makes the term pretty meaningless.

We disagree; we are using Bayesian parameter calibration of a dynamic model - which is typically referred to as data assimilation or model-data fusion; see "Ecological Forecasting" p. 168, by M. Dietze. However, we adjust our introductory text to improve clarity (S1; L100-104):

"To address these issues we integrate model and data more formally. We apply data assimilation (DA), defined as a Bayesian calibration process for a model of a dynamic

system. DA, through probabilistic parameterisation, supports robust model estimates of C stocks and fluxes consistent with multiple observations and their errors (Fox et al., 2009; Luo et al., 2009; Williams et al., 2005). By following Bayesian methods, the uncertainty on observations weights the degree of data constraint, and the outcome is a set of acceptable parameterisations for a given model structure linked to likelihoods."

3) After clearly diagnosing your photosynthesis scheme (ACM) as being at the root of model biases and compensating errors, the decision to not include any ACM parameters in the calibration (and toss the issue up to a lack of acclimation rather than simple miscalibration) strikes me as odd and I cannot understand why the authors are digging in their heels on this.

We do include an ACM parameter ( $C_{eff}$ ) in the calibration (and so it is adjusted by the MHMCMC), according to Bloom et al. (2016). We apologise for not making this clear. We consequently have adjusted the Methods text (S2.2.1; L143-145) to read:

"DALEC2 simulates canopy-level GPP via the Aggregated Canopy Model (ACM; Williams et al., 1997) and the most sensitive ACM parameter, related to canopy photosynthetic efficiency, is included in the CARDAMOM calibration."

4) Similar to (3), since NPP in DALEC is very tightly tied to GPP, and TT = Cstock/NPP, it sure seems like systematic biases in GPP will translate to systematic biases in TT. As noted earlier, I find some of the reported TT estimates to be implausible and don't understand the authors resistance to even considering comparing their results to independent field estimates.

We note that the mean NPP for GVMs across the region is 8% lower than in CARDAMOM, so the regional GVM-CARDAMOM NPP analyses are less different on average than the comparisons of CARDAMOM against data such as FLUXCOM (for GPP). We note that the high latitude TT estimates for CARDAMOM, GVMs (Figure 5) and reported in Carvalhais et al. (2014) are broadly similar. The critical issue we identify is that the spatial differences in NPP and Cveg between CARDAMOM and GVMs result in important spatial mismatches in TT estimated by both (compare Figure 5 and Figure 6).

We are confused at the statement that we have "**resistance to even considering comparing their results to independent field estimates**"; we have presented a clear evaluation against multiple independent FLUXNET site data, shown in Figure 4. Nonetheless, we add some further field-based estimates to complement these comparisons in the Discussion (S4.1, L421-435):

"For a further independent evaluation of CARDAMOM outputs, we compare the tundra and boreal estimates to plot scale flux and stock information. For tundra, Street et al. (2012) calculate growing season GPP estimates of 263-380 g C m-2 for *Empetrum nigrum* communities, and 295-386 g C m-2 for *Betula nana* communities, which is consistent with the ranges in Figure 1 for tundra. Biomass stocks for Arctic tundra recorded in the Arctic LTER at Toolik Lake range from 105-1160 g C m-2 (Hobbie and Kling, 2014), which are consistent with the estimates from CARDAMOM, albeit at the lower end of the model estimates. For boreal forests, Goulden et al. (2011) report annual GPP estimates across a chronosequence of stands, and thus a variation across canopy densities, which varied from 450-720 g C m-2 yr-1. These data are consistent with the span of GPP in CARDAMOM (Figure 1), again best matching the lower end of the model estimates. For the same study, the vegetation C stock estimates varied from 100-5000 g C m-2, consistent with CARDAMOM, and with measurements of 10 to 40-year old boreal stands best matching the CARDAMOM median estimate of ~1500 g C m-2. We conclude from comparisons against site data that CARDAMOM analyses are broadly consistent, with some tendency for CARDAMOM to have a high bias. This comparison is similar to the FLUXNET2015 evaluation of CARDAMOM. But it conflicts with the estimation of low bias from the comparison of CARDAMOM against FLUXCOM GPP and Carvalhais et al. (2014) biomass stock maps. It is possible that the scale differences between site level products and landscape estimates is confusing these comparisons, but there is clearly a need to understand better these inconsistencies in C cycle estimates."

**5) The differences between DALEC and observations are greater than the differences between DALEC and the ISI-MIP models, so why are the authors so hard on the ISI-MIP models?**

Our key point is that DALEC outputs match the spatial variation in independent (FLUXCOM) and assimilated data (LAI, biomass) well. There may be biases in these comparisons, indicative of model process error and/or upscaling error in the biomass and FLUXCOM products, but CARDAMOM can match the pattern in LAI, biomass, and SOC very well (Figure 2). The poor agreement with ISI-MIP models is with the spatial pattern (Table 3), not with regional median values (Figure 5). From these analyses we note that a reasonable regional estimate is not very useful if patterns are wrong, as this challenges the reliability of ISIMIP models when used for projections. Some models actually match CARDAMOM well, and we noted this clearly. We have edited the text to emphasise these points:

In Results (S3.4, L318-330):

"We used our highest confidence retrievals of NPP,  $C_{veg}$  and  $TT_{veg}$  (i.e. retrievals including assimilated LAI, biomass and SOC) to benchmark the performance of the GVMs from the ISI-MIP2a project. In this assessment we compared not only their spatial variability across the pan-Arctic, tundra and taiga region (Figure 5), but also the degree of agreement between their mean model ensemble within the 90% confidence interval of our assimilation framework (Figure 6, Table 3). NPP estimates (RMSE = 0.1 kg C m-2 yr-1; R2= 0.44) are in better agreement than  $C_{veg}$  (RMSE = 1.8 kg C m-2; R2= 0.22) and TTveg (RMSE = 4.1 years; R2= 0.12). The assessed GVMs estimated on average 8% lower NPP, 16% higher  $C_{veg}$  and 22% longer TTveg than CARDAMOM across the entire pan-Arctic domain (Figure 5 and 6) on average. Thus, at regional aggregation CARDAMOM analyses agreed more closely with ISI-MIP2a models than with FLUXCOM (51% difference) and with the Carvalhais et al. (2014) biomass data (28% bias).

The poor spatial agreement regarding  $TT_{veg}$  between CARDAMOM and ISI-MIP2a (Table 3) is indicative of uncertainties in the internal C dynamics of these models. For instance, the slopes in Table 3 are steep and the R2 are poor – so there is a substantial disagreement in the spatial pattern, not just a large bias. For ISI-MIP2a comparison R2 values ranged from 0.03-0.52 for NPP; 0.00-0.31 for Cveg; and 0.00-0.24 for TTveg."

In Discussion (S4.3, L449-451):

"Using CARDAMOM as a benchmarking tool for six GVMs we found disagreements that varied among models for spatial estimates of NPP,  $C_{veg}$  and  $TT_{veg}$  across the Pan-Arctic (Figure 6) in comparison against CARDAMOM confidence intervals."

**Detailed comments:**

L60: The authors responses suggested that a more complex calculation of TT was actually performed that relaxed the assumption of steady state. I would include that here (along with the steady state calculation) as I suspect a number of readers (myself included) would prefer to know that you're not relying on a steady state assumption to assess a system that's clearly not in steady state.

The residence time is calculated as per Bloom et al. (2016) equation S8 (SI text, S3 Global State and Process Variables), which specifically accounts for changes in stocks over time. We now adjust the text accordingly in the Introduction (S1) by removing "at steady state" and the Methods (S2.2.2; L202-203):

"We calculate the transit time for C pools using the approach for non-steady state pools described in Bloom et al. (2016), supplementary information S3."

**L160: This line refers to DALEC2 as an 'intermediate complexity' model, but later arguments actually hinge on it being a simple model, and most of us would consider DALEC to really be on the simple end of the process-model spectrum**

We have had internal discussions about where on the spectrum of complexity DALEC lies. We have decided that simple models would have only a handful of parameters and few state variables. DALEC has 17 parameters and 6 state variables, so it just qualifies as intermediate. We agree that this is partially a subjective categorisation (now in S2.2.2; L157). We also changed wording in L110, L447, L474, and L478 to keep consistency across the full text.

**L171: MODIS LAI reports an uncertainty estimate. How did you aggregate those uncertainties when aggregated the observations? This is nontrivial as neither the MODIS products or MODIS LAI validation papers report anything about the spatial or temporal autocorrelation in the product's errors.**

We have adjusted our text to report on MODIS uncertainties (S2.2.2, L191-193):

"For MODIS LAI we inflate the proportional error further to log(2) based on well reported biases in this product for evergreen forests (De Kauwe et al. 2011) and the estimated measurement and aggregation uncertainty for boreal forest LAI of 1 m2 m-2 reported by Goulden et al. (2011)."

We have also adjusted the discussion to note the challenge for scaling these errors (S4.3, L496-499):

"We also need to understand better the error associated with landscape heterogeneity of C stocks and fluxes, to upscale from flux tower observations, or direct measurements

of LAI, to landscape pixel. This could be achieved by constructing robust observation error models (Dietze, 2017) from field to pixel scale, for e.g. GPP, LAI and foliar N."

L188: Table S2 looks like it just contains a bunch of uniform priors for all other parameters. I think that should be stated here so that readers don't need to find the supplement to learn that. It's perfectly fair, however, to make readers go to the supplement to see the exact numerical values of the priors.

We now include a note (S2.2.1, L143):

"(Table S2; most priors are uniform with broad ranges)"

Moreover, we corrected a mistake with C pools units in Table S2. We replaced g C m-2 yr with g C m-2.

**L194: This sentence states that MODIS doesn't report an uncertainty estimate, but that's not accurate.**

The cited statement was removed and we have adjusted (see above) our text to report on MODIS uncertainties (S2.2.2, L191-193):

"For MODIS LAI we inflate the proportional error further to log(2) based on well reported biases in this product for evergreen forests (De Kauwe et al. 2011) and the estimated measurement and aggregation uncertainty for boreal forest LAI of 1 m2 m-2 reported by Goulden et al. (2011)."

L206: I'm concerned about the way the statistics are being reported here. For example, the RMSE of a model is traditionally based on the model error (difference between the model and the observations). Here, the authors are defining the model's RMSE as the RMSE after applying both a multiplicative and additive bias correction (i.e. the predicted/observed regression). Similarly, the R2 isn't the variance explained by the model, but the variance jointly explained by the model and a linear bias correction to that model. This results in a very optimistic view of the model's actual performance.

We have calculated RMSE following the traditional approach, and we have adjusted the text to clarify this (S2.3, L207-209):

"To assess the degree of statistical agreement we calculated linear goodness-of-fit (slope, intercept,  $R^2$ ) between CARDAMOM and the two independent datasets and determined RMSE and bias from direct comparison on model-data residuals."

Following the same logic, we have also clarified this in S2.3, L221-223:

"We performed a point-to-grid cell comparison to assess the degree of agreement between each flux magnitude and seasonality calculating the statistics of linear fit (slope, intercept, R2) per flux and site between CARDAMOM and FLUXNET2015 datasets and determined RMSE and bias from model-data residuals comparison."

L251: Just want to continue to express my skepticism about some of these pool and flux estimates. For example, in my own experiences in Alaska, the boreal forest has WAY

**more than 160% more structural tissue than the tundra. There needs to be some independent plot-scale validation of this.**

Independent data from Toolik Lake (tundra) and Boreas (boreal) sites shows the general validity of the CARDAMOM outputs at these intensively studied ecological field sites.

As we presented earlier on, we included the following text in S4.1, L421-430:

"For a further independent evaluation of CARDAMOM outputs, we compare the tundra and boreal estimates to plot scale flux and stock information. For tundra, Street et al. (2012) calculate growing season GPP estimates of 263-380 g C m-2 for *Empetrum nigrum* communities, and 295-386 g C m-2 for *Betula nana* communities, which is consistent with the ranges in Figure 1 for tundra. Biomass stocks for Arctic tundra recorded in the Arctic LTER at Toolik Lake range from 105-1160 g C m-2 (Hobbie and Kling, 2014), which are consistent with the estimates from CARDAMOM, albeit at the lower end of the model estimates. For boreal forests, Goulden et al. (2011) report annual GPP estimates across a chronosequence of stands, and thus a variation across canopy densities, which varied from 450-720 g C m-2 yr-1. These data are consistent with the span of GPP in CARDAMOM (Figure 1), again best matching the lower end of the model estimates. For the same study, the vegetation C stock estimates varied from 100-5000 g C m-2, consistent with CARDAMOM, and with measurements of 10 to 40-year old boreal stands best matching the CARDAMOM median estimate of ~1500 g C m-2".

**L258: Likewise, this stem turnover time seems much too fast and needs independent validation. I understand that grid cell to plot- or plant-scale validation isn't perfect, but it's better to report the performance explicitly, and then cushion it based on possible scale mismatch, rather than to ignore whether these estimates are consistent with prior research.**

Based on comparison to Carvalhais et al. (2014) TT estimates and to the GPP and  $C_{veg}$  estimates reported above, our TT estimates are consistent with independent calculations and their component parts. We understand that TT seem short compared to concepts of stand age. However, litterfall (plant mortality) occurs throughout succession, from all live pools, which means that C turns over faster than age suggests.

**L294: typo on "uncertainties"**

Corrected.

**L313: It would be good to have some sort of quantification of spatial coherence beyond RMSE & R2 (which are nonspatial). Look to the GIS and remote sensing literature for examples of what sort of statistics are available to do this.**

There are a number of potential statistics to use. We suggest that our choice of statistics is familiar to biogeochemists and earth system scientists. Coupled with direct mapping of ratios and confidence intervals for visual assessment, we suggest our analysis provides readers with the relevant information on spatial coherence. Adding further statistics is likely to provide only marginal gains, but also increase the intricacy of an already complex paper.

**L328: Don't introduce new Methods in the Results. Please document what this analysis is and why you are doing it earlier in the paper.**

We agree the reviewer 2 is correct and we have adjusted the text as requested, moving material into the last part of the Methods (S2.4, L235-239):

"To understand the sources of errors in  $TT_{veg}$  calculations, we used CARDAMOM to calculate two hypothetical  $TT_{veg}$  (i.e. EXPERIMENT A  $TT_{veg}$  = ISI-MIP2a  $C_{veg}$  / CARDAMOM NPP and EXPERIMENT B  $TT_{veg}$  = CARDAMOM  $C_{veg}$  / ISI-MIP2a NPP) and then assessed the largest difference with CARDAMOM's CONTROL  $TT_{veg}$ . We estimated the hypothetical  $TT_{veg}$  for each pixel in each model, and derived a pixelwise measure of the contribution of biases in NPP and  $C_{veg}$  to biases in  $TT_{veg}$  by overlapping their distribution functions."

**L378: Consistent with my previous concerns, DALEC appears to be running to fast. That said, this is still a comparison to other models, not to data.**

We agree that biases may exist in the CARDAMOM TT estimate, but see above about difference between stand age and TT (L258 comment). Also, note that we are exploring where ISI-MIP2a models lie outside the analysis confidence intervals of CARDAMOM for TT.

**L391: Here you say you had a 'strong prior on photosynthesis' but as far as I can tell the photosynthetic parameters were fixed at defaults, not assigned priors. According to Eqn 2, the only 2 parameters assigned non-uniform priors were canopy efficiency (which in Tables 2 and S2 is labeled as a phenology parameter) and autotrophic respiration**

As noted before, the canopy efficiency is the calibrated parameter in CARDAMOM for the photosynthesis model ACM; we apologise for confusion in not making this clear before. Now this point is clarified in text (S2.2.2, L143-145):

"DALEC2 simulates canopy-level GPP via the Aggregated Canopy Model (ACM; Williams et al., 1997) and the most sensitive ACM parameter, related to canopy photosynthetic efficiency, is included in the CARDAMOM calibration."

**L397: If you've demonstrated a bias in your photosynthetic model, I'm not sure I agree that this could be resolved with more precise data if you're not updating the parameters in the photosynthetic submodel**

Again, we have now clarified that a parameter in the photosynthesis model (canopy efficiency in ACM) is being updated by CARDAMOM.

**L427: I fundamentally disagree that models should be benchmarked against highlyderived, model-based data products. But this isn't the central point of the paper and thus I won't hold up this paper over that disagreement.**

Every data product used here is in some way model-derived – LAI from MODIS requires a model, biomass from radar and landcover maps also, SOC data from interpolation and machine learning approaches, even in-situ data such as GPP and  $R_{eco}$  are separated from NEE using a wide range of partitioning algorithms.

**L459: While it's true that brute-force MCMC is not feasible for complex models, but there are other options available that do work with larger models, such emulators (Fer et al 2018 Biogeoscience) and ensemble or particle filters.**

We agree that there are a range of alternative approaches beyond MCMC and decided to include a sentence in Discussion including reviewer 2's suggestion (S4.3; L476-477):

"Other viable options include using emulators (Fer et al., 2018) and particle filters (Arulampalam et al., 2002), but MCMC methods provide the most detailed description of error distributions."

We also re-arranged the following sentences and merged paragraphs to improve clarity (S4.3; L477-486):

"There remains a strong argument to utilize intermediate complexity models like DALEC2 to evaluate the minimum level of detail required to represent ecosystem processes consistent with local observations, and to allow testing of alternate model structures. And, assimilating further data products, for instance patterns in soil hydrology and snow states across the pan-Arctic from earth observation, could provide useful information on spatio-temporal controls on soil activity and microbial metabolism to constrain below ground processes. This information would need to be tied to process level information on SOM turnover generated from experimental studies, and included in updated versions of DALEC. Thus, more field observations are crucial across the pan-Arctic, specifically on decomposition and TT of SOC (He et al., 2016) and respiratory processes such as partitioning of Reco into Ra and Rh (Hobbie et al., 2000; McGuire et al., 2000), across the growing season and also during wintertime (Commane et al., 2017; Zona et al., 2016)."

**L477: For the record, if you didn't fit every grid cell independently then you wouldn't need to upscale/interpolate field observations.**

Our point is that critical ecological processes remain poorly understood, and so further field observations are required to constrain these processes. Also, if each pixel had not been treated independently, we would have then relied on PFTs with all their problems (clearly pointed at Introduction and Discussion sections), and which is basically the opposite to what CARDAMOM framework is about.

**L495: Where are the DALEC2 and CARDAMOM code repositories?**

Following up on reviewer 2 initial concern, we have submitted the code for DALEC2 on Edinburgh DataShare. We are working to release a community version of CARDAMOM.

We have adjusted the text (L517-520):

**"Data and software availability**

CARDAMOM output used in this study is available from Exbrayat and Williams (2018) from the University of Edinburgh's DataShare service at https://doi.org/10.7488/ds/2334. The DALEC2 code is also available on Edinburgh

DataShare at https://doi.org/10.7488/ds/2504. Contact MW for access to the CARDAMOM software."

**Table 2: I find it interesting that, given the papers focus on turnover times, turnover parameters are the least constrained part of the model.**

Yes, this is the case, and reinforces the focus on TT in this analysis – we will only improve forecasts of high latitude C dynamics from better understanding TT.

\_\_\_\_\_

**Evaluation of terrestrial pan-Arctic carbon cycling using a dataassimilation system**

Efrén López-Blanco1,2, Jean-François Exbrayat2,3, Magnus Lund1, Torben R. Christensen1,4, Mikkel P. Tamstorf1, Darren Slevin2, Gustaf Hugelius5, Anthony A. Bloom6, Mathew Williams2,3

[revised manuscript text omitted]

520 use efficiency (Manzoni et al., 2018), and or photosynthesis-respiration couplingdrivers of gross flux coupling (López-Blanco et al., 2017). There are opportunities to constrain such modelling using data on plant trait relationships across pan-Aretic regions (Reichstein et al., 2014; Sloan et al., 2013). We also need to assimilate data describing annual biomass maps, and landscape disturbances such as fires and moth outbreaks at the pan-Aretic scale. From a modelling perspective, wThus, me consider that more field observations are crucial across the pan-Aretic, specifically on decomposition and TT from plant and

- 525 soil\_C stocks decompositionof SOC (C stocks turnover rates)(He et al., 2016) and respiratory processes such as (partitioning of Reco into Ra and Rh) (Hobbie et al., 2000; McGuire et al., 2000), not only across the growing season, but and also during wintertime (Commane et al., 2017; Zona et al., 2016). These data could be upscaled using machine learning, following the approaches used for creating SOM maps, with uncertainty attribution, as further assimilation data sets for frameworks like CARDAMOM. An improved data-model integration will move us towards enhanced analytical robustness and a decrease of
- 530 model uncertainties.

Our approach has ascribed used estimated observation error, to observations only, and we have ignored and inflated this to include unknown errors associated with model process representation. We currently lack any better knowledge of the combined uncertainties arising from model representation errors and observation errors, stochastic events, and unaccounted for heterogeneity, for example. We acknowledge both that anyall models is are an imperfect representation of C dynamics,

- 535 which generates irreconcilable model-data errors due to the inherent assumptions in model structureand the need to quantify model process error. Future analyses should investigate model structural error, using for example error-explicit Bayesian approaches (Xu et al., 2017), or comparing the likelihoods of alternate model structures, of varying complexity. In our case, uUsing multiple sources of data, we have highlighted systematic errors in the model at landscape scale (Figure 2 and 3) for LAI, GPP and biomass, which may be related to the prior from the photosynthesis we used. However, these biases are not
- 540 consistent for site-scale evaluations. Thus, There is, therefore, irreconcilable model data errors due to the inherent assumptions in model structure, but attempts have been made pointing to potential methods for optimizing uncertainty choices (Caldararu et al., 2012) and error models in model data fusion systems (Schoups & Vrugt, 2010). We certainly need to consider these options as feasible ways to account for model structural errors in future implementations of CARDAMOM. aA next step for our analysis is towould be to include explicitly both random and systematic photosynthesis-process errors for C fluxes in the
- 545 data assimilation. These errors could be determined from field scale evaluation of model process representation (Table 2) using e.g. FLUXNET2015 data. However, wWe also need- to first-understand better the error associated with landscape heterogeneity of C stocks and fluxes, to upscale from flux tower observations, or direct measurements of LAI, to landscape pixel. This could be achieved by constructing robust observation error models (Dietze, 2017) from field to pixel scale, for e.g. GPP, LAI and foliar N. Evaluation of the sensitivity of C cycling DA analyses to observation error has shown relatively low sensitivity to
- 550 data gaps and random error on net ecosystem flux data (Hill et al., 2012), but further analyses of error sensitivity are required for multiple streams of stock data. We need to explain the contrasting photosynthesis biases at landscape (Figure. 3) and flux site scales (Figure. 4) in order to understand and scale the C flux process error.

**5** Conclusions**

- The Arctic is experiencing rapid environmental changes, which will influence the global C cycle. Using a dataassimilation framework we have evaluated the current state of key C flux, stocks and transit times for the pan-Arctic region, 2000-15. We found that the pan-Arctic was a likely sink of C, weaker in tundra and stronger in taiga, but uncertainties around the respiration losses are still large, and so the region could be a source of C. Comparisons with global and local scale datasets demonstrate the capabilities of CARDAMOM for analysing the C cycle in the Arctic domain. CARDAMOM is a dataconstrained and data-integrated analysis, evaluated for internal consistency, and is therefore a good candidate to benchmark
- 560 performance of global vegetation/ecosystem models. We conclude that a GVM bias found in transit time of vegetation C is the result of a joint combination of uncertainties from productivity processes and biomass in GVMs, and thus these are a major component of error in their forecasts. While spatial patterns in GVM predictions of NPP are reasonable, particularly in taiga,

they have significant biases against the CARDAMOM benchmark. Improved mapping of vegetation and soil C stocks and change over time is required for better analytical constraint. Moreover, future work is required on assimilating data on soil

565 hydrology, permafrost and snow dynamics to improve accuracy and decrease uncertainties on belowground processes. This work establishes the baseline for further process-based ecological analyses using the CARDAMOM data-assimilation system as a technique to constrain the pan-Arctic C cycle.

**Data and software availability**

CARDAMOM output used in this study is available from Exbrayat and Williams (2018) from the University of
 Edinburgh's DataShare service at <a href="http://dx.doi.org/10.7488/ds/2334">http://dx.doi.org/10.7488/ds/2334</a>. The <a href="http://dx.doi.org/10.7488/ds/2334">DALEC2</a>-code -is also available on Edinburgh DatasShare at <a href="https://doi.org/10.7488/ds/2504datashare.is.ed.ac.uk">http://dx.doi.org/10.7488/ds/2334</a>. The <a href="https://dx.doi.org/10.7488/ds/2334">DALEC2</a>-code -is also available on Edinburgh DatasShare at <a href="https://doi.org/10.7488/ds/2504datashare.is.ed.ac.uk">http://dx.doi.org/10.7488/ds/2334</a>. The <a href="https://doi.org/10.7488/ds/2504datashare.is.ed.ac.uk">DALEC2</a>-code -is also available on Edinburgh DatasShare at <a href="https://doi.org/10.7488/ds/2504datashare.is.ed.ac.uk">https://doi.org/10.7488/ds/2504datashare.is.ed.ac.uk</a>). Contact <a href="https://dx.doi.org/10.7488/ds/2504datashare.is.ed.ac.uk">MW for access to the CARDAMOM software.</a>

[revised manuscript text omitted]

770 D., Tian, H., Tilbrook, B., Tubiello, F. N., van der Laan-Luijkx, I. T., van der Werf, G. R., van Heuven, S., Viovy, N., Vuichard, N., Walker, A. P., Watson, A. J., Wiltshire, A. J., Zaehle, S., and Zhu, D.: Global Carbon Budget 2017, Earth Syst. Sci. Data, 10, 405-448, 10.5194/essd-10-405-2018, 2018.

Levy, P. E., Friend, A. D., White, A., and Cannell, M. G. R.: 'The Influence of Land Use Change On Global-Scale Fluxes of Carbon from Terrestrial Ecosystems', Climatic Change, 67, 185-209, 10.1007/s10584-004-2849-z, 2004.

175 López-Blanco, E., Lund, M., Williams, M., Tamstorf, M. P., Westergaard-Nielsen, A., Exbrayat, J. F., Hansen, B. U., and Christensen, T. R.: Exchange of CO2 in Arctic tundra: impacts of meteorological variations and biological disturbance, Biogeosciences, 14, 4467-4483, 10.5194/bg-14-4467-2017, 2017.

López-Blanco, E., Lund, M., Christensen, T. R., Tamstorf, M. P., Smallman, T. L., Slevin, D., Westergaard-Nielsen, A., Hansen, B. U., Abermann, J., and Williams, M.: Plant Traits are Key Determinants in Buffering the Meteorological Sensitivity of Net Carbon Exchanges of Arctic Tundra, Journal of Geophysical Research: Biogeosciences, 123, <a href="https://doi.org/10.1029/2018JG004386">https://doi.org/10.1029/2018JG004386</a>, 2018.

Lucht, W., Prentice, I. C., Myneni, R. B., Sitch, S., Friedlingstein, P., Cramer, W., Bousquet, P., Buermann, W., and Smith, B.: Climatic Control of the High-Latitude Vegetation Greening Trend and Pinatubo Effect, Science, 296, 1687-1689, 10.1126/science.1071828, 2002.

785 Lund, M., Falk, J. M., Friborg, T., Mbufong, H. N., Sigsgaard, C., Soegaard, H., and Tamstorf, M. P.: Trends in CO2 exchange in a high Arctic tundra heath, 2000-2010, Journal of Geophysical Research: Biogeosciences, 117, 2012.

Lund, M., Raundrup, K., Westergaard-Nielsen, A., López-Blanco, E., Nymand, J., and Aastrup, P.: Larval outbreaks in West Greenland: Instant and subsequent effects on tundra ecosystem productivity and CO(2) exchange, AMBIO, 46, 26-38, 10.1007/s13280-016-0863-9, 2017.

790 Luo, Y., Weng, E., Wu, X., Gao, C., Zhou, X., and Zhang, L.: Parameter identifiability, constraint, and equifinality in data assimilation with ecosystem models, Ecological Applications, 19, 571-574, doi:10.1890/08-0561.1, 2009. Mack, M. C., Bret-Harte, M. S., Hollingsworth, T. N., Jandt, R. R., Schuur, E. A. G., Shaver, G. R., and Verbyla, D. L.: Carbon loss from an unprecedented Arctic tundra wildfire, Nature, 475, 489, 10.1038/nature10283, 2011.

Manzoni, S., Čapek, P., Porada, P., Thurner, M., Winterdahl, M., Beer, C., Brüchert, V., Frouz, J., Herrmann, A. M., Lindahl,
B. D., Lyon, S. W., Šantrůčková, H., Vico, G., and Way, D.: Reviews and syntheses: Carbon use efficiency from organisms to ecosystems – definitions, theories, and empirical evidence, Biogeosciences, 15, 5929-5949, 10.5194/bg-15-5929-2018, 2018.

[revised manuscript text omitted]